# Understanding the neural code of stress to control anhedonia

Frances Xia[1,9], Valeria Fascianelli[2,3,9], Nina Vishwakarma[1,4], Frances Grace Ghinger[1], Andrew Kwon[1], Mark M. Gergues[1,4,5], Lahin K. Lalani[1,4,5], Stefano Fusi[2,3,6,7] & Mazen A. Kheirbek[1,4,5,8 ✉]

Anhedonia, the diminished drive to seek, value, and learn about rewards, is a core feature of major depressive disorder[1–3]. The neural underpinnings of anhedonia and how this emotional state drives behaviour remain unclear. Here we investigated the neural code of anhedonia by taking advantage of the fact that when mice are exposed to traumatic social stress, susceptible animals become socially withdrawn and anhedonic, whereas others remain resilient. By performing high-density electrophysiology to record neural activity patterns in the basolateral amygdala (BLA) and ventral CA1 (vCA1), we identified neural signatures of susceptibility and resilience. When mice actively sought rewards, BLA activity in resilient mice showed robust discrimination between reward choices. By contrast, susceptible mice exhibited a rumination-like signature, in which BLA neurons encoded the intention to switch or stay on a previously chosen reward. Manipulation of vCA1 inputs to the BLA in susceptible mice rescued dysfunctional neural dynamics, amplified dynamics associated with resilience, and reversed anhedonic behaviour. Finally, when animals were at rest, the spontaneous BLA activity of susceptible mice showed a greater number of distinct neural population states. This spontaneous activity allowed us to decode group identity and to infer whether a mouse had a history of stress better than behavioural outcomes alone. This work reveals population-level neural dynamics that explain individual differences in responses to traumatic stress, and suggests that modulating vCA1–BLA inputs can enhance resilience by regulating these dynamics.

A reduced ability to experience pleasure, termed anhedonia, is a core feature of depression. Besides blunting positive emotional responses to what should be pleasurable experiences, anhedonia also profoundly affects behaviour, diminishing the drive to seek rewards and causing deficits in reward learning and valuation[1–3]. This can be modelled in rodents using chronic stress: whereas some animals show resilience to prolonged stress, susceptible mice socially withdraw and become anhedonic, with less motivation to attain high-value rewards[3–5].

The neural dynamics that account for the behavioural differences in resilient and susceptible individuals remain unclear, and determining them may provide crucial insights into how this debilitating aspect of depression might be treated. In the extended brain network responsible for generating emotional and motivated behaviour, the reciprocally connected amygdala and ventral hippocampus are two crucial nodes[6–26]. In addition to its role in threat detection and anxiety-related behaviour, the BLA guides decision-making by generating outcome-specific representations of rewards[27–30]. vCA1 has been shown to encode stimuli that predict rewards and to drive reward-related approach behaviours[31–33]. However, how these reward-related functions of vCA1 and BLA are affected by changes in emotional state remain unclear.

Stimulus-evoked responses of individual neurons in the BLA and vCA1 have been well studied. However, substantially less is known about how animals' reward-related internal states are represented at the population level in the BLA and vCA1 and how these representations shape reward choice-related behaviour. It is also unclear how spontaneous activity patterns in the absence of task stimuli in the BLA or vCA1 may differ in mice susceptible or resilient to chronic stress, and whether targeted interventions in this circuit might reduce susceptibility to stress. As internal states can be detected and studied only by characterizing the correlated activity of multiple neurons, it is essential to record simultaneously from large numbers of neurons, and to analyse their activity at the population level. This approach can reveal dynamics of internal state more accurately than single-cell recordings. Therefore, we conducted high-density Neuropixels recordings[34] in vCA1 and BLA and used population decoders to analyse the reward-related and spontaneous dynamics of populations of neurons to identify distinctive neural signatures of susceptibility and resilience to chronic stress. Then we developed a new circuit-specific modulation approach to rescue aberrant BLA population dynamics and associated anhedonic behaviour in stress-susceptible mice.

[1]Department of Psychiatry and Behavioral Sciences, University of California, San Francisco, San Francisco, CA, USA. [2]Center for Theoretical Neuroscience, Columbia University, New York, NY, USA. [3]Zuckerman Mind Brain Behavior Institute, Columbia University, New York, NY, USA. [4]Neuroscience Graduate Program, University of California, San Francisco, San Francisco, CA, USA. [5]Weill Institute for Neurosciences, University of California, San Francisco, San Francisco, CA, USA. [6]Department of Neuroscience, Vagelos College of Physicians and Surgeons, Columbia University Irving Medical Center, New York, NY, USA. [7]Kavli Institute for Brain Science, Columbia University Irving Medical Center, New York, NY, USA. [8]Kavli Institute for Fundamental Neuroscience, University of California, San Francisco, San Francisco, CA, USA. [9]These authors contributed equally: Frances Xia, Valeria Fascianelli. ✉e-mail: mazen.kheirbek@ucsf.edu

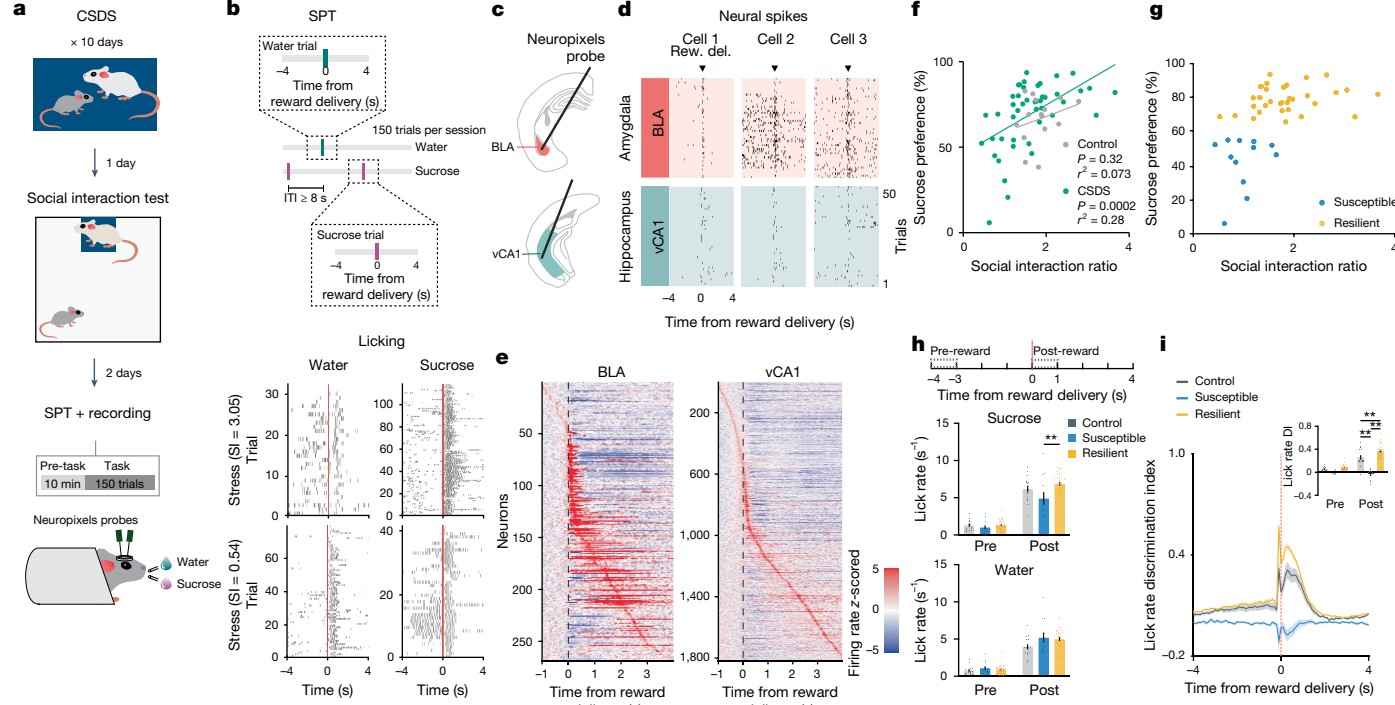

**Fig. 1 | Distinct behavioural signatures of resilient and susceptible mice following CSDS. a**, Schematic of SPT and Neuropixels recording protocol following CSDS. **b**, Schematic of SPT protocol. Example lick rasters are from two example mice with different sucrose preferences. Top, from a stress-resilient mouse with high sucrose preference. Bottom, from a susceptible mouse with low preference. ITI, inter-trial interval; SI, social interaction score. **c**, Neuropixels probes were targeted to BLA and vCA1. **d**, Example BLA and vCA1 spike rasters during task. Rew. del., reward delivery. **e**, Example peristimulus time histogram of BLA and vCA1 neurons around the time of reward delivery (time 0). **f**, CSDS ($n = 45$), but not control ($n = 15$), mice showed a significant correlation between sucrose preference and social interaction scores (Pearson correlation). **g**, Unsupervised $K$-means clustering revealed two distinct subgroups of CSDS mice ($n = 45$). **h**, Susceptible mice ($n = 12$) showed reduced sucrose lick rate during post-reward period compared to resilient mice ($n = 33$, repeated measures analysis of variance (ANOVA), group × time interaction: $F_{2,57} = 5.63$, $P = 0.0059$). **i**, Susceptible mice ($n = 12$) showed reduced lick rate discrimination index (DI) compared to control ($n = 15$) and resilient ($n = 33$) mice (two-way repeated measures ANOVA, group × time interaction: $F_{2,57} = 48.47$, $P < 0.0001$). Inset shows averaged lick rate DI during pre- and post-reward. Data are mean ± s.e.m. **$P < 0.01$.

## Behavioural classification following stress

To search for neural correlates of differential emotional and behavioural responses to traumatic social stress, we performed high-density single-unit electrophysiology using Neuropixels probes acutely inserted in the BLA and vCA1 of control mice and of mice subjected to chronic social defeat stress (CSDS; Fig. 1a–e). Activity was recorded during both a task- and stimulus-free condition and while mice performed a new head-fixed sucrose preference test (SPT). In this test, mice could freely choose to access either water or sucrose rewards by licking at the respective spout to trigger reward delivery. CSDS produced mice with varying degrees of sucrose preference and social interaction scores, which were highly correlated in both males and females (Fig. 1f and Extended Data Fig. 1a,b). These behavioural profiles allowed us to classify mice as stress resilient or susceptible (Fig. 1g). The susceptible mice identified using this classification showed lower lick rates during sucrose consumption, as well as markedly reduced lick rate discrimination between sucrose and water rewards—two behavioural subcomponents that suggest avoidance of higher value reward that is reflective of anhedonia[3,35,36] (Fig. 1h,i and Extended Data Fig. 1c–e).

## Reward discrimination in resilience

As we observed robust sucrose-seeking behaviours in resilient mice compared to susceptible mice, we looked for specific features of how rewards and reward-seeking behaviour were represented across the recorded BLA and vCA1 neuronal populations. We performed recordings as mice freely chose water or sucrose and indicated their choice

by licking a spout to trigger reward delivery (Fig. 2a). To assess neural activity patterns both before the mice behaviourally made their choices and after they consumed the reward, we defined a trial (sucrose or water) using the 8 s time window (4 s pre- to 4 s post-reward) around reward delivery.

First we quantified the proportion of reward-choice-selective neurons in the BLA and vCA1, defined as those that showed differential firing during water versus sucrose trials. In the BLA, during both the seconds before reward delivery (pre-reward) and the reward consumption period (post-reward), resilient mice had the greatest proportion of reward-choice-selective neurons compared with control or susceptible mice (Fig. 2b,c). In vCA1, we found that stress exposure increased the proportion of reward-choice-selective neurons in all previously stressed mice (that is, resilient and susceptible groups) in comparison to controls.

As the single-neuron analysis takes into account only a small subset of selective neurons, and the entire population is likely to contain relevant task-related information[37], we next investigated the differences in reward-choice coding between groups at the population level. We trained linear classifiers to discriminate trial types (water or sucrose choice), balancing the number of current and past rewards for each trial type (Methods and Fig. 2d). When analysing activity during pre-reward time bins, we again found distinctive signatures of stress resilience. In resilient mice, the upcoming choice of sucrose or water could be decoded from neural activity in BLA better than chance and better than from neurons in control or susceptible mice (Fig. 2e,f).

After reward consumption (post-reward), reward choice could be decoded from BLA activity better in all mice, but decoding was still

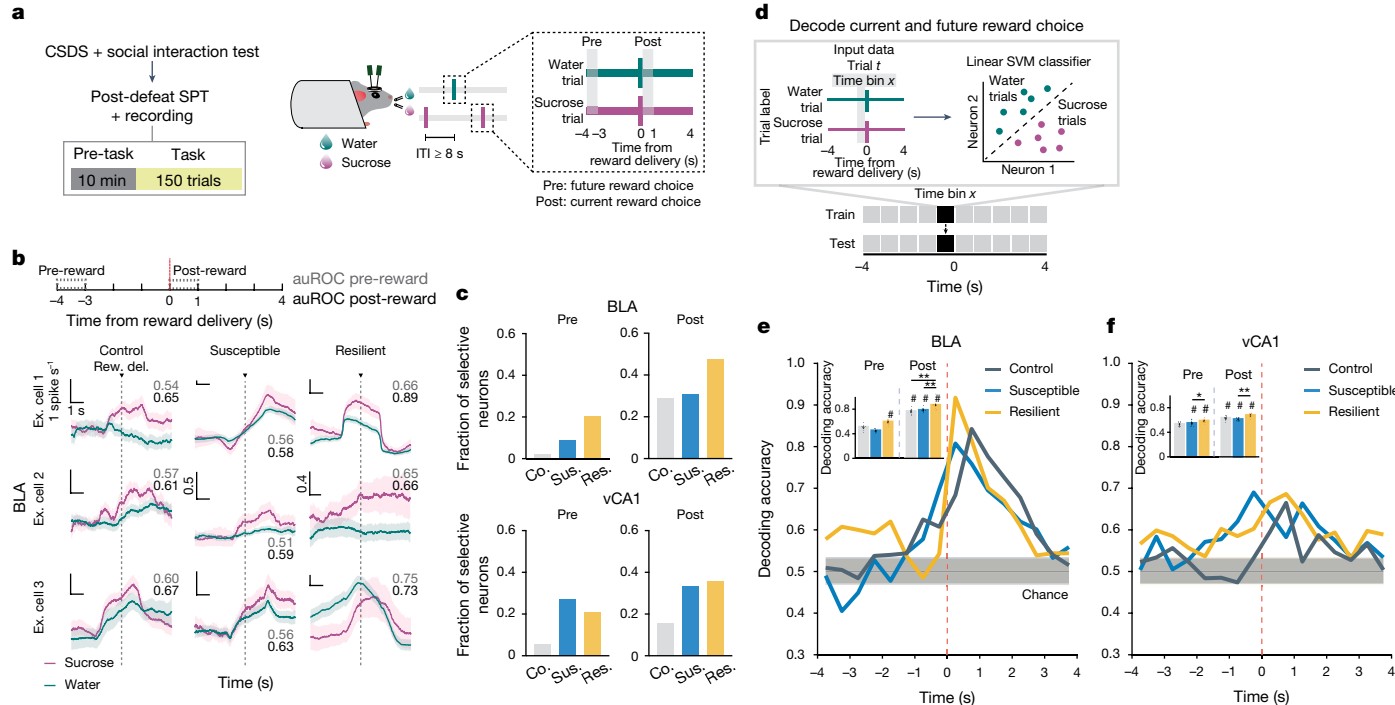

**Fig. 2 | Enhanced representations of reward choice in stress resilient mice.**
**a**, Schematic of SPT. **b**, Trial-averaged firing rates in sucrose and water trials from example (Ex.) BLA cells, with respective area under the receiver-operating-characteristic (auROC) curve during pre-reward (grey, −4 to −3 s) and post-reward (black, 0 to 1 s). Scale bars, 1 spike s$^{-1}$ and 1 s unless otherwise stated. **c**, In BLA, resilient group showed the highest fraction of selective neurons during both pre-reward (control (Co.) n = 132 neurons, susceptible (Sus.) n = 68 neurons, resilient (Res.) n = 69 neurons, Fisher's exact tests, resilient versus control, P < 0.0001, chi-squared, resilient versus susceptible, P = 0.057) and post-reward (chi-squared, resilient versus control, P = 0.0073, resilient versus susceptible, P = 0.042). In vCA1, both susceptible (n = 283 total neurons) and resilient (n = 528 total neurons) groups showed higher fraction of selective neurons than controls (n = 143 total neurons) during pre- (chi-squared, P < 0.0001) and post-reward (chi-squared, P < 0.0001). **d**, Schematic of

population decoding of current and future reward choices. Linear support vector machine (SVM) classifier was trained to distinguish between water versus sucrose trials. **e**, In BLA, resilient mice showed higher decoding accuracy than chance during pre-reward, and the highest decoding accuracy among all groups during post-reward (coloured lines indicate mean of subsampling (n = 10 of 60 neurons, with 100 cross-validations, Kruskal–Wallis, P < 0.0001)). Inset shows averaged decoding accuracy during pre- and post-reward. **f**, In vCA1, resilient mice showed higher decoding accuracy than susceptible mice during pre-reward (coloured lines indicate mean of subsampling (n = 10 of 60 neurons, with 100 cross-validations, Mann–Whitney, P = 0.045) and post-reward (Kruskal–Wallis, P = 0.0011). Inset shows averaged decoding accuracy during pre- and post-reward. Data are mean ± s.e.m. $^{\#}$Significantly different from chance; *P < 0.05; **P < 0.01.

strongest in resilient mice. The enhanced reward choice decoding in resilient mice was not driven by differences in direction coding (lick left versus lick right; Methods and Extended Data Fig. 2a–c) or lick rates (Methods and Extended Data Fig. 2d,e). A similar, although less pronounced, reward choice decoding pattern was observed in vCA1. These results indicate that neurons in the BLA, and to a lesser extent in vCA1, of resilient mice showed enhanced discrimination of reward choices both before and during reward consumption.

## Intention states in susceptibility

We next examined the nature of anhedonic behaviour in susceptible mice by analysing the sequence of reward choices that led them to less frequently choose sucrose rewards, and compared this to the sequences in control and resilient mice. We found that current and previous reward choices were not independent of each other, as the sequence could be described using a Markov model in which the probability of choosing water or sucrose depended on the choice made in the previous trial. The Markov models of control and resilient mice were similar: both switched from water to sucrose and repeated a sucrose choice more often than susceptible mice did (Fig. 3a,b and Extended Data Fig. 3a–d). By contrast, susceptible mice switched more from sucrose to water rewards and made more consecutive water choices.

Given these patterns, we examined whether we could use the four possible sequences of consecutive reward choices (water–water;

water–sucrose; sucrose–water; sucrose–sucrose) as the basis for identifying unique neural signatures of the intention to switch or stay on the same reward choice as the previous trial. To control for the potential confound of reward value differences between trial types, we balanced both previous and current reward types when analysing switch or stay trials (Fig. 3a and Methods).

Single-neuron analysis revealed that neurons that were differentially modulated on the basis of the intention to switch rewards or to stay on the same one as the previous trial were present only in the BLA of susceptible mice (Fig. 3c,d). In addition, a population decoder could successfully distinguish stay trials from switch trials using neural data from the seconds before reward delivery in the BLA of susceptible mice but not in the other groups (Fig. 3e,f). Decoding accuracy for switch versus stay was better than chance for both CSDS groups in vCA1 although accuracy was lower than that in the BLA (Extended Data Fig. 3e).

This led us to reason that specific population activity patterns existed in the BLA of susceptible mice in the seconds preceding the decision to switch or stay. We identified population hidden states in the 4 s pre-reward period using hidden Markov models[38–40] (HMMs; Fig. 3g,h and Extended Data Fig. 3f). Each hidden state is defined by the ensemble activity of simultaneously recorded neurons and reflects distinct population dynamics during pre-reward. The model then assigns each time interval (1-s bins) the most likely hidden state. We validated that a linear decoder, trained on these ensemble activities, could most strongly distinguish between stay versus switch trials in the BLA of susceptible

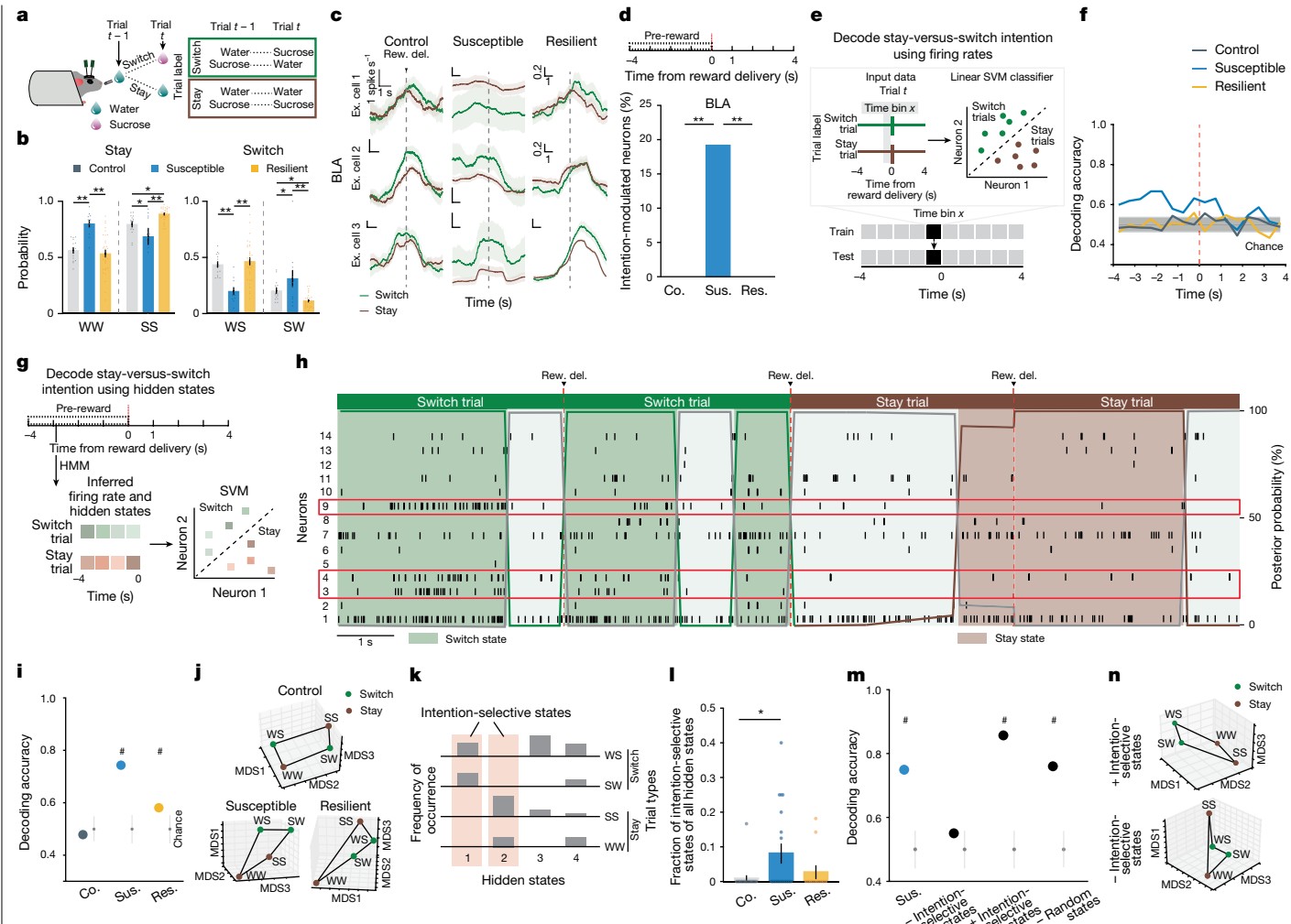

**Fig. 3 | Intention-specific states in BLA as a unique susceptibility signature.**
**a**, Schematic of switch and stay trials. **b**, Susceptible mice (*n* = 12) had more water–water (WW) and sucrose–water (SW) trials than control (*n* = 15) and resilient mice (*n* = 33, one-way ANOVA, group effects, sucrose–sucrose (SS), $F_{2,57}$ = 14.84, *P* < 0.0001; water–water $F_{2,57}$ = 11.71, *P* < 0.0001; water–sucrose (WS) $F_{2,57}$ = 14.84, *P* < 0.0001; sucrose–water, $F_{2,57}$ = 11.71, *P* < 0.0001). **c**, Switch and stay trial-averaged firing rates from example cells. Scale bars, 1 spike s$^{-1}$ and 1 s unless otherwise stated. **d**, Susceptible mice had more intention-selective neurons (*n* = 68 neurons, Fisher's exact test, *P* < 0.0001) in comparison to control (*n* = 132) and resilient (*n* = 69) groups. **e**, Schematic of switch versus stay decoding (raw firing rates). **f**, Susceptible group showed greater-than-chance decoding accuracy (mean of subsamplings, *n* = 10 of 60 neurons, 100 cross-validations). **g**, Schematic of switch versus stay decoding (hidden states). **h**, Example raster and hidden states. Neurons in red rectangles showed preferential firing during switch versus stay trials (Methods). **i**, Switch versus stay could be decoded using inferred firing rates in susceptible mice (*n* = 5, 10 subsampling of 60 neurons, 100 cross-validations). **j**, Switch versus stay representations can be linearly separated in susceptible mice (example multi-dimensional scaling (MDS); Methods). **k**, Intention-selective states are defined as those that occur only in switch or stay trials. **l**, Susceptible group had more intention-selective states (across 4 thresholds (0.1 to 0.4): control *n* = 5, susceptible *n* = 5, resilient *n* = 3 mice, Kruskal–Wallis, *P* = 0.045). **m**, Bidirectional modulation of trials containing intention-selective states bidirectionally changed decoding accuracy in susceptible mice (*n* = 100 cross-validations). **n**, Switch versus stay representations before and after removal of intention-selective states (example MDS). Data are mean ± s.e.m. Chance *n* = 100 shuffles. #Significantly different from chance; *P* < 0.05; **P < 0.01.

mice (and both vCA1 CSDS groups; Extended Data Fig. 3g,h). Accordingly, the population representations of stay versus switch trials were linearly separable in the BLA of susceptible mice (Fig. 3i,j).

Next we identified hidden states that uniquely characterized trials in which mice intended to either stay or switch, which we termed intention-selective states (Fig. 3k–l, Methods and Extended Data Fig. 3l). We found that the BLA of susceptible mice had a significantly higher fraction of these intention-selective states during the 4-s pre-reward period than that of controls (Fig. 3l). Removing trials that contained these states reduced decoding accuracy of stay versus switch trials to chance levels (Fig. 3m) and altered the geometry of population representations of switch versus stay trials (Fig. 3n). Considering only trials with intention-selective states improved the switch versus stay decoding

accuracy, whereas removal of random states did not affect decoding accuracy (Fig. 3m,n). The intention-selective states in susceptible mice were not due to action sequence coding, as they were not present when the two spouts delivered the same reward (Extended Data Fig. 3j–l), or lick rate differences (Extended Data Fig. 3m). Furthermore, we found that vCA1–BLA correlations in susceptible mice were enhanced during intention-selective states, in comparison to non-intention-selective states, raising the possibility that intention-related information may be transmitted between structures (Extended Data Fig. 3n). Finally, using these task-related neural features of susceptibility (intention-specific states) and resilience (reward choice discrimination) allowed us to decode group identity (Extended Data Fig. 3o). Altogether, our results indicate that BLA neurons in susceptible mice evaluate future decisions

with respect to their past choices (by representing switch and stay states), which may contribute to behavioural strategies that result in a reduced number of sucrose rewards.

## vCA1–BLA activity and anhedonia

Having established that the BLA exhibits distinct activity patterns associated with susceptibility to stress, we examined whether we could rescue the neurophysiological responses to stress in susceptible mice and whether rescuing the neural phenotype reversed the maladaptive behaviour of these animals.

The strategy we chose was to manipulate inputs to BLA from vCA1. We targeted this pathway because: vCA1 provides dense input to BLA[9]; our data here show that CSDS produces changes in representations of reward choice and intended task strategies in vCA1 of all stressed mice, which may communicate the information to the BLA to help further shape its reward value coding (Fig. 2e,f and Extended Data Fig. 3e,g); resilience was positively correlated with the strength of communication between vCA1 and BLA for sucrose versus water choices during the pre-reward period (Fig. 4a,b); and vCA1–BLA correlation was enhanced in intention-selective states in susceptible mice (Extended Data Fig. 3n).

To test whether manipulation of vCA1–BLA inputs would modulate signatures of susceptibility in the BLA and/or influence anhedonic behaviour, we increased the excitability of vCA1–BLA projection neurons by expressing the excitatory chemogenetic actuator hM3Dq in these cells[41] (Fig. 4c,d). We then subjected mice to CSDS and recorded BLA and vCA1 activity and behaviour in susceptible mice before and after injection of the hM3Dq activator clozapine-n-oxide (CNO).

CNO increased vCA1 firing rates (Extended Data Fig. 4a,b,k) and modified population activity patterns in vCA1 (Extended Data Fig. 4c–j). During the sucrose preference task, this manipulation enhanced vCA1–BLA correlations for sucrose versus water choices during the pre-reward period (Fig. 4e). In addition, we found that activating the vCA1–BLA pathway increased our ability to decode reward choice post-reward in both BLA and vCA1 (Fig. 4f,g and Extended Data Fig. 4l,m), a signature of enhanced reward choice representation in naturally resilient mice (Fig. 2e,f).

We next examined whether this manipulation of the vCA1–BLA pathway would reduce the occurrence of the unique intention-specific states we had observed in the BLA of susceptible mice. Replicating our previous results, we found that during the saline period, we could decode stay versus switch trials in susceptible mice (Fig. 4h–j and Extended Data Fig. 4n,o). However, activation of vCA1–BLA brought decoding accuracies to chance levels, changed the geometry of representations in the BLA such that switch and stay trials could no longer be linearly separated (Fig. 4j), and decreased the fraction of intention-specific states (Fig. 4k). In other words, activation of the vCA1–BLA pathway reversed this population-level signature of stress susceptibility in the BLA. In addition, a decoder trained to differentiate susceptible versus resilient mice generalizes well to differentiating between saline- versus CNO-treated susceptible mice (Extended Data Fig. 4p–r), further suggesting that activation of the vCA1–BLA pathway reversed the susceptibility phenotype to be more similar to the naturally resilient phenotype.

Finally, we found that vCA1–BLA activation rescued behavioural indices of anhedonia. Administering CNO increased sucrose preference (Fig. 4l), increased the lick rate discrimination index (Extended Data Fig. 4s), enhanced the proportion of sucrose stay trials (Extended Data Fig. 4t–v), and increased social interaction times (Extended Data Fig. 4w,x). No behavioural or neural differences were observed between saline and CNO periods in mice infused with the control mCherry virus (Extended Data Fig. 5).

In summary, these results show that activating the vCA1–BLA pathway rescued both aberrant population dynamics in the BLA of susceptible mice and associated behavioural hallmarks of anhedonia (Fig. 4m).

## Spontaneous activity following CSDS

Finally, we examined whether distinct patterns of population activity could be detected in the BLA of susceptible or resilient mice, in the absence of any overt stimuli or task demands. Clinical studies have revealed altered resting-state functional connectivity between the amygdala and hippocampus in individuals with depression, but the underlying neural mechanisms remain unknown[42].

To mimic a mildly stressful experience in human imaging studies, mice were head-fixed without task-relevant stimuli provided. In line with human studies, we found altered functional connectivity between BLA and vCA1 in CSDS mice, specifically a reduction in the dominant frequency of interaction between the two regions, suggesting a change in communication between the two regions[43,44] (Extended Data Fig. 6a,b). We then examined whether the geometry of spontaneous neural activity patterns differed between groups in each region. As the lack of behavioural time stamps made it difficult to align and directly compare neural representations across animals, we focused on the embedding dimensionality using principal component analysis (PCA), which can estimate population geometry without alignment to overt behaviour[45,46].

This analysis revealed a trend towards higher dimensionality in the BLA population activity of susceptible mice compared to controls (Extended Data Fig. 6c–g), suggesting a larger number of neural population states, with each state spanning a different dimension. Indeed, when we quantified the states using HMM and performed agglomerative clustering of states to identify those that were unique, we found that in the BLA, but not vCA1, susceptible mice showed a greater number of distinct neural states (Fig. 5a–c and Extended Data Fig. 6h–n). Consistent with this, average correlated BLA population activity across time was lower, and thus more variable, in susceptible mice (Extended Data Fig. 6o,p). The greater number of distinct states in susceptible mice could not be attributed to an increased firing rate, which was lower in the BLA of susceptible mice compared to controls (Extended Data Fig. 6q). Furthermore, across all mice, the number of distinct states was significantly correlated with behaviours used to assess susceptibility (Fig. 5d), with greater numbers of clusters strongly predicting social avoidance and anhedonic behaviour. This suggests that structures of population hidden states in the BLA may reflect anhedonia-related behaviour.

We next tested whether we could decode the group identity of individual animals from this resting-state activity by training a classifier using neural features including firing rates (mean and standard deviation), PCA cumulative variance, and the fraction of clustered neural states. Each feature alone allowed us to distinguish between control and susceptible mice to some extent (Extended Data Fig. 6r). However, using all of the feature sets in BLA, but not vCA1, we could significantly decode between all pairs of group identities (Fig. 5e and Extended Data Fig. 6s,t). Notably, cross-validated decoding of susceptible versus control mice was 100% accurate. When we visualized the geometry of the representations in individual mice, we found the greatest distance between control and susceptible mice in the BLA (Fig. 5f). In addition, the neural feature differences we observed were unlikely to be due to differences in movements of the face or the limbs of the head-fixed mice. Specifically, although minor differences in some facial and limb movements were found (Extended Data Fig. 7a–g), decoding accuracy for group identity using face and limb movements as input features was much lower than that of a decoder trained on neural features (Extended Data Fig. 7h–k), and face and limb movements could not explain BLA activity differences (Methods). Applying the same dimensionality and hidden state analysis to neural recordings from the task period could also differentiate between control and susceptible mice (Extended Data Fig. 8).

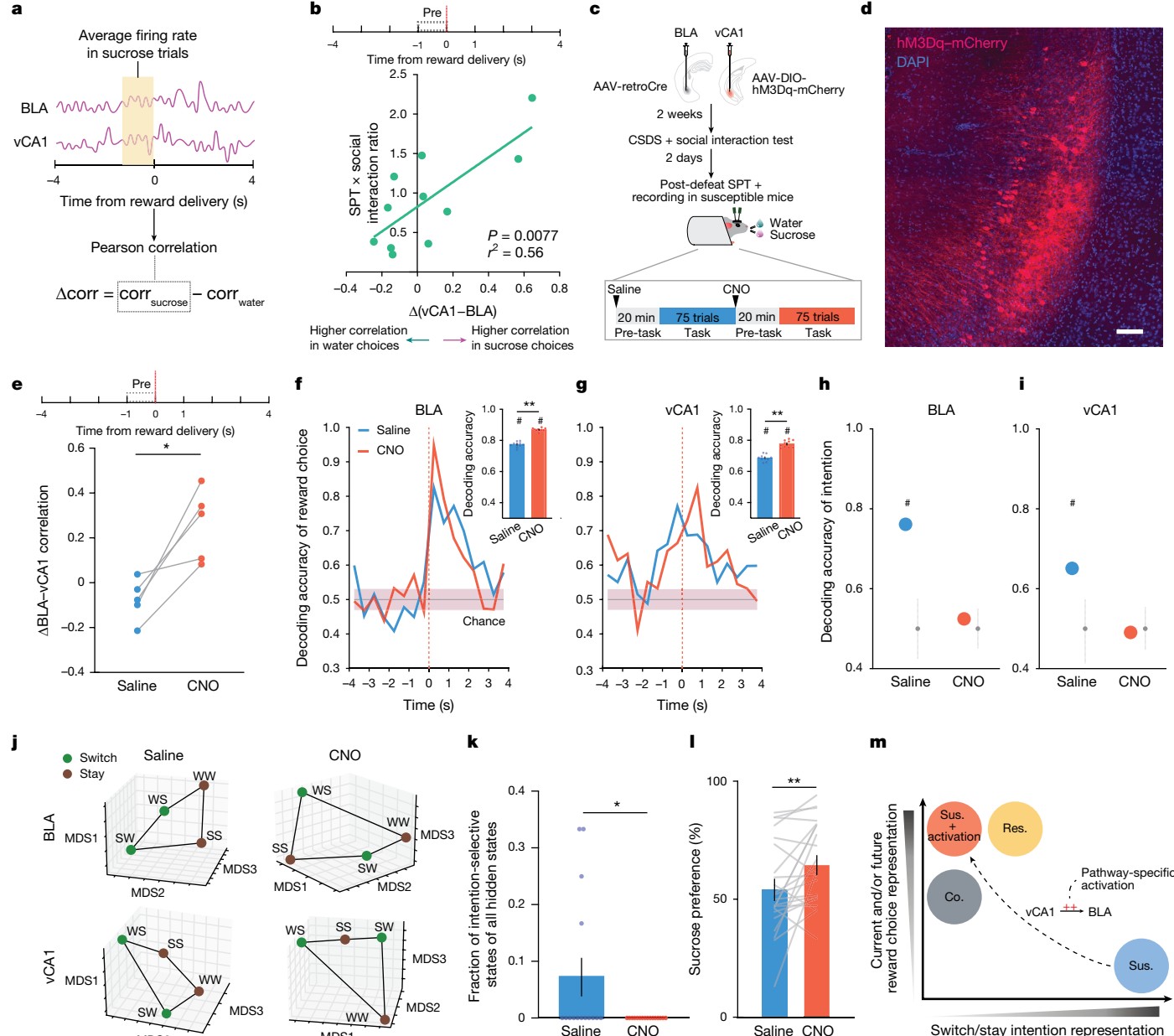

**Fig. 4 | Rescue of dysfunctional vCA1–BLA activity and signatures of anhedonia by circuit-specific manipulations. a**, Schematic for analysing vCA1–BLA interactions. **b**, Δ(vCA1–BLA) correlation pre-reward was significantly correlated with animals' behaviour (n = 11 mice, Fisher's z). **c**, Schematic of chemogenetic activation of BLA-projecting vCA1 neurons. **d**, Representative image of BLA-projecting vCA1 neurons transfected with hM3Dq–mCherry (observed in all 25 mice). Scale bar, 100 μm. **e**, Δ(vCA1–BLA) correlation was enhanced in CNO versus saline (n = 5 simultaneously recorded sessions, paired t-test, $t_4 = 4.22$, P = 0.014). **f,g**, CNO increased decoding accuracy of current reward choice compared to saline (Mann–Whitney, P < 0.0001) in BLA (**f**)

and vCA1 (**g**; Mann–Whitney, P < 0.0001). Insets show averaged decoding accuracy during pre-reward. Coloured lines indicate mean of subsampling (n = 10 subsamplings, 60 neurons, 100 cross-validations). **h,i**, CNO reduced decoding accuracy of switch versus stay trials to chance level in BLA (**h**) and vCA1 (**i**; n = 100 cross-validations, 100 shuffles). **j**, CNO altered the geometry of switch versus stay representations such that they can no longer be linearly distinguished. **k**, CNO reduced the fraction of intention-selective states in BLA (Mann–Whitney, P = 0.043). **l**, CNO increased sucrose preference (n = 23 mice, paired t-test, $t_{22}$ = 2.91, P = 0.0081). **m**, Summary schematic on the main findings. Data are mean ± s.e.m. #Significantly different from chance; *P < 0.05; **P < 0.01.

Finally, we found that a decoder using these neural features during the stimulus-free pre-task period in BLA better predicted whether an animal was exposed to stress than a decoder using only behavioural measures of anhedonia and anxiety-related behaviour (Fig. 5g). This suggests that neural activity features in the BLA in the absence of any stimuli or task demands may be a more powerful biomarker for identifying a history of chronic stress than classic behavioural indices such as social avoidance and anhedonia-related behaviours.

## Discussion

Our study reveals distinct neural signatures of stress resilience and susceptibility in BLA population activity. Using Neuropixels recordings while mice were either at rest or engaged in a free-reward-choice task and leveraging complementary analytic approaches, we identified new population dynamics that underlie distinct features of stress-induced anhedonic state. Critically, when we successfully reversed these neural signatures of anhedonia through targeted modulation of the vCA1–BLA

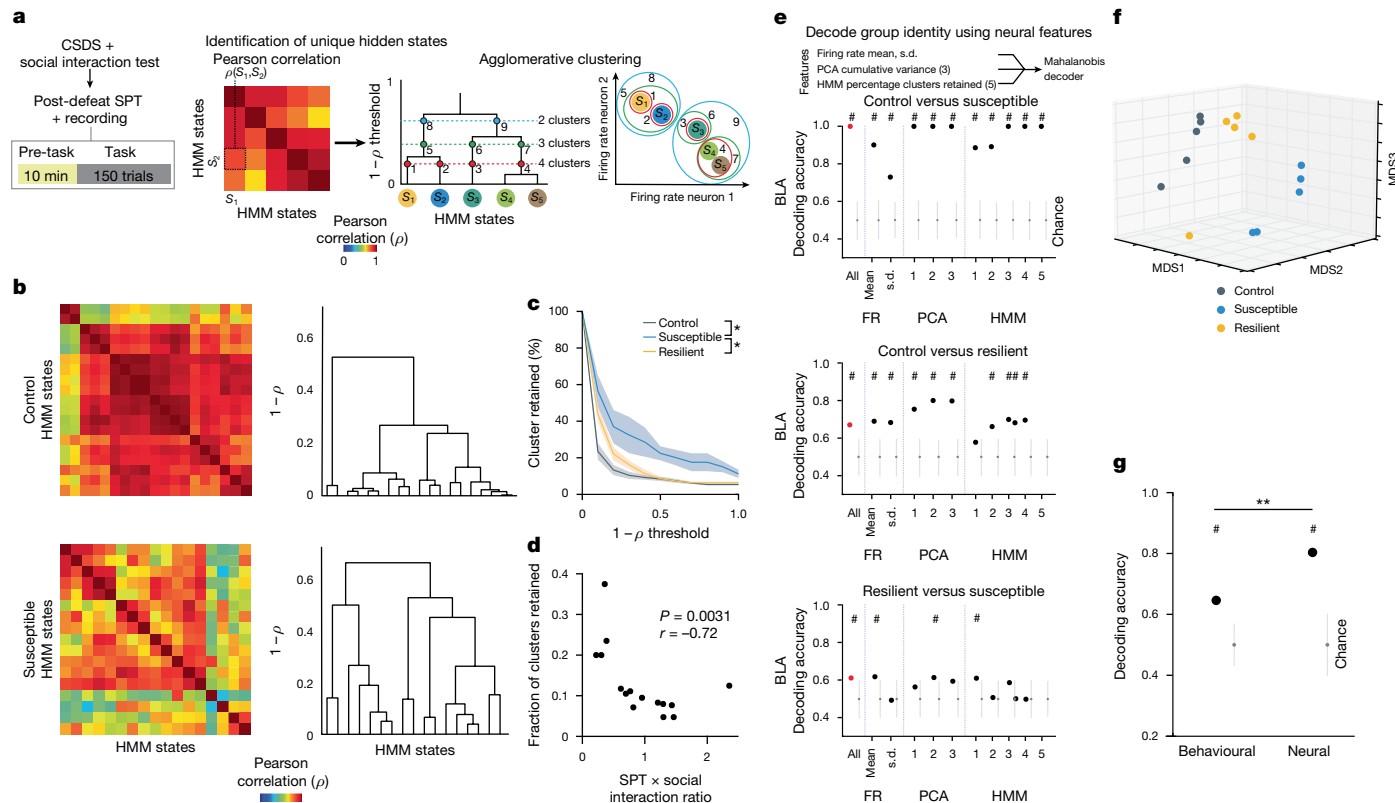

**Fig. 5 | Distinct neural signatures of CSDS mice in the absence of task.**
**a**, Schematic of analysis in pre-task. HMM was used to identify hidden states (S) and states similarity was assessed using agglomerative clustering. **b**, Example state correlation heat maps from BLA of a control ($n = 19$ hidden states) and a susceptible mouse ($n = 17$ hidden states) and respective agglomerative clustering (dendrograms on right). **c**, Susceptible mice had more distant hidden states in BLA (Mann–Whitney, control ($n = 5$ mice) versus susceptible ($n = 5$ mice) $P < 0.05$ for all correlation thresholds except at 0, specifically in ascending order of thresholds $P = (0.01, 0.01, 0.03, 0.03, 0.02, 0.02, 0.04, 0.02, 0.02, 0.04)$; resilient ($n = 5$ mice) versus susceptible $P < 0.05$ for all thresholds except at 0.4–0.6, specifically $P = (1, 0.1, 0.1, 0.1, 0.05, 0.05, 0.05, 0.1)$ for thresholds = $(0, 0.1, 0.2, 0.3, 0.7, 0.8, 0.9, 1)$). **d**, Fraction of clusters (at $1 - \rho$ threshold = 0.5)

was correlated with animals' behaviour ($n = 15$ mice, Spearman's correlation). **e**, Mahalanobis decoder trained on all neural features (All) could decode group identity better than chance in BLA ($n = 1,000$ cross-validations; chance: $n = 100$ shuffles). Feature importance in decoding was examined by systematic removal of each feature (subsequent columns). FR, firing rate. **f**, Multi-dimensional scaling (MDS) of neural features in BLA showed that controls were most distinct from susceptible mice. **g**, Mahalanobis decoder trained on neural features was better at distinguishing control versus CSDS mice than one trained on behavioural features ($n = 1,000$ cross-validations; chance: $n = 100$ shuffles; Mann–Whitney $P < 0.0001$). Data are mean ± s.e.m. Chance distributions are 2 s.d. around theoretical chance level. #Significantly different from chance; *$P < 0.05$; **$P < 0.01$.

circuit, the behavioural consequences of this maladaptive state were also rescued.

By analysing population dynamics during the SPT, we discovered a resilience signature characterized by heightened reward choice representations in the BLA before and during reward consumption. This enhanced reward choice perception or sensitivity may play a crucial role in reinforcing the behavioural processes that lead animals to seek more rewarding options (that is, choosing the sucrose reward)[47]. That is, it may serve as a mechanism for adapting to, or coping with, the experience of CSDS, thereby maintaining a robust behavioural preference for sucrose.

By contrast, the BLA of susceptible mice exhibited reduced representations of current and future reward choices, which may result in decreased reinforcement of behaviours associated with the more rewarding outcome, ultimately contributing to a reduced preference for high-value rewards[48]. Moreover, the BLA of susceptible mice also exhibited unique representations that reflected their intention to switch or stay on the previously chosen reward. This heightened evaluation of future choices with respect to the past is reminiscent of rumination-like states commonly observed in individuals with depression, such as repetitive thinking about past choices and upcoming decisions[49,50].

Furthermore, stress-susceptible mice showed reduced vCA1–BLA correlations during higher value sucrose reward choice trials, potentially driving anhedonic behaviour. When we used chemogenetics to activate BLA-projecting vCA1 neurons in susceptible mice, this led to distinct changes in neural features of resilience and susceptibility. Specifically, whereas the manipulation increased inter-regional communication between these regions and increased representations of current reward choice in both BLA and vCA1, activating vCA1–BLA projection reduced the rumination-like over-representation of the intention to stay or switch in the BLA. Critically, the activation decreased anhedonia-related behaviour.

Finally, by analysing the neural activity patterns in the absence of any task or stimuli, we also found an enhanced exploration of distinct neural states in the BLA of stress-susceptible mice. This may be related to the emergence of intention-selective states that we observed in susceptible mice during the task period, and akin to the intrusive thought patterns observed in patients with depression[51,52]. We speculate that under normal conditions, such as in control mice, the BLA plays a crucial role in evaluating reward values, which subsequently influences the decision to switch or stay. The decision probably occurs downstream of the BLA, because we could not decode the intention to switch or stay in the BLA of control mice. However, in susceptible mice, the BLA's ability

to evaluate reward values may be disrupted by the emergence of these intrusive, intention-selective states, which we could decode clearly in these mice. These intrusive states may interfere with downstream activity, biasing the decision to switch or stay towards the lower value reward. Notably, these intrusive states are not merely noise, as we could decode the signal as the intention to switch or stay. The ultimate effects on decision-making are probably probabilistic, with the downstream region reading out all states (both normal and intrusive) from the BLA. In susceptible mice, these intrusive states may sometimes increase the probability of staying, whereas in other instances, they may increase the probability of switching. Consequently, susceptible mice exhibit an aberrant reward decision-making process, resulting in anhedonia. A similar process might govern the pre-task period in the absence of reward stimuli, where the higher dimensionality reflects additional intrusive states in susceptible mice.

Notably, we found that features of neural activity in the BLA during task-free periods were more effective than classic behavioural readouts or spontaneous facial and limb movements in distinguishing between control mice and those with a history of CSDS. This suggests the possibility that resting-state neural activity patterns in the BLA may hold substantial potential as a new biomarker for identifying individuals who have experienced stressful life events. Although we did not observe any significant contribution of spontaneous facial or limb movements to BLA activity, it may be possible that other spontaneous behavioural features not captured here may contribute to BLA activity, and it may also be possible that the small differences in some behavioural features we observed could be encoded elsewhere in the brain[53,54].

Our data suggest that while both reward choice and intention information are present in vCA1 of stressed mice, the differences between susceptible and resilient groups become more pronounced in the BLA, suggesting that vCA1 probably relays stress information to the BLA to further shape distinct resilient and susceptible outcomes. In the BLA, estimation of reward values versus intention to switch or stay may represent two distinct modes during reward decision-making: the former is dominant in control and resilient mice, and the latter is dominant in susceptible mice. We reason that these intention-selective states are intrusive and disrupt normal decision-making in susceptible mice to promote anhedonic responding, as they are not present in control and resilient mice. Chemogenetic stimulation of BLA-projecting vCA1 neurons in susceptible mice disrupted the encoding of the intention to switch versus stay in BLA and vCA1, allowing for better reward value coding and reward-related information transfer between BLA and vCA1. In addition, it may also be possible that when reward values are more distinctly represented, mice may rely less on intention and more on reward value for decision-making.

While dysfunction in dopaminergic systems has been implicated in motivational changes in depression and chronic stress[2,55-59], this work provides crucial evidence for a role of the vCA1–BLA circuit in modulating stress-induced behavioural phenotypes. By demonstrating that boosting vCA1-to-BLA communication can normalize neural dynamics associated with susceptibility and promote those associated with resilience in the BLA, our findings shed light on how dysfunction in this circuit may contribute to stress-induced maladaptive states. Moreover, these results highlight the vCA1–BLA circuit as a promising target for neuromodulation in mood disorder treatments and open new avenues for potential therapies to more effectively address stress-induced pathologies.

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

## Methods

### Mice

All procedures were conducted in accordance with the National Institutes of Health's Guide for the Care and Use of Laboratory Animals and the institutional guidelines of the University of California, San Francisco's Institutional Animal Care and Use Committee. Adult (8–12 weeks old) male and female C57BL/6J mice were supplied by The Jackson Laboratory. Adult (5–6 months old) CD1 retired male breeder mice were supplied by Charles River. All mice were kept on a 12-h light/dark cycle, and all experiments were conducted during the light phase. We performed recordings in 60 mice for the original dataset, including 45 CSDS mice (30 males, 15 females) and 15 control mice (10 males, 5 females). The results shown are combined data using both males and females, as we did not observe significant differences between males and females. Mice were randomly assigned to control or CSDS groups before CSDS exposure. A separate cohort of 41 CSDS mice underwent chemogenetic manipulation experiments. Twenty-three of the mice received AAV-DIO-hM3Dq viral micro-infusion and 18 mice received AAV-DIO-mCherry infusion. Mice were randomly assigned to hM3Dq or mCherry groups at the time of surgery. From the hM3Dq and mCherry groups, we performed recordings in seven of the susceptible mice in each group. Experimenters were blind to the condition and group assignments of mice.

### Surgery

**Head bar and craniotomy surgery.** One week before lick training, head bar surgeries were conducted on all mice (8–9 weeks old). According to a previously described protocol[31], mice were anaesthetized with 1.5% isoflurane with an $O_2$ flow rate of 1 l min⁻¹, and head-fixed in a stereotaxic frame. A custom-made titanium head bar was then attached to the skull using Metabond adhesive cement (Parkell). Possible recording sites (see the section entitled Neuropixels recording and data preprocessing) were stereotaxically marked using a permanent marker on the skull surface, and the skull was covered using silicon (Smooth-On). At 3 days before Neuropixels recording, craniotomy surgery was performed, in which, under anaesthesia, craniotomies were made at the previously marked coordinates. The skull surface was covered with Kwik-Sil (World Precision Instruments).

**Viral micro-infusion surgery.** For mice that underwent chemogenetic manipulations, adult mice (8–9 weeks old) received viral micro-infusion in the same surgery as head bar attachment, as in a previously described protocol[31]. Specifically, AAV8-hSyn-DIO-hM3D(Gq)-mCherry (Addgene, 44361-AAV8, $2.9 \times 10^{13}$ viral genomes (vg) per millilitre) or AAV8-hSyn-DIO-mCherry (Addgene, 50459-AAV8, $1.0 \times 10^{13}$ vg ml⁻¹) was micro-infused into vCA1 bilaterally (500 nl per hemisphere, −3.52 mm anterior–posterior (AP), ±3.1 medial–lateral (ML), −4.2 (150 nl), −4.1 (200 nl) and −4.0 (150 nl) dorsal–ventral (DV), from bregma according to ref. 60), and AAV2retro-CAG-Cre (UNC Vector Core, Ed Boyden's stock, $4.1 \times 10^{12}$ vg ml⁻¹) was micro-infused into the BLA bilaterally (500 nl per hemisphere, −1.80 mm AP, ±3.1 ML, −5.0 (150 nl), −4.8 (200 nl) and −4.6 (150 nl) DV). Viral vectors were delivered using Nanoject 3 (Drummond Scientific). The needle was held in place for >5 min after infusion at each DV site, and for 10 min after the last DV site. Following viral micro-infusion, a head bar was attached to the skull as described above.

### Behaviour

**CSDS.** The CSDS procedure was conducted according to a previously established protocol[4]. Briefly, CD1 male mice were singly housed following arrival for >1 week and were then pre-screened for aggression over 3 consecutive days. Each day, a CD1 mouse was placed in a cage with a new screener BL/6 mouse for 3 min. An aggressive CD1 mouse is defined as one that attacked the BL/6 mouse within the first minute over a minimum of 2 consecutive days. Only aggressive CD1 mice were used in defeats and social interaction tests. Defeats occurred over 10 days, for which, each day, a BL/6 mouse was introduced to a new CD1 mouse's cage for 10 min. Defeats were terminated early if severe injuries on BL/6 mice were observed. After 10 min, a clear plastic divider with perforations was placed in the middle of the defeat cage for 24 h, to physically separate the BL/6 and CD1 mice while allowing visual and odour cues to transmit and reinforce the defeat experience during co-housing. After the tenth day of defeat, BL/6 mice were singly housed in new cages (without CD1 mice) for 24 h before the social interaction test. For female defeats, female BL/6 mice were first coated with urine from other aggressive CD1 male mice (not used in defeats) before being introduced to the defeat CD1 mouse cage[61], to minimize mounting behaviour and maximize defeats. Female defeats were terminated early if mounting was observed. For the control group, a BL/6 mouse was co-housed across from another conspecific across a divider for 10 days without any physical interaction or defeats. On each day, a new BL/6 mouse pairing was introduced.

**Social interaction test.** The social interaction test took place 1 day after termination of CSDS (or the control procedure). BL/6 mice were habituated to the social interaction test room for 1 h before the test. The test was performed under red light (10 lx) in a test arena (custom made, 42 cm (w) × 42 cm (d) × 42 cm (h)) in a sound attenuation chamber. During the first phase of the test, the BL/6 mouse was introduced to the test arena with an empty enclosure (10 cm (w) × 6.5 cm (d) × 42 cm (h)) at one end for 2.5 min, and its activity patterns were tracked using Ethovision (Noldus Information Technology). At the end of 2.5 min, the mouse was placed back in its home cage, and the empty enclosure was replaced with a second enclosure containing a new aggressive CD1 that had not been used in defeats. The BL/6 mouse was put back in the test arena for another 2.5 min. The social interaction score, as a measure for social avoidance, was calculated as the time spent in the interaction zone (14 cm × 24 cm) with the aggressor present versus absent. The lower the social interaction ratio, the more socially avoidant the animal was. The same test protocol was used for all experiments, except for when chemogenetic manipulations were performed during the social interaction test.

For chemogenetic manipulation during the social interaction test, the same surgery and social defeat procedures were used as before, and then we performed 2 days of social interaction tests. On day 1, mice were injected with saline (intraperitoneally (i.p.)) 20 min before the social interaction test. On day 2, mice were injected with CNO (i.p.) 20 min before the social interaction test. We performed the social interaction tests on two separate days to prevent habituation to the social interaction test chamber.

**Elevated plus maze.** The elevated plus maze assay was performed an hour after the end of the social interaction test using an established protocol[11]. Briefly, mice were placed in a standard maze (height from the floor, 13.5 in; length of each arm type, 25 in; arm width, 2 in; closed arm height, 7 inches; height and width of ledges on the open arms, 0.5 in; light over the open arms, 650 lx). Mice were positioned in the central region of the maze and allowed to explore for 15 min. Their behaviour was tracked and analysed using Ethovision (Noldus Information Technology). Open arm time, as a measure for anxiety-related behaviour, was calculated as the percentage of time spent in the open arms of the maze.

**Head-fixed SPT.** Following recovery from head bar surgery, mice were habituated to the experimenter and the head-fixed set-up for 15 min a day for a week. After habituation, mice were water-restricted to about 85–90% their ad lib body weight and were trained for 3 days to lick on the custom-designed dual-spout head-fixed reward delivery apparatus. On day 1, mice were introduced to 1 lick spout, from which sucrose rewards (10% sucrose, about 3.5 ml each) were intermittently

delivered following licking (that is, rewards were lick contingent) with 8 s ITI, with a maximum of 150 rewards per session. Sucrose rewards were delivered using a solenoid-gated gravity feed. Licks were detected using a piezo element (SparkFun). Stimulus delivery and sensor reading were controlled using a custom Arduino MEGA board and recorded using CoolTerm software. On days 2 and 3, mice were introduced to 2 lick spouts, one on each side of the mouse, separated by about 50°. Sucrose rewards were delivered in both spouts following licking with 8 s ITI. The goal was to teach mice that rewards were delivered from both spouts. Thus, if a mouse showed preference for the spout on one side, that spout was temporarily removed so the mouse could learn to lick from the other spout. Once the animal showed similar preference for both spouts, lick training was completed and pre-defeat SPT was initiated on the following day. SPT occurred over the course of 2 consecutive days, during which one spout delivered water and the other delivered sucrose. Rewards were delivered following licking with 8 s ITI and a maximum of 150 rewards in total per day. The spout designation was randomized across mice on day 1 and counterbalanced on day 2. Sucrose preference was calculated as the averaged percentage of sucrose rewards obtained across 2 days. On completion of day 2 of pre-defeat SPT, mice were taken off water restriction and housed in a social defeat room for 3 days before CSDS began. Post-defeat SPT was performed using the same protocol, with the addition of Neuropixels recording. Post-defeat SPT was used for all analysis shown.

To control for the possibility that reward choice and intention signals were driven by differences in direction or action sequence (left versus right) coding, a separate cohort of mice were recorded using the same-reward SPT, in which, instead of delivering water or sucrose in the two lick spouts (different-reward SPT), both spouts delivered sucrose rewards. The rest of the experiments were the same as described above.

For chemogenetic manipulation during the SPT, 3 weeks after viral micro-infusion (see the section entitled Viral micro-infusion surgery), CSDS and control mice went through the same CSDS or control procedure and social interaction test. On SPT days, saline (i.p.) was injected 20 min before the first half of the SPT (maximum 75 trials). Then, CNO (i.p.) was injected 20 min before the second half of the SPT (maximum 75 trials). The design allowed for within-animal within-session comparisons of behaviour and neural activity patterns before and after CNO injection.

## Neuropixels recording and data preprocessing
**Recording.** Mice were head-fixed to the SPT apparatus without lick spouts present. Kwik-Sil was removed from the skull surface. Before insertion, Neuropixels 1.0 probes (IMEC) were first coated with DiI, DiO or DiD dyes (ThermoFisher Scientific) and allowed to dry. Probes were inserted at about 1 mm min$^{-1}$ to the target coordinate using Sensapex manipulators. Probe targets and their coordinates are as follows: amygdala (−1.71 mm AP, −0.28 mm ML, −6.5 mm DV, at 31.3° ML) and ventral hippocampus (−3.9 mm AP, −2 mm ML, −4.5 mm DV, at 25.8° ML). One or two probes were inserted per session per mouse. Simultaneously recorded probes were coated in the same colour of dye but spaced at least several hundred micrometres apart to allow for unambiguous identification. Different colours of dyes were used across days to help differentiate probe tracks. After a probe reached the targeted DV site, it was left in place for 10 min before the start of recording, which includes 10 min of pre-task (no task stimulus) and SPT. Neuropixels action potential signals were recorded using Neuropixels acquisition system and SpikeGLX software (https://billkarsh.github.io/SpikeGLX/), at 30,000 Hz with gain of 500. Behavioural signals were recorded using a separate data acquisition board (National Instruments), along with a synchronization signal that was also recorded by Neuropixels to help synchronize clocks between different data streams. After each session of SPT, probes were slowly removed from the brain and the skull was covered with Kwik-Sil. Probes were cleaned using Tergazyme solution

(1%, Alconox) overnight and rinsed using deionized water before reusing or storage.

**Histology and probe track registration.** At the end of the experiments, mice were transcardially perfused with 1× PBS followed by 4% paraformaldehyde solution. Brains were fixed overnight at 4 °C, and then transferred to 30% sucrose solution for 48 h. Brains were sectioned coronally using a microtome (Leica SM2000) at 50 μm thickness and mounted on glass slides with Fluoromount G with DAPI (Southern Biotech). Images were obtained using a confocal microscope (Nikon Ti2-E Crest LFOV Spinning Disk/C2 Confocal) with a 20× objective. Probe tracks were traced using the AllenCCF toolbox (https://github.com/cortex-lab/allenCCF).

**Spike-sorting.** Neuropixels action potential signals were preprocessed and spike-sorted offline using Kilosort 2 (ref. 62) or Kilosort 4 (ref. 63), and after sorting, the clusters were manually validated using Phy[64]. Only well-isolated clusters (putative single units that are classified as 'Good' using Phy) were analysed. All other clusters, including multi-unit activity and noise, were not analysed.

## Data analysis
Animals were allowed to freely choose reward types after 8 s ITI had passed between trials, by licking at the spout of their choice. Reward deliveries were lick contingent. Trial types were defined as a ±4 s time window around the time of reward delivery. For all analyses, only sessions with at least five neurons in the region of interest were used. For analysis during the pre-task period, we used min 2–8 of the 10 min pre-task recording period. For analysis during the task period, we used time windows specified in each figure. All data analysis were performed using custom codes in MATLAB and Python.

## Behavioural data analysis
**Behavioural classification of mice.** The relationship between sucrose preference and social interaction ratio was assessed using a Pearson correlation. To classify CSDS mice into subtypes, we applied unsupervised $K$-means clustering using both behavioural metrics, sucrose preference and social interaction ratio. The optimal number of clusters was determined by evaluating cluster numbers from 2 to 10 and maximizing the silhouette score.

**Lick analysis.** Lick rasters were generated by binning licks using 0.02-s bin size. Lick rates were calculated using 0.1-s bin size and averaged across trials per mouse for each trial type as specified in the figures. As mice tend to sample from both lick spouts in a trial (with them ultimately choosing and obtaining a reward from one), we computed the lick rate DI to assess their preference for licking at each spout. We first quantified the difference between lick rates on sucrose versus water lick spouts for sucrose choice trials (lick rate sucrose spout − lick rate water spout), and separately, the difference between lick rates on sucrose versus water lick spouts for water choice trials. The two values were then averaged to obtain the DI for that session. A DI value greater than 0 suggests a greater lick rate on the sucrose spout in comparison to the water spout, and vice versa for a DI value less than 0.

To take into account reward history and assess how it affects current behaviour, we further divided sucrose and water trials into sucrose–sucrose (SS), water–sucrose (WS), water–water (WW) and sucrose–water (SW) trials (previous–current reward). The first trial of each session was discarded as it had no prior trial. To assess the probability of each trial type irrespective of the animal's overall sucrose preference, we normalized the number of trials to the total number of previous trials of a specific type. For example, we defined the overall transition probability from a water trial to a sucrose trial as $P(WS) = P(WS)/(P(WW) + P(WS))$, and from a sucrose trial to a sucrose trial as $P(SS) = P(SS)/(P(SW) + P(SS))$, in which $P(XY)$ is the transition

probability from reward $X$ to reward $Y$. We normalized the transition probabilities such that $P(WW) + P(WS) = 1$, and $P(SS) + P(SW) = 1$. Using this normalization, if $P(SW)$ is not significantly different from $P(WW)$, this would suggest that the current water reward choice is independent of the previous reward, because of the probability of switching from sucrose or staying on water is the same; otherwise, the current reward choice is dependent on the previous reward (that is, reward choices could be modelled as a first-order Markovian process).

We also computed the proportion of each of the four trial types normalized to the total number of trials per session, to assess how much each trial type contributes to the overall session. In this case, the percentages of trials of each of the four trial types were computed per session and averaged across sessions for each mouse. As the number of trials may be influenced by each animal's innate preference for different rewards, we computed the chance probability of the occurrence of each trial type by calculating the joint probability of the previous and current trial. For example, chance $P(SW) = P(S) \times P(W)$. The number of trials was then subtracted by the chance level in each mouse ('number of trials chance removed'). Sucrose–sucrose and water–water trials were combined when analysing stay trials, and sucrose–water and water–sucrose trials were combined when analysing switch trials. The preference between stay versus switch trials in each mouse was calculated as: percentage of stay trials − percentage of switch trials. To quantify the number of consecutive trials, we first obtained the average number of consecutive trials per trial type (sucrose or water) per session and then averaged across sessions for each mouse.

**Decoding group identity using behavioural features.** To examine whether group identity could be decoded using behavioural data, we defined a Mahalanobis-like binary decoder. Specifically, for each mouse, we considered four behavioural features: lick rate DI during pre-reward and post-reward, elevated plus maze open arm time (CSDS mice showed increased anxiety-like behaviour[5]; Extended Data Fig. 1c), sucrose preference, and social interaction ratio. Considering two groups at a time, we defined and constructed a Mahalanobis binary decoder to assign a single testing mouse to one of the two groups in the behavioural feature space. The input to the binary classifier consisted of an $N \times F$ training matrix and a $1 \times F$ testing matrix, in which $N$ represents the total number of training mice between the two classes, and $F = 4$ represents the total number of features. In each cross-validation, we first balanced the number of mice in each group by randomly subsampling the minimum number of mice between the groups. Next, we randomly selected one mouse as the testing sample and used the remaining mice as the training set, for a total of 1,000 cross-validations. We defined a Mahalanobis-like distance in the feature space as the Euclidean distance between the testing mouse and the centroid of the training groups, divided by the variance along the distance direction. The testing sample was assigned to the group identity with the minimum Mahalanobis-like distance. The performance of the decoder was evaluated by calculating the fraction of correct classifications out of the total 1,000 cross-validations, and the entire procedure was repeated for all possible pairs of the three groups (that is, control, susceptible and resilient mice).

**Pre-task spontaneous facial and limb feature analysis.** For a subset of mice, we recorded spontaneous facial and limb movements during the pre-task period using the Alvium 1800 U-158 camera (Allied Vision) with the 16 mm C VIS-NIR Fixed Focal Length Lens (Edmund Optics), at frame rate of 114 frames per second, using the MATLAB Image Acquisition Toolbox. We tracked 12 keypoints using DeepLabCut[65]. These include eye top, eye bottom, eye front, eye back, snout top, snout tip, snout bottom, whisker 1, whisker 2, mouth, left hand and right hand.

To quantify facial and limb movements, we calculated the following features from keypoints[66]: eye opening ratio, snout angle, mouth position, whisker position, left limb $X$ and $Y$ coordinates. Eye opening ratio

is defined as the ratio between the vertical and horizontal Euclidean distance of the eye (that is, (eye top − eye bottom)/(eye front − eye back)). An eye opening ratio of 1 represents a perfectly spherical opened eye. Snout angle is calculated as the angle formed by the vector of snout tip to snout top, and the vector of snout tip to snout bottom. A smaller angle represents a more pointed snout. The mouth position is calculated as the Euclidean distance between the mouth and the eye front. The whisker position is calculated as the Euclidean distance between whisker 1 and the eye front.

**Analysis of embedding dimensionality of face and limb features.** We used PCA to assess the embedding dimensionality of facial and limb features over time for each mouse. We examined the facial and limb features in a 250-ms bin during the 6-min window (min 2–8) within the 10 min pre-task recording period. We define the feature space as a six-dimensional space in which each axis is the value of one facial and limb feature. The PCA analysis allowed us to identify how much variance of these features in the feature space is accounted for by each principal component (PC). We applied PCA to the $K \times T$ matrix, for which $K = 6$ is the number of facial and limb features, and $T$ is the number of bins, and we determined the cumulative curve of the variance explained by each PC. We subsequently used the cumulative variance values for the first three PCs as features to decode the group identity.

**HMM for face and limb features.** We fitted HMMs to facial and limb features recorded in a 250-ms bin during 6 min (min 2–8) of pre-task recording. The HMM identifies patterns of behaviour along time, with each pattern corresponding to a specific behavioural state, defined by the combination of the six facial and limb features, that is not directly measurable. We fitted an HMM separately for each mouse using the same software framework developed by the Linderman Lab (https://github.com/lindermanlab/ssm) we used to analyse neural data. The input data for the HMM consisted of a $K \times T$ matrix, for which $K = 6$ represents the total number of facial and limb features in the session, and $T$ represents the total number of time bins, and we assumed a Gaussian model as the observation model. For each time series, we fitted 5 models with a maximum of 100 iterations for each value of the total number of states ranging from 2 up to 100, using randomized initial conditions. The model with the smallest Akaike information criterion score was retained as the best model for further analyses.

**Agglomerative clustering analysis for HMM behavioural states.** To better characterize the spatial structure of the HMM states in the facial and limb features space, we examined the pairwise correlation between the states. For state 1 defined by $X = (x_1, x_2, \ldots, x_k)$, in which $x_i$ is the value of the feature $i$, and state 2 defined by $Y = (y_1, y_2, \ldots, y_k)$, we computed the Pearson correlation coefficient $\rho(X,Y)$ to assess the distance between the states in the facial and limb feature space. We calculated the correlation coefficients for all pairs of total $N$ states and stored them in an $N \times N$ correlation matrix $J$. Subsequently, we performed agglomerative clustering on the correlation matrix. Specifically, we defined a new distance matrix $D$ as $1 - J$, in which 1 is an $N \times N$ matrix of ones. This matrix served as the input to the agglomerative clustering algorithm, which iteratively combines states to define new clusters according to the pairwise distance. The algorithm initialized each state as a separate cluster with minimum distance (maximum correlation) and iteratively merged two clusters $v$ and $u$ with the smallest distance into a new cluster. The new distance $d$ assigned to the agglomerated clusters was defined as $d(u,v) = \max(\text{dist}(u[p], v[q]))$, in which $p$ and $q$ represent all of the points in the merged clusters $u$ and $v$, also known as the farthest point algorithm (sklearn.cluster.AgglomerativeClustering, built-in class in scikit-learn in Python[67]). Agglomerative clustering has the advantage of producing a hierarchical structure of clusters, and this hierarchical representation allowed us to examine the relationships and similarities between states, specifically how behavioural states may be

nested differently within large clusters in different groups (for clustering analysis on neural data, see also the section entitled Agglomerative clustering analysis). Agglomerative clustering does not require any assumption regarding the total number of clusters. It iteratively merges the closest states and clusters until all states are merged into one final cluster. We performed the clustering analysis separately for each mouse. After examining the clusters, we counted the total number of clusters at different levels of distance, or thresholds, for which the higher the levels of distance, the lower the number of clusters, until reaching only one cluster at the highest distance. We assessed the number of total clusters and the proportion of total clusters retained relative to the total number of states as a function of thresholds. A higher number of states at the same threshold value indicates a greater degree of dissimilarity among the inferred hidden states. We retained the proportion of total clusters along these curves from a threshold of 0.01 up to 0.05, because of the high facial and limb feature correlation values between inferred states, resulting in a total of five features that were subsequently used in the decoding of group identity.

**Decoding group identity using facial and limb features.** This analysis aimed to decode the group identity (that is, control, susceptible or resilient) on a single-mouse basis by analysing the facial and limb features recorded during 6 min of the pre-task period. For each mouse, we assessed the embedding dimensionality using PCA (see the section entitled Analysis of embedding dimensionality of face and limb features), and we considered the cumulative variance explained by the first three PCs as features for decoding. Following the inference of hidden states and the clustering analysis, we calculated the proportion of clusters retained at different thresholds and extracted the values at five distinct thresholds (see the section entitled Agglomerative clustering analysis for HMM behavioural states). Additionally, we computed the mean and standard deviation of the facial and limb features as the last two features. Overall, we assessed a total of $F = 10$ features for each mouse. We used the Mahalanobis binary decoder procedure, in which the input to the binary classifier consisted of an $N \times F$ training matrix and a $1 \times F$ testing matrix, in which $N$ represents the total number of training mice between the two classes, and $F = 10$ represents the total number of features (see the section entitled Decoding group identity using behavioural features). The decoder was trained and tested for 1,000 iterations, with a new random testing subject selected and removed from the training set for each of them.

**Neural feature decoding when facial and limb feature decoding is at chance.** We compared neural to facial and limb feature decoding accuracy during only the time bins when facial and limb decoding accuracy is at chance to assess how well a decoder using neural features (see the section entitled Decoding group identity using neural features) performs even when facial and limb feature decoding is at chance level. Specifically, we trained an SVM with a linear kernel on the six facial and limb features to differentiate between control versus susceptible mice in a subset of 10 randomly selected 1-s bins of training mice and tested on the pre-task time window of 6 min (min 2–8, 360 time bins total) of one held-out testing mouse. We then selected the test time bins when facial and limb decoding accuracy is within 1 or 2 s.d. of the chance level (0.5), obtained from the distribution accuracies of 100 null models after shuffling the labels. In these same time bins for which the classification based on facial and limb features is at chance, we performed decoding of control versus susceptible mice using neural activity of BLA.

**Contribution of facial and limb movements to neural activity in the BLA.** We investigated whether the facial and limb features contributed to BLA neural activity during the pre-task period. We fitted facial and limb features to neural activity (firing rate) using linear regression in each mouse separately. We binned neural and facial and limb feature data using a 1 s time window (total of $T = 360$ bins) and defined our model as $Y = AX^T + \beta$, in which $Y$ is an $N \times T$ matrix with the firing rate of $N$ recorded neurons, $A$ is an $N \times K$ matrix with the regression coefficients of $K = 6$ facial and limb features, and $X$ is a $T \times K$ matrix with the $K$ facial and limb features values. $\beta$ is the intercept (a constant). Before fitting, the data were centred to zero. We used the linear least square error as a loss function and added an L2-norm regularization term to prevent overfitting. We tried a range of values for the L2-norm regularization term, ranging from 0 (equivalent to ordinary least squares) to $10^3$, with no significant difference in the final coefficient of determination ($R^2$) estimate.

We did not find a positive $R^2$ from any of the linear models, suggesting that using facial and limb features that we recorded, we could not predict BLA neural activity better than chance. In other words, these facial and limb features are unlikely to contribute significantly to BLA neural activity, and consequently, any group differences that we observed.

### Single-neuron analysis
**Firing rate.** For task period, spike trains were aligned at the time of reward delivery (time 0) and neurons within the same region were pooled across animals of the same group to construct pseudo-populations. Only neurons with at least ten trials per trial type (sucrose and water) were included. For peristimulus time histograms, spikes were binned at 10-ms resolution, z-scored to pre-reward (−1 to 0 s), and smoothed with a 50-ms moving average filter. For analysis of raw firing rates, spikes were binned at 500-ms resolution.

**Reward-choice-selective neurons.** Analysis was performed using pseudo-population and only neurons with at least ten trials per trial type (sucrose and water) were included. Mice with fewer than five neurons in regions of interest were excluded. Reward-choice-selective cells were identified[68,69], and the magnitude of the selectivity was quantified, using the auROC method, which compares single-neuron firing rates between trial types, across levels of response thresholds for each time bin. Spikes were binned at 500-ms resolution. Shuffled distributions were computed for each time bin by randomly shuffling trial type ten times per neuron. A neuron is deemed reward choice selective if its auROC is >2 s.d. of the shuffled distribution for that neuron. The fraction of selective neurons in a region was calculated as: number of selective neurons/total number of neurons. Differences in the fraction of selective neurons across groups were assessed using Fisher's exact tests.

**Intention-modulated neurons.** Analysis was performed using pseudo-population and only neurons with at least ten trials per trial type (switch and stay) were included. Intention-modulated neurons were identified using a similar method as reward-modulated neurons. Mice with fewer than five neurons in regions of interest were excluded. In this case, a cell is deemed intention-modulated if the distribution of firing rates during the 4 s pre-reward period (−4 to 0 s) in switch trials is significantly different from stay trials, as identified using Wilcoxon rank-sum test followed by false discovery rate correction across all neurons in that group ($P < 0.05$). As the fraction of neurons was small and did not meet the criteria for using Chi-squared test, Fisher's exact tests were used to perform statistical comparisons between percentages of intention-modulated neurons across groups.

### Population analysis
**Analysis of embedding dimensionality.** PCA was used to evaluate the embedding dimensionality of population activity of simultaneously recorded neurons over time. The method aims to identify how much variance of the population representation in the firing rate space is accounted for by each PC. We chose this method because the pre-task period lacks behavioural labels. PCA has the advantage of allowing us to compare neural data between animals because the method is

invariant for rotations and global stretching, transformations normally needed to align a neural representation of one subject into another. We examined the activity of each neuron in 1-s bins during the 6 min time window (min 2–8) within the 10 min pre-task recording period, resulting in 360 bins. The ensemble activity across these bins can be represented as a geometrical object in the firing space, with each axis representing the firing rate of a neuron and each point representing the ensemble's activity in a time bin. We calculated the embedding dimensionality of this geometrical object for each mouse. We included only mice with at least five simultaneously recorded neurons in the region of interest during the pre-task recording. We randomly selected five neurons for each mouse and calculated the $z$-scored firing rate matrix $N \times T$, in which $N$ is the number of neurons, and $T$ is the number of time bins. We applied PCA to this matrix and determined the cumulative curve of the variance explained by each PC. We repeated this procedure 1,000 times and averaged the results across the subsamples for each mouse. Our goal was to compare cumulative variance curves across groups and determine whether a group had a higher cumulative value at $M$ PCs ($M \leq 5$), indicating a lower dimensionality of the geometrical object. We subsequently used the cumulative variance values for the first three PCs as features to decode the group identity.

We also assessed the participation ratio (PR), which is a normalized measure of dimensionality based on the full distribution of PCA eigenvalues (that is, how much variance is explained by each PC), and it is defined as:

$$PR = \frac{(\sum_{i=1}^{N} \lambda i)^2}{\sum_{i=1}^{N} (\lambda i^2)}$$

in which $\lambda_i$ are the eigenvalues of the covariance matrix of the neural activity, and $N = 5$. If only one eigenvalue explains all of the variance ($\lambda_i \neq 0$ for $i = 1$ and $\lambda_i = 0$ for all $i \geq 2$), then PR = 1. On the other hand, if all eigenvalues are equal, the dimensionality is maximum, PR = $N$ (refs. 70,71).

During the task period, the same analysis was repeated during the 1 s of pre-reward and post-reward periods, using a $z$-scored firing rate with 0.2-s bins (5 bins for each period).

**HMM.** We used HMMs to identify patterns of population activity in the time series, with each pattern corresponding to a specific neural state that is not directly measurable[38,40,72]. We fitted an HMM separately for each mouse for the pre-task and task period. For the DREADD dataset, HMMs were fitted for saline and CNO periods of each mouse separately. To perform model fitting, we used the software framework developed by the Linderman Lab (https://github.com/lindermanlab/ssm).

To prepare the data for the HMM analysis, we binned the 6-min pre-task recordings of each session into 1-s bins, resulting in 360 bins. We computed the spike count of each neuron in each bin. The input data for the HMM consisted of an $N \times T$ matrix, in which $N$ represents the total number of simultaneously recorded neurons in the session, and $T$ represents the total number of time bins.

For the analysis during the task in the pre-reward and post-reward periods, we computed the spike count in 0.2-s time bins. We fitted separate HMMs for the pre-reward and post-reward periods for sucrose and water trials. To accomplish this, we concatenated the $M$ trials within a single session and arranged the input data in an $N \times T \times M$ matrix, for which $T = 5$. We chose the bin size of 0.2 s, because this bin size balanced the inference of maximum possible transition states and total spike count used to fit HMMs.

For decoding of switch versus stay using HMM states, we focused on the 4 s pre-reward period. Spike counts were binned using a 1-s bin size, and concatenated across the 4-s window of all trial types. This resulted in an $N \times T \times M$ input matrix, for which $T = 4$, and $M$ represents the total number of recorded trials in the session. Consistent with previous analyses, in our analysis, we retained only sessions with at least five simultaneously recorded neurons.

Given the recorded (observed) spike count over time, we modelled the neuronal activity as a Poisson process, with the mean value dependent on the current neural state. We represented the probability of observing the spike count vector $n(t)$ of $N$ neurons at time bin $t$, given the hidden neural state $S_t = j$, as being distributed as a multivariate Poisson process: $P(n_t | S_t = j) \sim \text{Poisson}(\Lambda; n_t)$, where $\sim$ denotes 'distributed as'. Here, $\Lambda = \{\lambda_1, \lambda_2, ... \lambda_N\}$, and $\lambda_i$ represents the estimated mean activity for the $i$th neuron in state $j$. The vector $\Lambda$ corresponds to the column of the $N \times K$ 'emission matrix' $E$, which provides the firing rates or activation probabilities of observing a specific neuronal pattern when the population activity is in a particular state.

We assumed the dynamics of the neural states to evolve according to a first-order Markovian process, for which the probability of transitioning from one state to another depends only on the current state. This process is summarized by the $K \times K$ 'transition probability' matrix $T$. Additionally, we incorporated an initialization vector $A$, which provides the probability of starting in each state. The HMM was fully described by the set of parameters $\{E, T, A\}$, which were inferred by fitting the model to the recorded neuronal spike counts[73]. We used the Baum–Welch expectation-maximization algorithm to update the model parameters and maximize the likelihood of the observed data. For each time series, we fitted 5 models with a maximum of 100 iterations for each value of the total number of states ranging from 2 up to 50, using randomized initial conditions. The model with the smallest Akaike information criterion score was retained as the best model for further analyses[38]. Subsequently, we used the Viterbi algorithm to estimate the most likely sequence of states over time.

**Agglomerative clustering analysis.** To better characterize the spatial structure of the hidden states, we examined the pairwise correlation between the inferred activity of the states. For state 1 with an activity vector $X = (x_1, x_2, ..., x_N)$, in which $x_i$ represents the activity of neuron $i$, and state 2 with an activity vector $Y = (y_1, y_2, ..., y_N)$, we computed the Pearson correlation coefficient $\rho(X,Y)$ to assess the distance between the states in the neuronal activity space. We calculated the correlation coefficients for all pairs of states and stored them in an $N \times N$ correlation matrix $K$. Subsequently, we performed agglomerative clustering on the correlation matrix.

Specifically, we defined a new distance matrix $D$ as $1 - K$, in which 1 is an $N \times N$ matrix of ones. This matrix served as the input to the agglomerative clustering algorithm, which iteratively combines states to define new clusters according to the pairwise distance. The algorithm initialized each state as a separate cluster with minimum distance (maximum correlation) and iteratively merged two clusters $v$ and $u$ with the smallest distance into a new cluster. The new distance $d$ assigned to the agglomerated clusters was defined as $d(u,v) = \max(\text{dist}(u[p], v[q]))$, in which $p$ and $q$ represent all of the points in the merged clusters $u$ and $v$, also known as the farthest point algorithm. Agglomerative clustering has the advantage of producing a hierarchical structure of clusters, which we represented as a dendrogram. This hierarchical representation allowed us to examine the relationships and similarities between states, specifically how neural states may be nested differently within large clusters in different groups. Agglomerative clustering does not require any assumption regarding the total number of clusters. It iteratively merges the closest states and clusters until all states are merged into one final cluster. We performed the clustering analysis separately for each mouse, visualizing the results with a dendrogram that summarizes the merging of clusters at different levels of distance, ranging from 0 (original states) to 1 (a single cluster).

After examining the clusters, we counted the total number of clusters at different levels of distance, or thresholds, for which the higher the levels of distance, the lower the number of clusters, until reaching only one cluster at the highest distance. We assessed the curves of the number of total clusters and the proportion of total clusters retained relative to the total number of states as a function of thresholds. Comparing

these curves between two groups, a higher number of states at the same threshold value indicates a greater degree of dissimilarity among the inferred states. We retained the proportion of total clusters along these curves from a threshold of 0.1 up to 0.5, resulting in a total of five features that were subsequently used in the decoding of group identity.

We applied the clustering analysis to the pre-task activity using the previously inferred states described in the section entitled HMM, as well as to the pre-reward and post-reward task periods for water and sucrose trials separately.

**Correlation of population activity across time.** To examine how variable population activity was across time during the pre-task period, we performed Pearson correlation on population vectors of neuron firing rates across all time bins (1-s bins). The correlation values were then averaged to assess differences between groups.

**Decoding group identity using neural features (dimensionality, hidden states, firing rates).** This analysis aimed to decode the group identity (that is, control, susceptible or resilient groups, or saline versus CNO groups for the DREADD experiment) on a single-mouse basis by analysing the pre-task activity, for which no behavioural labels were available. As described in the section entitled Analysis of embedding dimensionality, the pre-task activity can be represented as a geometrical object in the firing space, with each axis representing the firing rate of a neuron and each point in the space representing the activity of the neuronal ensemble in each time bin. We sought features that characterized the representational object and were invariant to rotations and scaling transformations, or a subset of these transformations, ensuring shape invariance of the object.

We included only mice with at least five neurons simultaneously recorded during the pre-task period. For each mouse, we computed the cumulative variance explained across the PCs (for more details, see the section entitled Analysis of embedding dimensionality). We considered the cumulative values of the first three PCs as features for decoding. Following the inference of hidden states and the clustering analysis, we calculated the proportion of clusters retained at different thresholds and extracted the values at five distinct thresholds (see the section entitled Agglomerative clustering analysis). Additionally, we computed the mean and standard deviation of the spike count as the last two features. All of the neural features were computed using 1-s bins to optimize the final decoding performance. Overall, we assessed a total of 10 neural features for each mouse.

We used the same Mahalanobis binary decoder procedure as previously described in the section entitled Decoding group identity using behavioural features. Specifically in this case, the input to the binary classifier consisted of an $N \times F$ training matrix and a $1 \times F$ testing matrix, in which N represents the total number of training mice between the two classes, and $F = 10$ represents the total number of features. Before running the classification algorithm, we preprocessed the input matrices by applying a minimum–maximum scaler to the mean and standard deviation of the spike count, ensuring that all features were scaled between 0 and 1 (because the PC cumulative variance and fraction of HMM clusters are defined between 0 and 1 by construction). The decoder was trained and tested for 1,000 iterations, with a new random testing subject selected and removed from the training set for each of them.

The same decoder procedure was also applied during the pre-reward and post-reward periods of the task. For the decoding using vCA1 activity, the training set was defined as 20% of the total number of mice owing to the initial larger sample size.

**Neural population decoding.** As in a previously described method[31], a linear SVM classifier was trained to classify patterns of activity into two discrete categories. Results are reported as the generalized performance of the decoder using cross-validation with a 80:20 training/ testing split. Patterns of activity are defined as the mean firing rate during 0.5-s non-overlapping time bins. Pseudo-population recordings were generated by combining all neurons within the same region and the same group. As it is well known that neural activity in previous trials could strongly influence activity in current trials[74], for all pseudo-population decoding analyses, we balanced the number of trials of each trial type by taking into account both the previous and current trial types. In other words, we have equal numbers of water–water, sucrose–sucrose, water–sucrose and sucrose–water trials (previous–current trials, respectively). Only neurons with at least eight trials per each of the four trial types were included.

To decode current reward, we combined equal numbers of water–water and sucrose–water trials for water trials, and similarly, equal numbers of sucrose–sucrose and water–sucrose trials for sucrose trials. To decode previous reward, we combined equal numbers of water–water and water–sucrose trials for water trials, and similarly, equal numbers of sucrose–water and sucrose–sucrose trials for sucrose trials. To decode intention (stay versus switch), we combined equal numbers of sucrose–sucrose and water–water trials for stay trials, and similarly, equal numbers of sucrose–water and water–sucrose trials for switch trials. We balanced the previous and current reward values when defining switch and stay trials to rule out the confound of reward choices on intention. In other words, the intention signal that we define here is an intention to switch away or stay on the same reward as the previous trial, irrespective of the specific reward value.

To control for the possibility that differences in direction or action sequence (left versus right) coding contributed to reward choice or intention coding, we performed decoding in mice that were given the same value reward in the two lick spouts (same-reward SPT; for more details, see the section entitled Head-fixed SPT). All decoding procedures are the same.

As each group may have different number of cells and trials, we used subsampling procedures to randomly subsample cells (60 neurons for both BLA and vCA1), and within those cells, randomly subsample trials equal to the group with the smallest number of trials. The resulting dataset was used to train SVM and obtain cross-validated decoding accuracies. For each set of subsampled cells, decoding accuracies across random subsampling of trials (repeated ten times) were averaged to obtain a single sample of decoding accuracy. We repeated the whole procedure ten times to obtain statistical comparisons across groups and against shuffled distribution.

For within-time-bin decoding, SVMs were trained using data from one time bin and tested using held-out data from the same time bin. For cross-time-bin decoding, SVMs were trained using data from one time bin and tested using data from the other time bins.

To control for the possibility that differences in lick rates contributed to differences in decoding accuracy for reward choice, we performed additional analysis in which we equalized the lick rates by using only trials with the same lick rates between groups for decoding. Specifically, for susceptible and resilient mice, we analysed only those trials with lick rates within 3–14 Hz in both groups, whereas for saline and CNO mice, we analysed those trials with lick rates within 3–10 Hz in both groups.

For statistical comparisons, decoding accuracy during pre-reward (−4 to −3 s) and post-reward (0 to 1 s) periods was averaged. If the mean decoding accuracy in a group was significantly higher than 2 s.d. of its respective mean shuffled distribution, we then performed additional between-group comparisons (two-way comparison: Mann–Whitney test; three-way comparison: Kruskal–Wallis test followed by Dunn's multiple comparisons test).

**Decoding switch versus stay using HMM states.** In addition to using recorded firing rates during the 4 s pre-reward window to decode switch versus stay, we also trained separate decoders using the smoothed activity of the hidden states inferred by the HMMs. This approach uniquely allowed us to identify population hidden states within this time window,

and specifically those states that may be intention selective, which can then be artificially manipulated to assess their necessity in decoding. It is important to note that the training of the HMM was performed on concatenated trials, which includes the four 1-s bins pre-reward across all trial types. We then rearranged the sequence of hidden states in each trial type a posteriori.

Once the parameters of the HMMs were inferred, the models could smooth the observed data by computing the mean observed activity under the posterior distribution of hidden states[39]. For instance, given the observed activity vector $X$ during a time bin of a trial pre-reward, the HMM inferred a 0.2 probability of being in state $S = 1$ and a 0.8 probability of staying in state $S = 2$. More precisely, $P(S = 1 | X) = 0.2$, and $P(S = 2 | X) = 0.8$. The smoothed observations used to train and test the linear decoder were calculated as $Y = 0.2\mu_1 + 0.8\mu_2$, in which $\mu_j$ represents the inferred mean for the observations in state $j$. Figure 3h is an example spike raster of 15 simultaneously recorded neurons in two switch and two stay trials during 4 s pre-reward from one representative mouse. The different colour-shaded areas are different HMM hidden states, with coloured lines showing the posterior probability for each state.

To ensure robustness, we randomly sampled 60 neurons from each mouse for 10 neuronal subsamples. We generated 1,000 pseudo-trials for each of the 4 trial types, resulting in a total of 4,000 pseudo-trials for the training and testing sets, separately. The input data to train and test the decoder consisted of the smoothed activity assigned to each time bin. We trained and tested a SVM classifier with a linear kernel, similar to the approach used in the population decoding using original firing rates, to decode switch versus stay. In each cross-validation iteration, we randomly selected 100 pseudo-trials as the training set and 20 pseudo-trials as the testing set, for a total of 100 cross-validations. The final decoder accuracy was computed as the average across neuronal subsamples and cross-validations.

To assess the significance of the decoding signal, we compared it to a chance level, defined as 2 s.d. around the theoretical mean of the distribution of accuracies obtained after 100 shuffles of the labels.

**Defining intention-selective states.** We conducted a detailed analysis of the distribution of hidden states across trial types to identify intention-selective states. For each mouse, we computed the fraction of occurrence of each hidden state within the 4-s bins pre-reward across all trials. This distribution was then normalized to the total number of trials multiplied by the number of bins. We assessed this normalized distribution separately for each trial type.

Consistent with the decoding results, we observed that certain states appeared exclusively in either the stay or switch trials, with no occurrences in the other trial types. To quantify the amount of information each state held for the intention value (that is, stay or switch), we computed the Shannon entropy[75]. Specifically, for a given state, we normalized its occurrence frequency in each trial type to the total number of trials. The entropy of each state for the intention value was calculated using the following formula:

$$H_{state} = - [P_{switch} \times \log(P_{switch}) + P_{stay} \times \log(P_{stay})]$$

in which $P_{switch}$ is the occurrence frequency of the state in switch trials (water–sucrose, sucrose–water) and $P_{stay} = 1 - P_{switch}$. An entropy value of 0 indicates that the state provides highly informative signals for the intention to switch or stay. Therefore, we defined an intention-selective state as one with an entropy value of 0 for the intention value.

To decode the intention of switch/stay using hidden states, we first examined the distribution of the fraction of intention-selective states at different clustering thresholds for each mouse, and selected a threshold that yielded the highest number of intention-selective states. We then used the inferred firing rates from these identified intention-selective states to train a linear decoder for classifying the intention of mice to switch or stay.

To compare the fraction of intention-selective states across groups, we calculated the fraction of the intention-selective states out of the total number of hidden states using the first four clustering thresholds (ranging from 0.1 up to 0.4, stepped by 0.1), and compared the resulting distribution.

To examine the necessity and sufficiency of intention-selective states, we first excluded trials that contained intention-selective states in at least three time bins pre-reward. In the opposite approach, we enhanced the presence of intention-selective states in the decoding procedure by considering only those trials that included intention states in at least three time bins before the reward delivery.

**Generalization of susceptible versus resilient decoder to saline versus CNO.** We trained an SVM with a linear kernel to classify whether an animal is susceptible or resilient in the feature space defined by the three behavioural features (sucrose preference, and lick rate DI in the pre- and post-reward) and the four neural features (reward decoding accuracy in the pre- and post-reward, the intention decoding accuracy pre-reward using raw firing rates, and the intention accuracy pre-reward using HMM states). We used one held-out mouse as a testing sample, and the remaining ones as the training set, after balancing the number of training samples per each class. We repeated this procedure for a total of 1,000 cross-validations. We subsequently tested the generalization performance of the decoder in classifying new susceptible mice, not used for training, before and after the treatment of CNO. We assessed the significance of the average decoding performance across the 1,000 cross-validations with respect to a chance interval defined as 2 s.d. around the chance level of 0.5 of the distribution accuracies obtained from 100 shuffles of the labels.

**Decoding group identity of susceptible versus resilient mice using behavioural and neural features.** We trained an SVM with a linear kernel to classify the group identity (control, susceptible and resilient) using neural signatures of the task phase, specifically reward choice decoding performance during the pre- and post-reward period, and the fraction of intention-selective states (see the section entitled Defining intention-selective states). We used one held-out mouse as a testing sample, and the remaining ones as the training set, after balancing the number of training samples per group. We repeated this procedure for 1,000 cross-validations. We compared the average decoding performance across the 1,000 cross-validations to a chance interval defined as 2 s.d. around the chance level of 0.5 of the distribution accuracies obtained from 100 shuffles of the labels.

**MDS.** To visualize the geometric structure of the data, we used multi-dimensional scaling (MDS) transformation to obtain a low-dimensional representation of the data. For pre-task data, we started with the $N \times F$ matrix used for the Mahalanobis decoder, in which $N$ represents the total number of subjects across all three groups, and $F$ denotes the number of features used for decoding the group identity. Before the dimensionality reduction analysis, we normalized each group's data by its variance to reduce noise and enhance the clarity of the final visualization. Next, we performed a diagonalization of the dissimilarity matrix $N \times N$, which contained the Euclidean distances between each pair of subjects in the feature space. We used the same procedure for the task period. In these cases, the input matrix was a $T \times N$ matrix, in which $T$ represents the total number of pseudo-trials, and $N$ denotes the number of neurons. In the example MDS plots, each point is the average firing rate across neurons during the specified time window for the specified trial type, with $n = 1$ subsampling, 60 neurons, 1,000 pseudo-trials per condition.

**Inter-regional connectivity. vCA1–BLA connectivity during pre-task period.** We computed the firing rates of each recorded neuron in 1-s bins within the same region (BLA and vCA1) during a 6-min window (min 2–8) of pre-task recording. We set a minimum of five neurons

simultaneously recorded in both BLA and vCA1. Given the matrix $N \times T$, in which $N$ denotes the number of neurons, and $T$ denotes the number of time bins, we computed the PCs of this matrix for each area, separately. We subsequently aligned the neural dynamics of the first PC between the two simultaneously recorded signals by using canonical correlation analysis (CCA). CCA is a linear transformation used to find common patterns between two signals defined in two different spaces, with the goal of maximizing their correlation. Given two matrices $X \in \mathbb{R}^{N \times T}$ and $Y \in \mathbb{R}^{M \times T}$, in which $N$ and $M$ are the numbers of variables and $T$ is the number of time bins, CCA finds linear combinations $U$ and $V$ of the features in $X$ and $Y$ such that,

$$U = a^T X,$$

$$V = b^T Y,$$

for which the coefficients $a \in \mathbb{R}^{N \times 1}$ and $b \in \mathbb{R}^{M \times 1}$ are chosen to maximize the correlation between $U$ and $V$. We refer to $U$ and $V$ as canonical components for BLA and vCA1, respectively. We subsequently analysed the cross-correlogram between the first canonical component of BLA and vCA1 with time lags from (−50, +50) s and its corresponding power spectral density. We computed the power spectral density from the squared magnitude of the fast Fourier transform coefficients[76] divided by the length of the input signal. We used the frequency at which the power spectral density peaked as an estimate of the dominant frequency of the oscillations between BLA and vCA1. The highest frequency we could access is determined by the Nyquist frequency $f = f_s/2 = 0.5$ Hz, in which $f_s$ is the sampling frequency that in our case is 1 Hz. We tested smaller time bin sizes and chose 1-s bins (hence 1 Hz sampling frequency) owing to low firing rates during the pre-task period, which would otherwise result in many bins with 0 spikes per second.

**vCA1–BLA correlation during pre-reward period.** Given the shorter time window during the task period, we could not use the same CCA analysis. Therefore, to analyse the vCA1–BLA interaction before reward, we computed the correlation of regional average firing rates between simultaneously recorded neurons in the two regions. Specifically, firing rates (10-ms bins) were averaged across all simultaneously recorded neurons in each mouse within the same region (BLA and vCA1). Then Pearson correlation was computed across simultaneously recorded regions within each 1-s time window. The correlation was performed for each trial type (sucrose, water, switch, stay) separately, and Pearson correlation $r$ was transformed to Fisher $z$ to make it normally distributed. To assess how different the inter-regional correlation is in sucrose versus water trials for each animal, we calculated the change in correlation ($\text{corr}_{\text{sucrose}} - \text{corr}_{\text{water}}$).

**vCA1–BLA correlation during intention-selective states.** We subsequently studied the functional connectivity between BLA and vCA1 in susceptible mice during the presence of intention states in the 4 s before reward delivery. We started by selecting time bins (1 s) for which the intention states were detected in BLA ('intention-selective') (see the section entitled Defining intention-selective states), and those bins without intention states ('non-intention-selective'). We then analysed the neural activity of simultaneously recorded BLA and vCA1 neurons during these inferred states, comparing the correlation between the two regions during intention-selective versus non-intention-selective states. We randomly sampled five neurons from each state in BLA and vCA1 and defined the activity matrices $X_{\text{(area)}} \in \mathbb{R}^{N \times K}$ and $Y_{\text{(area)}} \in \mathbb{R}^{N \times L}$, for which $N = 5$ is the number of simultaneously recorded neurons, area is BLA or vCA1, and $K$ and $L$ are the number of intention and no-intention bins, respectively, for a total of four activity matrices. We computed the PCs of each of the four matrices as a denoising procedure and subsequently assessed the Pearson correlation between BLA and vCA1 in each of the first five PCs for each mouse, for intention-selective and non-intention-selective bins separately. We repeated the above procedure 1,000 times, each iteration with different neuron sampling

from each brain area, and we computed the average correlation across different sampling.

## Statistical analysis

No statistical tests were used to predetermine sample size, but the sample sizes used are similar to those generally used within the field[5]. All tests were two-tailed. Data were analysed using parametric one- or two-way repeated measures ANOVA, or paired t-test. In cases in which it was appropriate, ANOVA was followed by post hoc pairwise comparisons with corrections for multiple comparisons. If data were significantly non-normal (with $\alpha = 0.05$), non-parametric tests were used, including the Kruskal–Wallis test or the Mann–Whitney test (between-group comparisons) and Wilcoxon signed-rank test (within-group comparisons), and if appropriate, followed by post hoc comparisons with corrections for multiple comparisons. Categorical data were assessed using chi-squared, or Fisher's exact test if sample size was <5. When comparing to chance, data were considered significant if they were outside 2 s.d. of chance distribution centred around the theoretical chance level (marked by hash symbols on figures). Statistical comparisons between groups were performed for groups that were significantly different from respective chance distribution. Statistical analyses were performed using Graphpad Prism V10.

## Statistics and reproducibility

All experiments were repeated across a minimum of two independent cohorts and showed similar results.

## Reporting summary

Further information on research design is available in the Nature Portfolio Reporting Summary linked to this article.

## Data availability

All source data are provided with this paper. The raw electrophysiology data will be provided upon request to the corresponding author.

## Code availability

All analysis code is provided at https://github.com/mkheirbek.

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

**Acknowledgements** We thank Loren Frank, Vijay Namboodiri, Vikaas Sohal, Jeremy Biane, Alexandra Klein, Joshua Bratsch-Prince and Liam Drew for comments and discussion. F.X. was supported by the Canadian Institutes of Health Research Postdoctoral Scholarship, the Brain and Behavior Research Foundation Young Investigator Award and the Ray and Dagmar Dolby Family Fund. V.F. and S.F. were supported by the Simons Foundation, Neuronex (NSF1707398), the Gatsby Charitable Foundation (GAT3708), the Kavli Foundation and the Swartz Foundation. M.M.G. was supported by the National Institute of Mental Health (F31 MH130127) and the National Institute of Neurological Disorders and Stroke (DSPAN F99/K00 NS130927). M.A.K. was supported by the National Institute of Mental Health (R01 MH108623, R01 MH111754, R01 MH117961 and R01 MH125515), the National Institute on Deafness and Other Communication Disorders (R01 DC019813), the One Mind Rising Star Award, the Human Frontier Science Program (RGY0072/2019), the Esther A. and Joseph Klingenstein Fund, the Pew Charitable Trusts, the McKnight Memory and Cognitive Disorders Award, and the Ray and Dagmar Dolby Family Fund.

**Author contributions** F.X. and M.A.K. conceptualized the project. F.X., V.F., S.F. and M.A.K. were responsible for the methodology. F.X., V.F., N.V., F.G.G., A.K., M.M.G. and L.K.L. were responsible for investigation. F.X. and V.F. were responsible for visualization. F.X. and M.A.K. wrote the first draft of the manuscript. F.X., V.F., S.F. and M.A.K. reviewed the edited the final version of the article.

**Competing interests** The authors declare no competing interests.

**Additional information**
**Correspondence and requests for materials** should be addressed to Mazen A. Kheirbek.

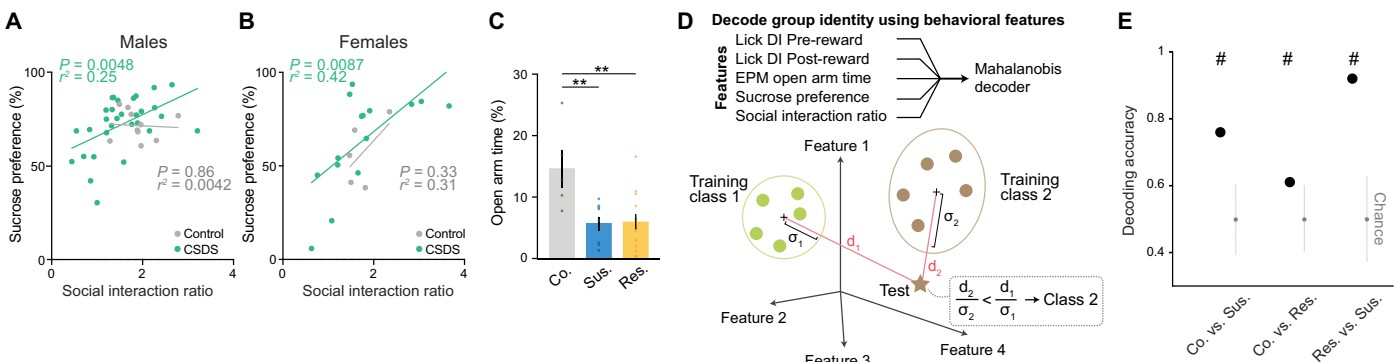

**Extended Data Fig. 1 | Decoding of group identity using behavioral features.**
**A**,**B**, Male CSDS ($n$ = 30) (**A**) and female CSDS ($n$ = 15) (**B**) mice showed a
significant correlation between sucrose preference and social interaction
scores (Pearson correlation). This effect was not observed in controls (male
$n$ = 10, female $n$ = 5). **C**, Both susceptible ($n$ = 10) and resilient ($n$ = 14) mice
showed reduced open arm time in elevated plus maze, in comparison to
controls ($n$ = 5, ANOVA, group x time interaction: $F_{2,26}$ = 7.26, $P$ = 0.0031).
**D**, Schematic of the Mahalanobis decoder trained on behavioural features to
decode group identity. **E**, As further verification that behavioural features

between groups classified using K-means clustering were different, group
identity can be successfully decoded using Mahalanobis decoder trained on
behavioral features including lick rate discrimination index (DI) during pre-
and post-reward, elevated plus maze open arm time, sucrose preference, and
social interaction ratio (control $n$ = 5 mice, susceptible $n$ = 10 mice, resilient
$n$ = 14 mice, 100 cross-validations). Bar plots data are mean ± s.e.m. Chance
distributions are ± 2 x s.d. around theoretical chance level. #Significantly
different from chance; ** $P$ < 0.01.

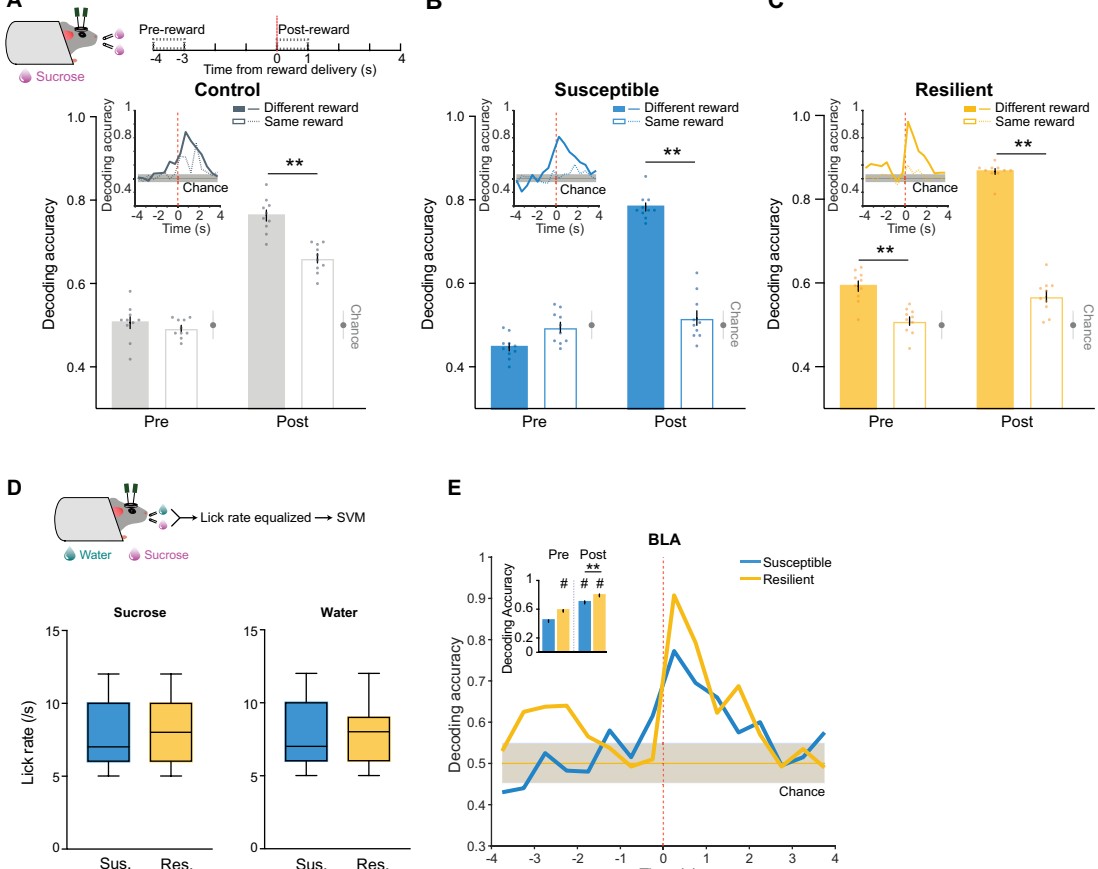

**Extended Data Fig. 2 | Reward choice decoding was not driven by direction coding. A-C**, Mice were given two lick spouts with both delivering the same value reward (sucrose versus sucrose, Same reward) to assess direction (left versus right) coding. Decoding accuracy of BLA neurons in Different reward (sucrose versus water) versus Same reward (sucrose versus sucrose, i.e., left versus right) task in (**A**) control ($n = 10$ subsamplings of 21 neurons per subsampling, 21 total neurons, 100 cross-validations, Post-Reward, Same reward versus chance, $P < 0.05$, Different reward versus Same reward, Mann-Whitney, $P < 0.0001$), (**B**) susceptible ($n = 10$ subsamplings of 21 neurons per subsampling, 123 total neurons, 100 cross-validations, Post-Reward, Same reward versus chance, $P > 0.05$, Different reward versus Same reward, Mann-Whitney, $P < 0.0001$), and (**C**) resilient mice ($n = 10$ subsamplings of 21 neurons per subsampling, 97 total neurons, 100 cross-validations, Pre-Reward, Different reward versus Same reward, Mann-Whitney, $P < 0.001$; Post-Reward, Same reward versus chance, $P < 0.05$, Different reward versus Same reward, Mann-Whitney, $P < 0.0001$). In Pre-reward, resilient group showed greater decoding accuracy of reward choice in Different reward in comparison to Same reward.

In Post-reward, all groups showed greater decoding accuracy of reward choice in Different reward in comparison to Same reward. **D**, To control for the possibility that differences in Post-reward lick rates between resilient and susceptible groups contributed to differences in reward choice decoding, a subset of trials with similar lick rates between the two groups were chosen for SVM decoding (lick rate of 3-14 Hz during Post-reward, box extends from 25th to 75th percentiles, with median in the middle, whiskers extend from minima to maxima). These trials did not differ in lick rates for sucrose (susceptible, $n = 315$ trials, resilient, $n = 2372$ trials) or water trials (susceptible, $n = 311$ trials, resilient, $n = 603$ trials). **E**, A linear SVM decoder was trained to decode reward choice (sucrose versus water) using only trials of similar lick rates between the two groups. BLA neurons in the resilient mice showed significantly higher decoding accuracy than susceptible mice ($n = 10$ subsamplings of 60 neurons, 100 cross-validations, Mann-Whitney, $P < 0.0001$). Coloured lines in line plots indicate mean of subsampling. Bar plots data are mean ± s.e.m. Chance distributions are ± 2 x s.d. around theoretical chance level. #Significantly different from chance; ** $P < 0.01$.

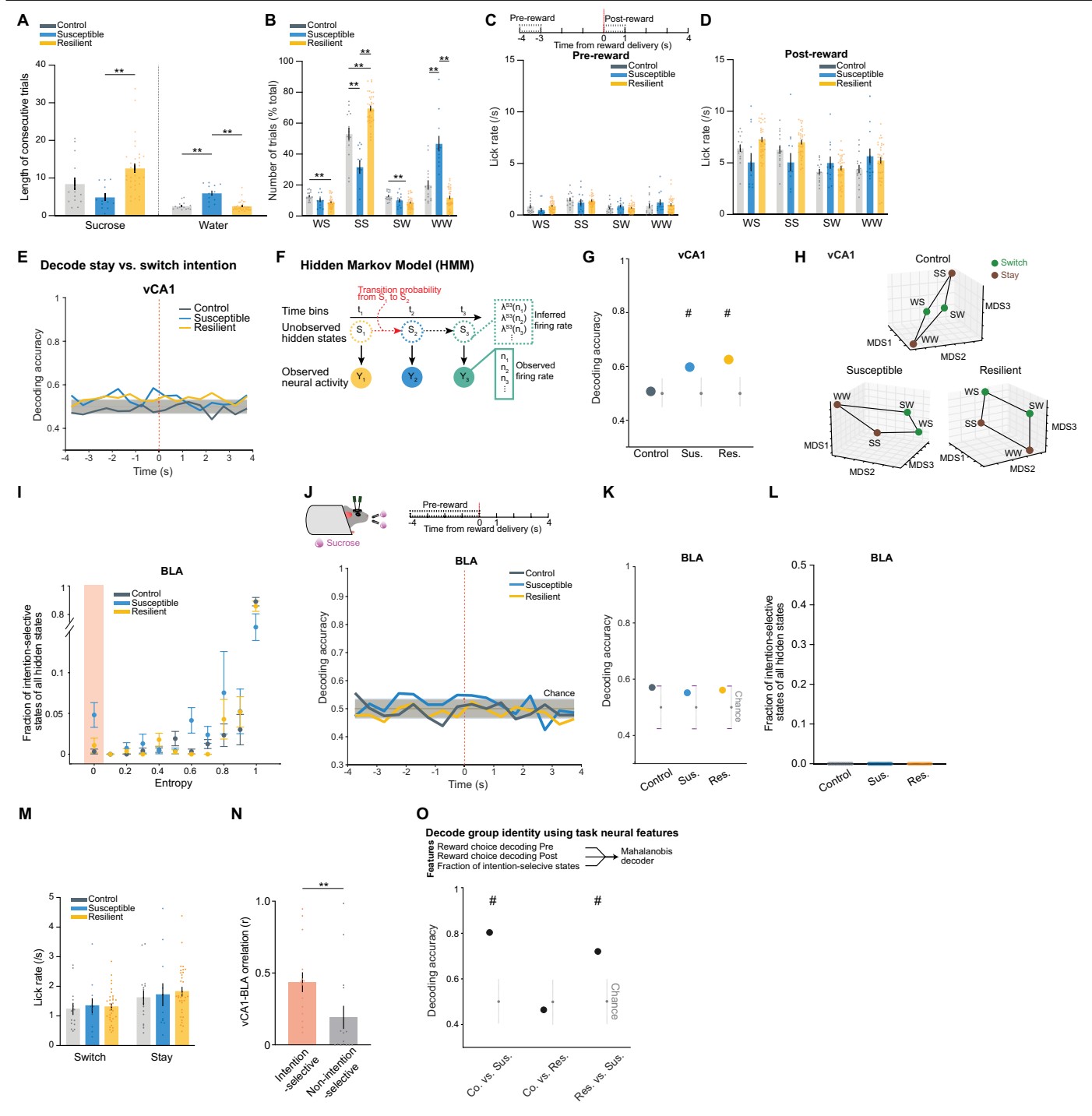

**Extended Data Fig. 3** | See next page for caption.

**Extended Data Fig. 3 | Intention selectivity in BLA as a unique susceptibility signature. A**, Susceptible mice ($n = 12$) showed fewer consecutive sucrose trials (ANOVA, effect of group: $F_{2,57} = 7.60$, $P = 0.0012$), and greater number of consecutive water trials in comparison to control ($n = 15$) and resilient mice ($n = 33$, ANOVA, effect of group: $F_{2,57} = 25.09$, $P < 0.0001$). **B**, Sucrose and water trials were further divided into sucrose–sucrose (SS), water–sucrose (WS), water–water (WW), and sucrose–water (SW) trials after taking into account the previous trial. Comparison of the proportion of trials in each of the 4 trial types revealed that controls showed greater proportion of switch trials (WS, SW). Resilient mice ($n = 33$) showed greatest proportion of SS trials, while susceptible mice ($n = 12$) showed greatest proportion of WW trials (RM-ANOVA, trial type x group interaction: $F_{6,171} = 39.99$, $P < 0.0001$). **C**,**D**, Lick rates of susceptible ($n = 12$), resilient ($n = 33$), and control ($n = 15$) mice for each of the 4 trial types during (**C**) Pre-reward (RM-ANOVA, trial type x group interaction: $F_{6,171} = 2.38$, $P = 0.031$) and (**D**) Post-reward period (RM-ANOVA, trial type x group interaction: $F_{6,171} = 9.80$, $P < 0.0001$). **E**, In vCA1, decoding accuracy of switch versus stay intention using raw firing rates in susceptible and resilient mice was above chance. Coloured lines indicate mean of subsampling ($n = 10$ of $n = 60$ neurons, with $n = 100$ cross-validations. **F**, Schematic of HMM to obtain population hidden states. **G**, Similarly, in vCA1, decoding accuracy of switch versus stay intention using inferred firing rates from HMM in the 4 s preceding reward delivery in susceptible and resilient mice was above chance ($n = 100$ cross-validation; chance: $n = 100$ shuffles). **H**, MDS visualization of inferred firing rates showed that population representations of switch versus stay trials can be linearly separated in vCA1 neurons in susceptible and resilient mice than in controls (MDS example of $n = 1$ subsampling, 1000 pseudo trials/condition). **I**, Average distribution of the of fraction of intention-states across mice at different entropy values (states correlation thresholds of 1-ρ = (0.1, 0.4): control $n = 5$ mice, susceptible $n = 5$ mice, resilient $n = 3$ mice). For each mouse, the state entropy was computed at fixed threshold on the clustering dendrogram (see Methods). Data are mean ± s.d. **J**, In Same reward task with both lick spouts delivering sucrose reward, decoding of intention to switch or stay during the Pre-reward period in the BLA was at chance, suggesting there was no encoding of action sequence of left versus right. Coloured lines in line plots indicate mean of subsampling ($n = 10$ subsamplings of 60 neurons, 100 cross-validations). **K**, In the Same reward task, decoding accuracy of switch versus stay intention using inferred firing rates from HMM in the 4 s preceding reward delivery was at chance in all groups ($n = 100$ cross-validation; chance: $n = 100$ shuffles). **L**, In the Same reward task, there were no intention-selective states in any group (states across 4 correlation thresholds of 1-ρ = (0.1, 0.4): control $n = 5$ mice, susceptible $n = 5$ mice, resilient $n = 3$ mice). **M**, No lick rate differences in switch versus stay trials during the 4 s Pre-reward period were observed across groups (control $n = 15$ mice, susceptible $n = 12$ mice, resilient $n = 15$ mice). **N**, vCA1–BLA correlation in susceptible mice was higher in intention-selective states, in comparison to non-intention-selective states ($n = 3$ susceptible mice, 5 PCs each, see Methods). **O**, Group identity can be successfully decoded between control versus susceptible, and resilient versus susceptible, but not control versus resilient, using Mahalanobis decoder trained on task neural features including reward choice decoding during Pre-reward, Post-reward, and fraction of intention-selective states (control $n = 5$ mice, susceptible $n = 5$ mice, resilient $n = 3$ mice). Data are mean ± s.e.m. unless otherwise stated. Chance distributions are ± 2 x s.d. around theoretical chance level. #Significantly different from chance; *$P < 0.05$, **$P < 0.01$.

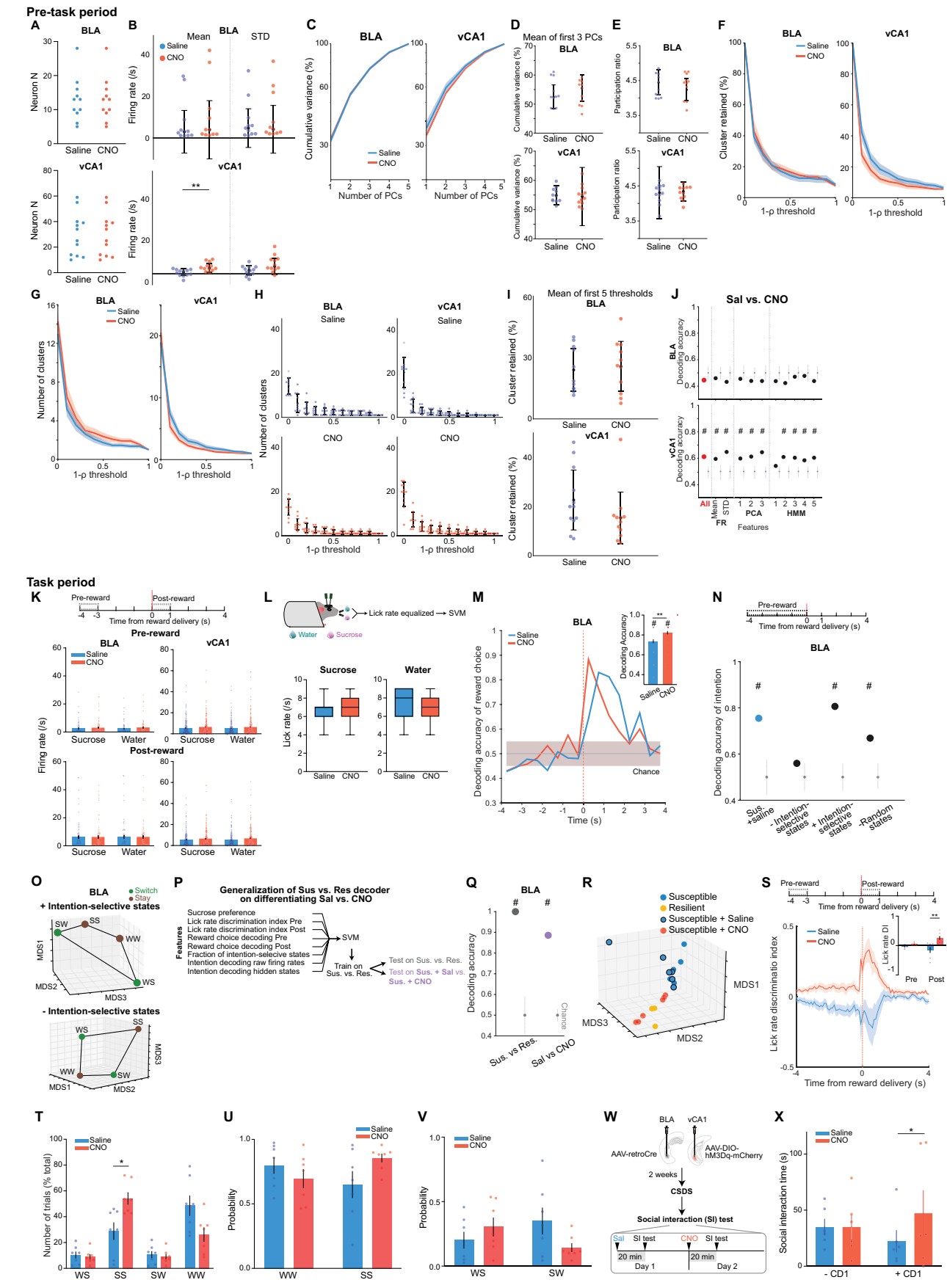

**Extended Data Fig. 4** | See next page for caption.

**Extended Data Fig. 4 | Rescue of dysfunctional vCA1-BLA activity and anhedonia by circuit-specific manipulations. A**, Number of neurons used in BLA and vCA1 in each mouse during saline and CNO period for analysis (BLA $n = 11$ mice per group, vCA1 $n = 12$ mice per group). Same neurons were recorded during both periods. **B**, Mean vCA1 firing rates during Pre-task period was increased following CNO (BLA $n = 11$ mice per group, vCA1 $n = 12$ mice per group, Mann-Whitney, $P = 0.008$). Data are mean ± s.d. **C**, Cumulative variance of PCs did not differ between saline and CNO periods in BLA or vCA1. **D**, Mean of the cumulative variance of the first 3 PCs in BLA and vCA1 (BLA $n = 11$ mice per group, vCA1 $n = 12$ mice per group). Data are median ± s.d. **E**, Participation ratio of BLA and vCA1 (BLA $n = 11$ mice per group, vCA1 $n = 12$ mice per group). **F**, Proportion of HMM clusters across different thresholds in BLA and vCA1. **G**, The number of HMM clusters across different thresholds in BLA and vCA1. **H**, The number of HMM clusters of individual mice across different thresholds in BLA and vCA1 (BLA $n = 11$ mice per group, vCA1 $n = 12$ mice per group). Data are mean ± s.d. **I**, Mean of the proportion of clusters in the first 5 thresholds (BLA $n = 11$ mice per group, vCA1 $n = 12$ mice per group). Data are mean ± s.d. **J**, Despite no statistical difference in each of the FR, PCA, and HMM features, the decoder trained using all features could successfully decode between saline versus CNO periods in vCA1 better than chance (±2 x s.d., $n = 100$ cross-validations; chance: $n = 100$ shuffles). **K**, Firing rates of pseudo-population of BLA ($n = 76$) and vCA1 ($n = 274$) neurons during task showed that vCA1 neurons had elevated firing rates after CNO during both Pre-reward (RM-ANOVA, effect of CNO: $F_{1,1092} = 7.60$, $P = 0.0060$) and Post-reward periods (RM-ANOVA, effect of CNO: $F_{1,1092} = 9.57$, $P = 0.0020$). **L**, To control for the possibility that differences in Post-reward lick rates between saline and CNO periods contributed to differences in reward choice decoding, a subset of trials with similar lick rates between the two groups were chosen for SVM decoding (lick rate of 3–10 Hz during Post-reward). These trials did not differ in lick rates for sucrose (saline, $n = 89$ trials, CNO, $n = 228$ trials) or water trials (saline, $n = 158$ trials, CNO, $n = 121$ trials, box extends from 25th to 75th percentiles, with median in the middle, whiskers extend from minima to maxima). **M**, A linear SVM decoder was trained to decode reward choice (sucrose versus water) using only trials of similar lick rates between the two groups. CNO increased decoding accuracy in BLA neurons in comparison to saline period ($n = 10$ subsamplings of 60 neurons, 100 cross-validations, Mann-Whitney, $P < 0.0001$). Coloured lines in line plots indicate mean of subsampling. Bar plot data are mean ± s.e.m. **N**, Removal of trials containing intention-selective states (-Intention-selective states) during the saline period reduced decoding accuracy of switch versus stay trials to chance, whereas keeping only trials containing intention-selective states (+Intention-selective states) allowed successful decoding of stay versus switch trials. Removal of trials with random states had little effect on decoding accuracy ($n = 6$). Chance distributions are ± 2 x s.d. around theoretical chance level. **O**, MDS visualization showed that keeping only intention-selective states allowed the representations of switch trials to be linearly distinguished from stay trials, whereas removal of intention-selective states prevents the representations of the two trial types from being linearly separated. **P**, An SVM decoder was trained using the listed features to differentiate between susceptible and resilient groups and tested on held-out susceptible versus resilient data, or saline versus CNO. **Q**, The decoder generalizes well to saline versus CNO dataset (susceptible $n = 3$, resilient $n = 3$, saline $n = 6$, CNO $n = 6$). **R**, MDS visualization showed that susceptible mice clustered together with susceptible mice given saline, while resilient mice clustered together whether susceptible mice treated with CNO. **S**, CNO increased lick rate discrimination index during Post-reward period in comparison to saline ($n = 7$ mice, RM-ANOVA, treatment x time interaction: $F_{1,12} = 10.80$, $P = 0.0065$). **T**, CNO increased the proportion of SS trials ($n = 7$ mice, RM-ANOVA with Bonferroni's multiple comparisons test, trial type x treatment interaction: $F_{3,36} = 6.23$, $P = 0.0016$). **U**, CNO altered the proportion of stay (water-water and sucrose-sucrose) trials ($n = 7$ mice, RM-ANOVA with Holm-Sidak's multiple comparisons test, effect of group: $F_{1,12} = 5.61$, $P = 0.036$). **V**, CNO altered the proportion of switch (water-sucrose and sucrose-water) trials ($n = 7$ mice, RM-ANOVA, effect of group: $F_{1,12} = 5.61$, $P = 0.036$). **W,X**, (**W**) CSDS mice were tested in 2 sessions of social interaction (SI) tests, with saline (i.p.) on day 1 and CNO (i.p.) on day 2. (**X**) CNO modestly increased the amount of social interaction time in the present of the aggressor mouse (+CD1, $n = 6$ mice, Saline versus CNO, Fisher's LSD test, $P = 0.044$). Data are mean ± s.e.m. unless otherwise stated. Chance distributions are ± 2 x s.d. around theoretical chance level. #Significantly different from chance; *$P < 0.05$, **$P < 0.01$.

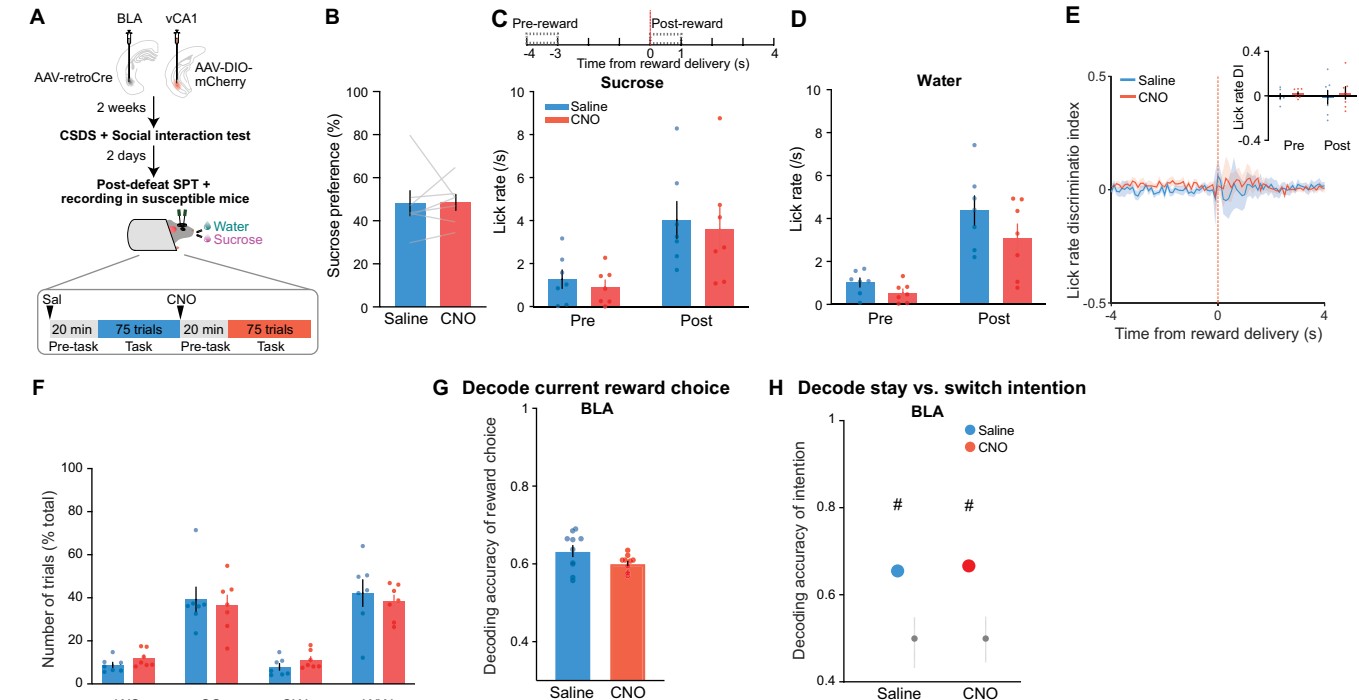

**Extended Data Fig. 5 | CNO had no effect on behaviour or neural activity patterns in mCherry-infused mice. A**, Mice microinfused with the control virus (AAV-DIO-mCherry) and given saline and CNO during the sucrose preference task ($n = 7$ mice). **B**, CNO had no effect on sucrose preference ($n = 7$ mice). **C**,**D**, CNO ($n = 7$ mice) had no effect on lick rates in (**C**) sucrose or (**D**) water trials. **E**, CNO and saline groups showed similar lick rate discrimination index ($n = 7$ mice). **F**, CNO had no effect on the number of trials for each of the 4 trial types ($n = 7$ mice). **G**, CNO had no effect on the decoding accuracy of current reward choice during Post-reward period ($n = 10$ subsamplings of 108 neurons per subsampling, 108 total neurons, 100 cross-validations). **H**, CNO had no effect on the decoding accuracy of switch/stay intention using HMM hidden states during the 4 s Pre-reward period ($n = 10$ subsamplings of 108 neurons per subsampling, 108 total neurons, 100 cross-validations). Bar and line plots data are mean ± s.e.m. Chance distributions are ± 2 x s.d. around theoretical chance level.

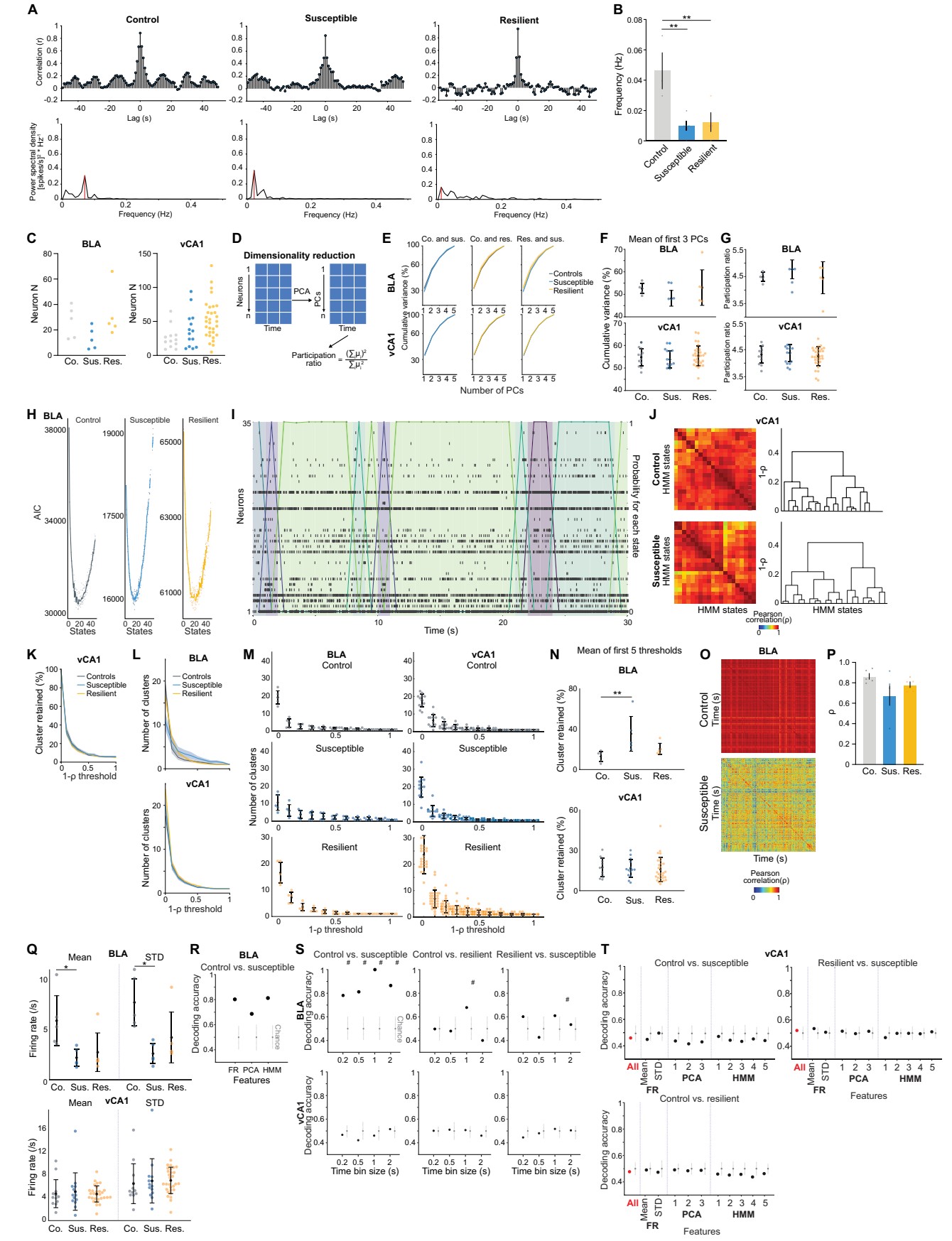

**Extended Data Fig. 6** | See next page for caption.

**Extended Data Fig. 6 | Distinct neural signatures of CSDS mice in the absence of task. A**, Cross-correlogram (top) between BLA and vCA1 activity with time lags from [−50, +50] s and its corresponding power spectral density (PSD, bottom) for an example mouse from each group. The vertical red line in PSD plot indicates the frequency with the highest PSD value. **B**, Susceptible and resilient groups showed lower dominant frequency of vCA1–BLA interaction during Pre-task period (control $n = 3$, resilient $n = 4$, susceptible $n = 5$, Kruskal-Wallis, $P = 0.028$). **C**, Number of neurons used in BLA and vCA1 in each mouse for analysis (BLA $n = 5$ mice per group, vCA1 control $n = 12$, susceptible $n = 14$, resilient $n = 31$ mice). **D**, Schematic of dimensionality reduction using principal component analysis (PCA). The embedding dimensionality was quantified using participation ratio. **E**, There were no statistically significant differences in cumulative variance explained by principal components (PCs) in BLA and vCA1 between groups ($n = 1000$ subsampling, $n = 5$ neurons). Data are mean ± s.e.m. **F**, Mean of the cumulative variance of the first 3 principal components in BLA and vCA1 (BLA $n = 5$ mice per group, vCA1 control $n = 12$, susceptible $n = 14$, resilient $n = 31$ mice). **G**, Participation ratio of BLA and vCA1 (BLA $n = 5$ mice per group, vCA1 control $n = 12$, susceptible $n = 14$, resilient $n = 31$ mice). **H**, Akaike information criteria (AIC) from one example mouse in each group. HMM with the lowest AIC was selected as the best model ($n = 5$ models/#state). **I**, Example of spike raster of 35 neurons simultaneously recorded in the Pre-task period from one representative mouse. The different coloured shaded areas indicate the different HMM hidden states. The colored lines indicate the posterior probability for each state. **J**, Two examples of HMM states correlation matrices for one control ($n = 20$ hidden states) and one susceptible ($n = 20$ hidden states) mouse, with respective dendrograms of agglomerative clustering in vCA1. **K**, There was no difference in the proportion of distant hidden states in vCA1 between groups. **L**, The number of clusters across thresholds did not differ between groups in BLA and vCA1. Data are mean ± s.e.m. **M**, The number of clusters of individual mice (BLA $n = 5$ mice per group, vCA1 control $n = 12$, susceptible $n = 14$, resilient $n = 31$ mice). **N**, Mean of the proportion of clusters in the first 5 thresholds showed that susceptible mice in BLA had greater proportion of unique hidden states in comparison to controls (BLA $n = 5$ mice per group, vCA1 control $n = 12$, susceptible $n = 14$, resilient $n = 31$ mice, Kruskal-Wallis, $P = 0.0018$). No group difference was found in vCA1. **O**, Two example heatmaps of population activity correlation in BLA over time, showing that population activity patterns were much more correlated in the control (top) than the susceptible (bottom) mouse. **P**, Average correlation of population activity across time in the BLA showed a trend towards lower correlated activity in the susceptible mice ($n = 5$ mice per group). **Q**, In BLA, susceptible mice showed reduced firing rates mean (Kruskal-Wallis, $P = 0.020$) and s.d. (Kruskal-Wallis, $P = 0.0077$) in comparison to controls (BLA $n = 5$ mice per group, vCA1 control $n = 12$, susceptible $n = 14$, resilient $n = 31$ mice). **R**, Firing rate (FR), PCA, and HMM features each alone could successfully decode control versus susceptible mice in BLA (control $n = 5$ mice, susceptible $n = 5$ mice). **S**, Different time bin sizes were tested and the one that allowed the highest decoding accuracy between groups was chosen as the optimal bin size ($n = 100$ cross-validations; chance: $n = 100$ shuffles, BLA $n = 5$ mice per group, vCA1 control $n = 12$, susceptible $n = 14$, resilient $n = 31$ mice). **T**, Group identity could not be decoded using Mahalanobis decoder trained on neural features in vCA1. The importance of each neural feature in decoding was examined by systematic removal of each of the features (subsequent columns) (control $n = 12$ mice, susceptible $n = 14$ mice, resilient $n = 31$ mice). Data are mean ± s.d., unless otherwise stated. Chance distributions are ± 2 x s.d. around theoretical chance level. #Significantly different from chance; *$P < 0.05$; **$P < 0.01$.

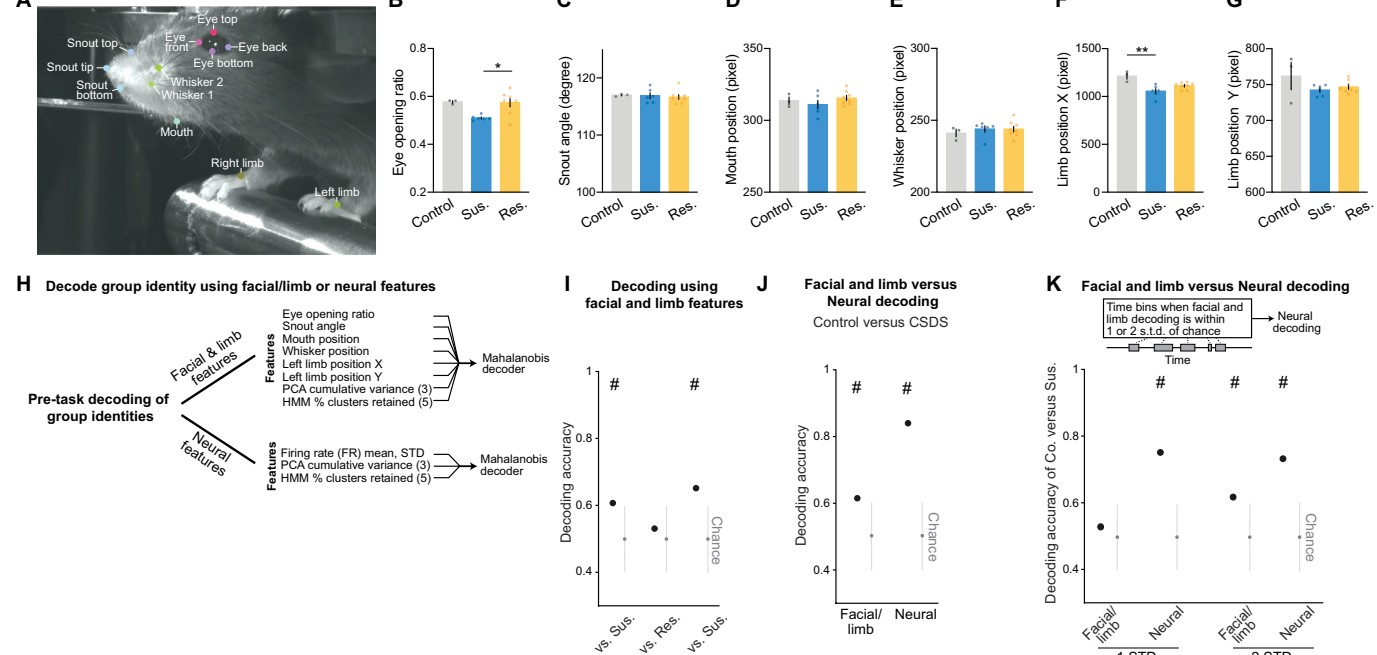

**Extended Data Fig. 7 | Spontaneous face and limb movements during Pre-task did not decode group identity as well as neural features.** **A**, Schematic of 12 keypoints tracked using DeepLabCut. **B**, Susceptible mice showed smaller eye opening ratio than resilient mice (control $n = 3$ mice, susceptible $n = 6$ mice, resilient $n = 8$ mice, Kruskal-Wallis, $P = 0.014$). **C**–**E**, Mice (control $n = 3$ mice, susceptible $n = 6$ mice, resilient $n = 8$ mice) showed similar (**C**) snout angle, (**D**) mouth position, and (**E**) whisker position across groups. **F**,**G**, Susceptible mice showed a smaller magnitude of limb movement in horizontal (X, control $n = 3$ mice, susceptible $n = 6$ mice, resilient $n = 8$ mice, Kruskal-Wallis, $P = 0.003$) but not (**G**) vertical (Y) direction (control $n = 3$ mice, susceptible $n = 6$ mice, resilient $n = 8$ mice). **H**, Mahalanobis decoders were trained on either facial and limb features or neural features to decode group

identity. **I**, Decoding accuracy using facial and limb features showed above chance decoding for control versus susceptible, and resilient versus susceptible groups ($n = 1000$ cross-validations). **J**, Neural feature decoding for control versus CSDS group identities outperformed facial and limb features ($n = 1000$ cross-validations). **K**, Decoding using neural features was performed using only time bins when facial and limb feature decoding was within 1 or 2 s.d. of chance, as a further comparison of neural versus facial and limb feature decoding. Decoding accuracy of control versus susceptible group identities using neural features outperformed facial and limb features ($n = 1000$ cross-validations). Data are mean ± s.e.m. Chance distributions are ± 2 x s.d. around theoretical chance level. #Significantly different from chance *$P < 0.05$; **$P < 0.01$.

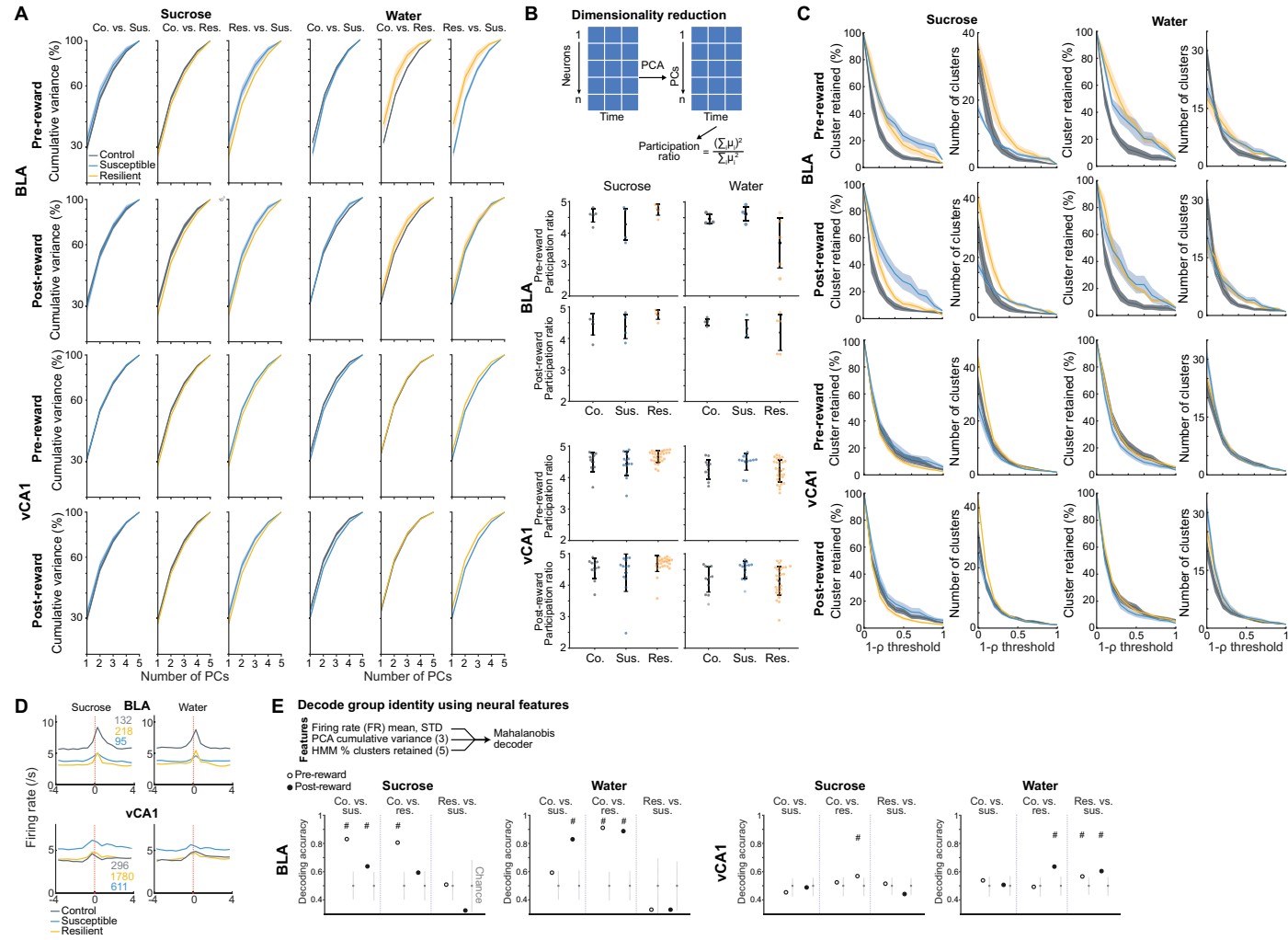

**Extended Data Fig. 8 | Dysfunctional single cell and population-level correlates for reward choice in mice susceptible to chronic stress.**
**A**, Cumulative variance of PCs in BLA and vCA1 during Pre-reward and Post-reward periods revealed no difference between groups (n = 1000 subsampling, n = 5 neurons). **B**, Participation ratio of BLA and vCA1 during Pre-reward and Post-reward periods showed no difference between groups (BLA n = 5 mice per group, vCA1 control n = 12, susceptible n = 14, resilient n = 31 mice). Data are mean ± s.d. **C**, The proportion and number of HMM clusters across different thresholds showed no statistical difference between groups, but BLA neurons

in susceptible mice showed a trend towards higher proportion of unique clusters. **D**, Trial-averaged firing rates of pseudo-populations of BLA and vCA1 neurons across groups. Number of neurons are labelled in corresponding group colours. **E**, Group identity could be decoded better than chance during specific time windows and trial types (n = 100 cross-validations; chance: n = 100 shuffles). Data are mean ± s.e.m. unless otherwise stated. Chance distributions are ± 2 x s.d. around theoretical chance level. #Significantly different from chance.

# Reporting Summary

## Statistics

For all statistical analyses, confirm that the following items are present in the figure legend, table legend, main text, or Methods section.

| n/a | Confirmed | |
|---|---|---|
| ☐ | ☒ | The exact sample size (*n*) for each experimental group/condition, given as a discrete number and unit of measurement |
| ☐ | ☒ | A statement on whether measurements were taken from distinct samples or whether the same sample was measured repeatedly |
| ☐ | ☒ | The statistical test(s) used AND whether they are one- or two-sided *Only common tests should be described solely by name; describe more complex techniques in the Methods section.* |
| ☒ | ☐ | A description of all covariates tested |
| ☐ | ☒ | A description of any assumptions or corrections, such as tests of normality and adjustment for multiple comparisons |
| ☐ | ☒ | A full description of the statistical parameters including central tendency (e.g. means) or other basic estimates (e.g. regression coefficient) AND variation (e.g. standard deviation) or associated estimates of uncertainty (e.g. confidence intervals) |
| ☐ | ☒ | For null hypothesis testing, the test statistic (e.g. *F*, *t*, *r*) with confidence intervals, effect sizes, degrees of freedom and *P* value noted *Give P values as exact values whenever suitable.* |
| ☒ | ☐ | For Bayesian analysis, information on the choice of priors and Markov chain Monte Carlo settings |
| ☒ | ☐ | For hierarchical and complex designs, identification of the appropriate level for tests and full reporting of outcomes |
| ☐ | ☒ | Estimates of effect sizes (e.g. Cohen's *d*, Pearson's *r*), indicating how they were calculated |

*Our web collection on statistics for biologists contains articles on many of the points above.*

## Software and code

Policy information about availability of computer code

| Data collection | Neuropixels data were collected using SpikeGLX (v20220101-phase30). Behavioral data were collected using Arduino and CoolTerm 1.9. Facial and limb movement videos were recorded using the MATLAB Image Acquisition Toolbox. |
|---|---|
| Data analysis | Data were analyzed using custom codes in MATLAB and Python, including the use of the following: scikit-learn, Kilosort 2, Kilosort 4, Phy2, DeepLabCut, SSM toolbox (https://github.com/lindermanlab/ssm), and Allen CCF (https://github.com/cortex-lab/allenCCF). Code available on https://github.com/mkheirbek. |

For manuscripts utilizing custom algorithms or software that are central to the research but not yet described in published literature, software must be made available to editors and reviewers. We strongly encourage code deposition in a community repository (e.g. GitHub). See the Nature Portfolio guidelines for submitting code & software for further information.

## Data

Policy information about availability of data

All manuscripts must include a data availability statement. This statement should provide the following information, where applicable:
- Accession codes, unique identifiers, or web links for publicly available datasets
- A description of any restrictions on data availability
- For clinical datasets or third party data, please ensure that the statement adheres to our policy

All source data is provided with this paper. The raw electrophysiology data will be provided upon reasonable request.

# Research involving human participants, their data, or biological material

Policy information about studies with human participants or human data. See also policy information about sex, gender (identity/presentation), and sexual orientation and race, ethnicity and racism.

| | |
|---|---|
| Reporting on sex and gender | N/A |
| Reporting on race, ethnicity, or other socially relevant groupings | N/A |
| Population characteristics | N/A |
| Recruitment | N/A |
| Ethics oversight | N/A |

Note that full information on the approval of the study protocol must also be provided in the manuscript.

# Field-specific reporting

Please select the one below that is the best fit for your research. If you are not sure, read the appropriate sections before making your selection.

☒ Life sciences   ☐ Behavioural & social sciences   ☐ Ecological, evolutionary & environmental sciences

For a reference copy of the document with all sections, see nature.com/documents/nr-reporting-summary-flat.pdf

# Life sciences study design

All studies must disclose on these points even when the disclosure is negative.

| | |
|---|---|
| Sample size | Sample sizes for animals were based on previous studies using the techniques described (e.g. Krishnan et al., Cell, 2007). No further sample size calculations were used. |
| Data exclusions | For DREADD experiments, animals with off-target expressions of viruses were excluded, in order to specifically analyze the effects of manipulating only the region of interest. |
| Replication | All recording and behavioral data were acquired from a minimum of 2 independently performed experimental cohorts. Data were randomly subdivided during analysis to ensure reproducibility across subsets. |
| Randomization | Mice were randomly assigned to control vs. CSDS groups before CSDS training. For DREADD experiments, mice were randomly assigned to mCherry vs. hM3Di groups at time of surgery. |
| Blinding | Experimenters were blind to the condition and group assignments of mice. The same analysis pipeline was applied to all animals. |

# Reporting for specific materials, systems and methods

We require information from authors about some types of materials, experimental systems and methods used in many studies. Here, indicate whether each material, system or method listed is relevant to your study. If you are not sure if a list item applies to your research, read the appropriate section before selecting a response.

## Materials & experimental systems

| n/a | Involved in the study |
|---|---|
| ☒ ☐ | Antibodies |
| ☒ ☐ | Eukaryotic cell lines |
| ☒ ☐ | Palaeontology and archaeology |
| ☐ ☒ | Animals and other organisms |
| ☒ ☐ | Clinical data |
| ☒ ☐ | Dual use research of concern |
| ☒ ☐ | Plants |

## Methods

| n/a | Involved in the study |
|---|---|
| ☒ ☐ | ChIP-seq |
| ☒ ☐ | Flow cytometry |
| ☒ ☐ | MRI-based neuroimaging |

# Animals and other research organisms

Policy information about studies involving animals; ARRIVE guidelines recommended for reporting animal research, and Sex and Gender in Research

| | |
|---|---|
| Laboratory animals | Adult (8-12 weeks old) male and female C57BL/6J mice, and adult (5-6 months old) CD1 retired male breeder mice were used. |
| Wild animals | No wild animals were used in the study. |
| Reporting on sex | Both male and female mice were used. |
| Field-collected samples | No field-collected samples were used in the study. |
| Ethics oversight | All procedures were conducted in accordance with the NIH Guide for the Care and Use of Laboratory Animals and institutional guidelines. |

Note that full information on the approval of the study protocol must also be provided in the manuscript.

# Plants

| | |
|---|---|
| Seed stocks | N/A |
| Novel plant genotypes | N/A |
| Authentication | N/A |

