## [Peer Review File · Nature]

Understanding the neural code of stress to control anhedonia

Corresponding Author: Dr Mazen Kheirbek

Version 0:

Reviewer comments:

Referee #1

(Remarks to the Author)

This is an elegant study in which the authors used the chronic social defeat stress procedure to record signatures of stress susceptibility versus stress resilience across the brain's limbic circuitry. While other studies have reported signatures of stress susceptibility, the present investigation is the first to identify such clearly distinctive signatures of stress resilience. The present study is further distinguished by its identification of unique patterns of single neuron activity in the BLA that differentiate susceptible and resilient outcomes—this is the first characterization of such responses in the amygdala and the results are impressive. Another unique feature of this study is its focus on CA1-to-BLA projections which have not, as far as I know, previously been characterized for a role in vulnerability to chronic stress. (BLA-to-CA1 projections have been studied in fear conditioning which is very different.) The authors provide causal evidence, through the use of chemogenetics, to establish the role of BLA-projecting CA1 neurons in controlling susceptibility versus resilience to CSDS. Finally, I am especially impressed by the authors' use of relatively sophisticated behavioral outcome measures that begin to address psychological domains such as intention which are obviously challenging to access in a mouse. Importantly, they very nicely stick to experimental observations and do not over-interpret their findings.

All told, I am enthusiastic about this study and cannot identify additional experiments that are required to supplement the already extensive experimental work presented. The manuscript is well-written, the figures are clear, and the data are compelling. The authors also did an admirable job in citing the relevant literature.

My only request for clarification concerns the sex of the animals studied. In Methods, the authors say that they used males and females and sufficiently describe the CSDS procedures used for both sexes. However, it's not clear to me whether the data in all of the figures represent a combination of results from males and females (assuming the authors did not observe a sex difference in their main measures). This is totally fine—but it would be worth a sentence in the Results to this effect.

Referee #2

(Remarks to the Author)

The manuscript by Xia et al. describes neural signatures associated with an animal's stress susceptibility and resilience, which may underlie a clinical symptom of anhedonia in major depressive disorder. The authors examined the neural activity in the basolateral amygdala (BLA) and identified the significant differences in its representations of reward choices and intentions between stress-susceptible and resilient mice. Furthermore, the authors demonstrated the ensemble BLA activity when animals were at rest could be used to distinguish whether animals were subjected to chronic social defeat stress or not. Finally, the authors found that targeted manipulation of vCA1 inputs to the BLA in susceptible mice can rescue their abnormal features in both neural activity and behaviours. The manuscript addresses an important question in neuroscience and the results and analyses are overall well performed. However, in several parts of the manuscript, important control experiments and analyses are missing, which should be addressed before consideration of publication. The following are detailed comments:

1. In Figure 2. The authors found that the activity of BLA neurons discriminates between sucrose and water trials, and then concluded that these neurons discriminate between reward choices (sucrose or water). However, because these choices are also associated with different actions (left and right licking), these results can also be explained from the aspect of differences in action planning and its neural coding. For example, I wonder if the authors monitored the same neurons before and after the change of spout designation to distinguish if the neurons are tuned to reward identity or action. Furthermore, it is also plausible that licking behaviour itself can directly influence the neural activity level in the BLA. If so, the difference in the proportion of reward-selective neurons between resilient and susceptible mice (Fig. 2C) could simply result from the difference in their licking rate (Fig. 1H). The authors should exclude these alternative possibilities.

2. Figure 3 shows the animal's intention-specific states - whether the mice switched or stayed on the previous choice. While the authors interpret the results as rumination-like states in these susceptible animals, I wonder if this could simply be a consequence of mice's indiscrimination between water and sucrose rewards. I think that the mice's decision process in this task would depend on the degrees of how much mice care about the reward type (water or sucrose) or the action type (right or left). When the difference in reward valence between choices is small, a mouse's choice strategy will rely more on action sequences, which may be observed as intention states in susceptible mice? I think the authors should explore this possibility, for example, by examining whether these intention-specific states emerge even in control mice when the two spouts provide the same reward.

3. The experiments in Fig. 4 examined neural activity in the absence of any stimuli or task demands under the head-fixed condition. However, the details of behavioural states of these mice are not described in the manuscript (or were all the movements restrained?). Because even spontaneous behaviors of mice (e.g. running, sleeping, staying still etc.) can largely affect the brain's activity states and social defeat stress is known to increase an animal's anxiety level (Krishnan et al., 2007), it is important to examine whether these behavioral features during the head-fixed condition can explain the observed differences in neural activity or not.

4. In Fig 4G, the main claim is that the activity patterns in the BLA at rest can serve as a powerful biomarker to distinguish whether mice had chronic stress or not. Here, my confusion is that the compared behavioural parameters used to distinguish between control and stress-exposed mice were the lick rate, sucrose preference, and social interaction, which are the features used to distinguish resilient and susceptible mice after social defeat stress (Fig. 1). I thus feel that this conclusion may be misleading because these compared behavioural features are probably not optimal for distinguishing mice's stress-experience, and indeed, a previous study identified the other behavioural feature that is commonly observed in both susceptible and resilient mice after social defeat stress (i.e., increased anxiety-like behaviour in the elevated plus maze; Krishnan et al., 2007). If the authors want to claim the superiority in classification performance by neural signatures, it would be fair to compare with this defined behavioural feature in its ability to distinguish mice with stress experience.

5. In Fig. 4E, the authors tested whether the decoder based on neural activity at rest can distinguish between stress-susceptible and resilient animals (not necessarily good performance, though). But it would be important to perform a similar analysis by using the authors' other identified neural signatures during the task phase (representations of reward choice and intention), which would be a critical test of the usefulness of these neural signatures in distinguishing mice with stress susceptibility and resilience without relying on behavioural features.

6. For Fig. 5F, in the same line as my previous comment 1, it is important to show that the increased decoding performance is not a simple consequence of the difference in licking rates between water and sucrose because licking behaviour itself might affect firing rates directly.

7. In Fig. 5HI, although the authors consider the impaired decoding of stay/switch intention after the CNO injection as evidence of rescued abnormality, when considering the DREADDs enhancer is often used as activity perturbation, the disturbance in certain neural representations and resulting impairment in decoding ability would naturally be expected by this manipulation. I would thus consider it necessary for the authors to show the specificity of the manipulation on a particular neural feature, rather than its general disturbance of neural representations. For example, I wonder whether hidden-state representations in the BLA behave in the same way as control and resilient animals during the task (Fig. 3) and the rest (Fig. 4) after the CNO injection to susceptible mice?

8. In Fig. 5L, the authors only tested the sucrose preference as a behavioural feature rescued by the manipulation, which I do not think is sufficient, and they should also examine the social interaction score to align with the definition of stress susceptibility (Fig. 1). It would also be ideal if the authors construct a classifier of animal susceptibility/resilience using either behavioural features or neural signatures, and examine whether the DREADD manipulation can change the decoder's classification of the treated mice from stress susceptible to resilient.

9. CNO has been shown to have a nonspecific impact through its metabolite clozapine (e.g. Gomez et al., 2017). While the authors confirmed that CNO alone (without hM3D) did not affect the animal's sucrose preference (Ext Data Fig. 5R), similar control experiments would be necessary for other behavioural features (Fig. 1) and neural signatures (Fig. 5F-K), as the impact of clozapine is uncertain here.

Referee #3

(Remarks to the Author)

This innovative and important study effectively couples behavior, neural recordings and powerful computational analytical approaches to provide genuinely novel insight into the differential effects of (social defeat) stress on neuronal population dynamics in the basolateral amygdala and ventral hippocampus. It's also noteworthy that although the computational treatment of the data is unusually comprehensive for a mouse behavioral study, the authors are to be commended for doing a good job of keeping the main text accessible to a broad readership. Nonetheless, there are several areas where further clarification is needed.

The observation that BLA population activity exhibits higher dimensionality in stress-susceptible mice (and cross-group decodability) during periods in which there are no explicit task demands is very interesting. As this phenotype is evident on the period in which the mouse is head-fixed and awaiting the start of the task ('stimulus-free pre-task condition'), can it be

solely attributable to neuronal activity 'at rest,' or is there potentially an element of expectation about the upcoming task demands? Given the susceptible mice exhibit evidence of 'rumination-like' activity on-task, this may be a non-trivial consideration – and will partly depend on how much task-training the mice have had by this point. Whatever is driving the phenotype makes it no less interesting, but it would be good to get some clarity or discussion of this.

The enhanced representation of upcoming responses based on past choice in the stress-susceptible mice raises the perennial issue with behavior-neuronal correlations as to the extent to which the neuronal activity reflects differences in movement rather than/in addition to intention (e.g., do stress-susceptible mice have slower response latencies during these trials than could impact the presentations). Please address this question.

The hippocampus-BLA chemogenetic stimulation representation-rescue experiment nicely sews up the narrative. Were these effects associated with a change in BLA neuronal activity (as they were in hippocampus)?

Please provide an explanatory note for the reader for cases in which a subsample of neurons were analyzed (e.g., due to matching population size across groups). Also for clarification, my reading is that the definition of reward-selective neurons is that they are preferentially responses to either sucrose or water (rather than sucrose-preferring) – if this is the case please make it explicit for the reader because 'reward-selective' implies selectivity for the higher value reward option (i.e., sucrose).

Line 804: 'activity' not 'acidity'

Referee #4

(Remarks to the Author)

Xia et al. uncovered neuronal mechanisms underlying anhedonia. By applying high density spike recordings to stress mouse models, they identified reward-related and intention-related neuronal activity in BLA and vCA1 specific to susceptible mice, suggesting a rumination-like signature. BLA spike patterns in such susceptible mice showed larger numbers of spike activity states, correlated with their susceptibility. Chemogenetic activation of vCA1 inputs to the BLA in susceptible mice restored these neural dynamics and anhedonic behavior.

Overall, the experiments are well designed and provide novel findings. I am enthusiastic about this manuscript. Data and methodology is valid, but some experimental conditions need more explanation. Some figures were a bit too small, making the presentation hard to follow. The statistical values were appropriately used and reported. The conclusion is clear, but there were some challenging results to interpret, such as in Figure 4. The references were appropriate. I have summarized additional comments below.

(Major)

(1) What is meant by 'control'? I could not find a description regarding this. In the context of the CSDS model, 'control' should typically refer to conditions such as segregating mice within the same cage without physical contact. Is this the intended meaning, or does it refer to 'naive' mice with no interventions?

(2) Were there any differences in behavioral outcomes between male and female mice (Fig. 1G)? If resilient and susceptible phenotypes can be simply distinguished based on gender, the interpretation of the results in this manuscript could significantly change, necessitating clarification.

(3) Figure 1F indicates that many susceptible mice show a preference for water intentionally, with 50% or less preference for sucrose. Indeed, as observed in Fig. 3B, susceptible mice exhibit a clear increase in WW (water-water preference) and a decrease in SS (sucrose-sucrose preference). Is this intentional avoidance of sucrose known as a sign of anhedonia?

(4) In addition, if this is a prominent phenotype in susceptible mice, it suggests that WW and SS may have significantly different meanings for the mice. In that case, beyond combining them into an overall measure like 'intension,' it is necessary to conduct a more detailed analysis and provide an explanation, especially regarding the differences between WW (intentional reward-avoiding) and SS, to determine if there are variations in the representations of BLA neural activity.

(5) Line 194-200: This paragraph might be challenging for someone not analytically inclined. As researchers interested in anhedonia and behavioral neuroscience are likely to be in the minority when it comes to those well-versed in such analyses, a more detailed explanation and interpretation from the analysis would be beneficial.

(6) From Line 238 and Figure 4: I could not understand how the story on spontaneous activities connects with the other task sections. Certainly, spontaneous activity is an intriguing aspect of brain function and could potentially serve as one of the physiological markers, as suggested by the authors. However, discussions and figures from 1 to 3 primarily focus on tasks, and Figure 5 specifically addresses the impact of chemogenetics on task performance. Furthermore, CNO had little impact on the spontaneous activity of the BLA, as shown in Extended Fig. 5I and 5J. Take together, the data on spontaneous activity is likely to be clearer if presented separately from the main task-related data, perhaps following all the data on the tasks.

(7) Related to (6), the authors mentioned "functional connectivity" at Line 241. However, no tests on inter-regional correlations or coherent analyses between BLA and vHC are provided. To provide a more comprehensive description of such resting-state activity, it is crucial to investigate the extent of spike (or LFP) (cross-)correlations among neurons related to reward and intention in the vHC and BLA for each group. These analyses might contribute to strengthening the authors' hypothesis regarding the difference in the stress group between vCA1 and BLA (described at Line 388-393).

(8) Related to (6), Fig. 4D: It is hard to interpret why a greater number of distinct neural population states of spontaneous activity in susceptible mice are related to anhedonic behavior and intension specific states in the task. Is it possible for the authors to provide some discussion on this?

(9) Fig. 5A-B: This study simply subtracted correlations between sucrose and water. As the correlation coefficient is not an additive measure, this subtraction is mathematically not ideal. For example, please consider to use other more reliable

measures such as Fisher's Z transformation, in which Pearson correlation coefficients (cofiring) were transformed into Z value that can be used to calculate a confidence interval for coefficients.

(Minor)

- (1) As this recording is acute, not chronic (implantation), recording, please mention this more clearly in the first Results section.
- (2) Fig. 1D: I could not correctly read this rasterplot. Why plots from multiple cells seem to be combined in time? Some spaces between cells may be needed.
- (3) Fig. 1I (and some figures): they say "Controls". How it was different from "Control"?
- (4) Fig. 2B: Please explain what is "auROC (acronym)" in the manuscript.
- (5) Fig. 2B: The claim of these traces is that water (blue) and sucrose (magenta) trials are well separated in resilient mice. But it is not clear from these traces. Please consider to provide more representative traces that better illustrate this claim.
- (6) Fig. 3A-E (and overall figures): The use of colors is confusing. For instance, in Fig. 3B, 'control' is represented in purple and 'resilient' in orange, while in Fig. 3C, 'stay trial' is in purple and 'switch trial' in orange. Furthermore, in Fig. 3H, orange and cyan are used for another purpose. Furthermore, some bar graphs (e.g. Fig. 2C, 3B, 3L...) use gray for control while it is represented in purple in the legends. Throughout the manuscript, please ensure a more coherent and consistent use of colors for better clarity.
- (7) Fig. 3H, 3J, 3K and Extended Fig. 3G: it was hard to understand what these graphs of HMM models meant (while their conclusions are understandable). Please add more explanations.
- (8) Fig. 4C and 4D: Why the Y-axis (%) is so different between these two graphs?
- (9) Fig. 5D: the word "DAPI (?)" is hard to see.
- (10) Fig. 5I legends: "n = 100" (spaces needed)
- (11) Fig. 5L: gray lines are too thin.
- (12) Line 321: should be "Fig. 5J" (period needed)
- (13) Line 660: "4 C" (misspelling)
- (14) Line 803: "neural acidity"?

Version 1:

Reviewer comments:

Referee #1

(Remarks to the Author)

The authors have done an admirable job of address my and the other reviewers' comments.

Referee #2

(Remarks to the Author)

The authors have addressed all my concerns and confusions. I sincerely thank them for their detailed answers to my questions by performing a series of new experiments and analyses.

Referee #3

(Remarks to the Author)

The authors have conscientiously addressed the Referees' comments, including my own, with thoughtful responses/revisions and thorough experimental additions. The manuscript is further improved as a result and I have no further requests.

Referee #4

(Remarks to the Author)

This manuscript has been significantly improved in response to the reviewers' comments. I have no further suggestions for revision except for the following minor comments.

(Minor)

- (1) Fig.1I legend: " $*P < 0.05$ " is not necessary.
- (2) Line 336: we found that "BLA, but not vCA1," should be "in BLA, but not vCA1,"

Response to reviewer comments

We sincerely thank the reviewers for their enthusiasm for our manuscript and their constructive feedback. In response, we have conducted new behavioral experiments, recordings, and analyses, which we believe have significantly strengthened our work. Below, we present a point-by-point response to each reviewer with detailed descriptions of experiments, analyses, and discussions that address each comment.

Referee #1 (Remarks to the Author):

This is an elegant study in which the authors used the chronic social defeat stress procedure to record signatures of stress susceptibility versus stress resilience across the brain's limbic circuitry. While other studies have reported signatures of stress susceptibility, the present investigation is the first to identify such clearly distinctive signatures of stress resilience. The present study is further distinguished by its identification of unique patterns of single neuron activity in the BLA that differentiate susceptible and resilient outcomes—this is the first characterization of such responses in the amygdala and the results are impressive. Another unique feature of this study is its focus on CA1-to-BLA projections which have not, as far as I know, previously been characterized for a role in vulnerability to chronic stress. (BLA-to-CA1 projections have been studied in fear conditioning which is very different.) The authors provide causal evidence, through the use of chemogenetics, to establish the role of BLA-projecting CA1 neurons in controlling susceptibility versus resilience to CSDS. Finally, I am especially impressed by the authors' use of relatively sophisticated behavioral outcome measures that begin to address psychological domains such as intention which are obviously challenging to access in a mouse. Importantly, they very nicely stick to experimental observations and do not over-interpret their findings.

All told, I am enthusiastic about this study and cannot identify additional experiments that are required to supplement the already extensive experimental work presented. The manuscript is well-written, the figures are clear, and the data are compelling. The authors also did an admirable job in citing the relevant literature.

My only request for clarification concerns the sex of the animals studied. In Methods, the authors say that they used males and females and sufficiently describe the CSDS procedures used for both sexes. However, it's not clear to me whether the data in all of the figures represent a combination of results from males and females (assuming the authors did not observe a sex difference in their main measures). This is totally fine—but it would be worth a sentence in the Results to this effect.

We thank the reviewer for the enthusiastic response to the study. We used both female and male mice in the study, and indeed, since we did not observe significant differences between sexes, we combined the results from both sexes for the results presented. We now include the data in Extended Data Fig. 1A-B where we analyzed males and females separately for our main behavioral measures. Specifically, we found that both male and female CSDS mice showed a significant correlation between sucrose preference and social interaction ratio. We have also now clarified in Results (lines 100-103) and Methods (lines 641-644), indicating that the data shown are combined from both males and females.

Referee #2 (Remarks to the Author):

The manuscript by Xia et al. describes neural signatures associated with an animal's stress susceptibility and resilience, which may underlie a clinical symptom of anhedonia in major depressive disorder. The authors examined the neural activity in the basolateral amygdala (BLA) and identified the significant differences in its representations of reward choices and intentions between stress-susceptible and resilient mice. Furthermore, the authors demonstrated the ensemble BLA activity when animals were at rest could be used to distinguish whether animals were subjected to chronic social defeat stress or not. Finally, the authors found that targeted manipulation of vCA1 inputs to the BLA in susceptible mice can rescue their abnormal features in both neural activity and behaviours. The manuscript addresses an important question in neuroscience and the results and analyses are overall well performed. However, in several parts of the manuscript, important control experiments and analyses are missing, which should be addressed before consideration of publication.

The following are detailed comments:

1. In Figure 2. The authors found that the activity of BLA neurons discriminates between sucrose and water trials, and then concluded that these neurons discriminate between reward choices (sucrose or water). However, because these choices are also associated with different actions (left and right licking), these results can also be explained from the aspect of differences in action planning and its neural coding. For example, I wonder if the authors monitored the same neurons before and after the change of spout designation to distinguish if the neurons are tuned to reward identity or action. Furthermore, it is also plausible that licking behaviour itself can directly influence the neural activity level in the BLA. If so, the difference in the proportion of reward-selective neurons between resilient and susceptible mice (Fig. 2C) could simply result from the difference in their licking rate (Fig. 1H). The authors should exclude these alternative possibilities.

We thank the reviewer for their interest in the study and for suggesting important control experiments. We have now performed a series of new experiments and analyses to

address the two key points brought up by the reviewer: 1) whether reward decoding can be explained by action planning (left vs. right), and 2) whether reward decoding differences are driven by lick behavior differences between resilient and susceptible mice.

To address point #1, the possibility that the enhanced reward in the resilient mice could be explained by differences in action planning and its neural coding in the stress groups, we generated new cohorts of animals in which mice went through the exact same CSDS procedure (or control, non-defeat) protocol, but then were recorded in the BLA in a reward-seeking task where both spouts delivered the same reward, sucrose (“Same reward” condition, as opposed to “Different reward” condition in sucrose preference test). Since the reward choice is equivalent between the 2 spouts in this scenario, this allowed us to assess the decoding accuracy of the direction of choice (left vs. right). We compared the decoding accuracy for the “Same reward” (sucrose vs. sucrose, i.e., left vs. right) to the “Different reward” (sucrose vs. water) condition. We found that during the Pre-reward period (-4 to -3s), decoding accuracy was at chance levels for the “Same reward” conditions for all groups of mice (control, resilient, and susceptible), suggesting that the future action chosen was not represented in the BLA at this time. Only in resilient mice, did we find that decoding accuracy for the Different reward conditions (sucrose vs. water) was greater than chance (and was significantly higher than Same reward), suggesting enhanced representation for future reward choice in resilient mice. During the Post-reward (0 to 1s) period, decoding accuracy for the “Different reward” was significantly greater than “Same reward” for all groups. These data are now presented in Extended Data Fig. 2A-C, Results (lines 152-155), and Methods (lines 1178-1181). This important control experiment suggests that the enhanced reward choice selectivity in resilient mice is due to the different reward values assigned to the two spouts rather than a general difference in action planning and associated neural coding in the BLA.

To address point #2, controlling for the possibility that reward choice decoding differences are driven by lick rate differences between resilient and susceptible mice during the Post-reward period (Fig. 1H), we performed a new analysis where we equalized the licking rate between resilient and susceptible mice during the Post-reward period (lick rates during the Pre-reward period did not differ between groups, Fig. 1H). Specifically, we limited our neural analysis to include only trials where the licking rates were not statistically different between the two groups (see Methods lines 1192-1196 and Extended Data Fig. 2D). We then trained a linear decoder to classify reward choice (sucrose vs. water) using only these trials where the lick rate was not statistically different between the two groups (see Methods lines 1192-1196). We found that after lick rate equalization, we again observed significantly greater decoding accuracy in the BLA of resilient mice compared to susceptible mice during both Pre-reward and Post-reward

time windows. These data are now presented in Extended Data Fig. 2D-E and Results (lines 152-155).

These important control experiments suggest that the differences in reward choice decoding that we observed are not likely to be driven by the direction of licking action or lick rate differences between groups.

2. Figure 3 shows the animal's intention-specific states - whether the mice switched or stayed on the previous choice. While the authors interpret the results as rumination-like states in these susceptible animals, I wonder if this could simply be a consequence of mice's indiscrimination between water and sucrose rewards. I think that the mice's decision process in this task would depend on the degrees of how much mice care about the reward type (water or sucrose) or the action type (right or left). When the difference in reward valence between choices is small, a mouse's choice strategy will rely more on action sequences, which may be observed as intention states in susceptible mice? I think the authors should explore this possibility, for example, by examining whether these intention-specific states emerge even in control mice when the two spouts provide the same reward.

We thank the reviewer for suggesting this important control experiment. To address the possibility that intention (stay/switch) decoding that we observed in the BLA may be driven by action sequence (left vs. right) as opposed to staying or switching depending on the previous reward type, we performed the experiments suggested by the reviewer: mice were exposed to CSDS (or control, non-defeat) then recordings were taken in the BLA as mice licked from the two spouts that delivered the same value reward (both sucrose). Since the reward value was the same between the two spouts, if we could decode intention, it would suggest that the action sequence (left vs. right) was encoded in BLA neurons. However, we found that decoding accuracy for intention was at chance level for all three groups. This was true if we used either raw firing rates or inferred neural activity of the hidden states (Hidden Markov Model) as inputs for the decoder. In line with this, the fraction of intention-specific states was at zero. This important control experiment suggests that when there is no difference in the reward value between choices, we cannot decode intention or detect intention-specific states in stress or control mice. In other words, BLA intention states (switch/stay) do not rely solely on action sequences but are dependent on the different reward types, a signature that is specific to susceptible mice. These data have been added to Extended Data Fig. 3J-L and Results (lines 218-220) of the revised manuscript.

3. The experiments in Fig. 4 examined neural activity in the absence of any stimuli or task demands under the head-fixed condition. However, the details of behavioural states of these mice are not described in the manuscript (or were all the movements restrained?). Because even spontaneous behaviors of mice (e.g. running, sleeping, staying still etc.) can largely affect the

brain's activity states and social defeat stress is known to increase an animal's anxiety level (Krishnan et al., 2007), it is important to examine whether these behavioral features during the head-fixed condition can explain the observed differences in neural activity or not.

We thank the reviewer for this suggestion. The reviewer is correct; mice were head-fixed, and all movements were restrained in a tube that covered the animal's body. As a result, mice could not freely move their body or their head. In addition, at the point of the experiment when recordings occurred, mice had been well habituated to the head fixation setup. In other words, mice could not run in our task and stayed still during the recordings. However, motivated by the reviewer's comment, we sought to determine whether other movements, such as facial features or limb movements could differ between control, susceptible, and resilient mice, and if so, whether such differences could contribute to the neural activity differences we observed during the spontaneous (Pre-task) period.

In order to address this possibility, we ran a new cohort of mice that went through CSDS (or control non-defeat) protocol as before. Then, while performing Neuropixels recording in the BLA, we acquired high-resolution, high-speed video recordings of mouse facial features and limb movements during the spontaneous (Pre-task) period. We identified and tracked 12 keypoint features that included the face and limbs (Extended Data Fig. 7A) throughout the 10-minute recording window using DeepLabCut.

First, we analyzed how facial and limb movements differ between groups. We analyzed 6 facial/limb features: eye-opening ratio, snout angle, mouth position, whisker position, and limb positions (Moene & Larsson, 2023). We found modest differences in eye-opening and limb movement in susceptible mice when compared to resilient and control groups. All other features analyzed did not differ between groups (Extended Data Fig. 7B-G).

We next tested whether we could decode the group identity of individual animals by training a classifier on facial/limb features. As done for neural data, we quantified facial/limb hidden states during the Pre-task using the Hidden Markov Model (HMM) and performed agglomerative clustering of states to identify those that were unique. We also assessed the embedding dimensionality of the representation using Principal Component Analysis (PCA) in the facial/limb feature space. We subsequently trained a classifier on the newly defined features, including the mean and standard deviation of each facial/limb feature, the PCA cumulative variance, and the fraction of clustered HMM facial/limb states, using the same cross-validation procedure as before. We found decoding group identity was slightly above chance for control vs. susceptible and

susceptible vs. resilient, and at chance for resilient vs. control (Extended Data Fig. 7H-I).

We then compared decoding using these spontaneous facial/limb features to neural decoding. We first confirmed that the new mice we recorded for this experiment replicated our original finding that, using BLA neural features during the Pre-task period, we could differentiate between control and susceptible mice very well. Specifically, a decoder trained on differentiating control vs. susceptible mice in the original dataset generalizes well to differentiating control vs. susceptible mice in the new dataset, to a similar performance level as a decoder trained on the new dataset alone (Referee 2 Fig. 1). In comparing facial/limb vs. neural decoding, we found that the decoding accuracy using facial/limb features (Extended Data Fig. 7I) was significantly lower than the decoding accuracy using neural features (Fig. 5E). Spontaneous neural activity in BLA also outperformed facial/limb features in differentiating whether mice have gone through the CSDS experience or not (Extended Data Fig. 7J).

Referee 2 Fig. 1. SVM decoder trained using Pre-task neural features of the original dataset to differentiate between control and susceptible mice generalizes well when tested on the new dataset, to a similar performance level as a decoder trained and tested on the new dataset alone.

As an additional control, we also compared neural to facial/limb decoding during only the time bins when facial/limb decoding is at chance. Specifically, we trained a decoder using the 6 facial/limb features to differentiate between control vs. susceptible mice in a subset of time bins (10 time bins, 1s each), and tested on the entire Pre-task time

window (360 time bins) of one held out testing mouse ($n = 1000$ cross-validations). We then selected the test time bins when facial/limb decoding is within 1 or 2 STD of chance distribution obtained from 100 null models. In these same time bins, we performed decoding of control vs. susceptible mice using neural activity of BLA. We found that even in these time bins when we cannot decode group identity using facial/limb features, we could still decode using neural features well above chance (Extended Data Fig. 7K).

Altogether, these results suggest that while there may be minor differences in spontaneous behavior in susceptible mice in comparison to the other groups, neural features in the BLA still greatly outperformed the facial/limb features in differentiating between control and susceptible mice. These data have been added to Extended Data Fig. 7, Results (lines 354-359) and Discussion (lines 432-440) of the revised manuscript.

Finally, we investigated whether these facial/limb features contributed to BLA neural activity. We fitted facial/limb features to neural activity (firing rate) using linear regression in each mouse separately. We binned neural and facial/limb feature data using 1s time window (total of $T = 360$ bins) and defined our model as $Y = AX^T + \beta$, where Y is a $N \times T$ matrix with the firing rate of N recorded neurons, A is a $N \times K$ matrix with the regression coefficients of $K = 6$ facial/limb features, and X is a $T \times K$ matrix with the K facial/limb features values. Beta (β) is the intercept (a constant). Before fitting, the data was centered to zero. We used the linear least square error as a loss function and added an L2-norm regularization term to prevent overfitting. We tried many values for the L2-norm regularization term, ranging from 0 (equivalent to ordinary least squares) to 10^3 , with no significant difference in the final coefficient of determination (R^2) estimate.

We found that in all mice across all groups, there was not a positive R^2 value from the linear model, suggesting that using facial/limb features that we recorded, we could not predict BLA neural activity better than chance. In other words, these facial/limb features were unlikely to contribute significantly to BLA neural activity, and consequently, any group differences that we observed. It may be possible that other facial/limb features not captured here may contribute to BLA activity, and it may also be possible that the small differences in some facial/limb features we observed could be encoded elsewhere in the brain that was not recorded. However, as far as we could tell, the tracked keypoints during the spontaneous (Pre-task) period could not explain BLA neural activity. We include this analysis in Results (lines 354-359), Discussion (lines 432-440), and Methods (lines 965-979).

Furthermore, if facial/limb movement differences were a key contributor to neural activity differences we observed between groups, we would likely see such differences

manifest across different regions such as the hippocampus, as behavior tends to be encoded across many regions (Stringer et al., Science, 2019). However, the neural activity differences we observed during the spontaneous period were only present in the BLA of susceptible mice. Altogether, our results suggest that the neural signature of susceptibility that we observed in the BLA during the Pre-task period was not likely to be related to differences in the movements of the face and limbs in the head-fixed mice.

4. In Fig 4G, the main claim is that the activity patterns in the BLA at rest can serve as a powerful biomarker to distinguish whether mice had chronic stress or not. Here, my confusion is that the compared behavioural parameters used to distinguish between control and stress-exposed mice were the lick rate, sucrose preference, and social interaction, which are the features used to distinguish resilient and susceptible mice after social defeat stress (Fig. 1). I thus feel that this conclusion may be misleading because these compared behavioural features are probably not optimal for distinguishing mice's stress-experience, and indeed, a previous study identified the other behavioural feature that is commonly observed in both susceptible and resilient mice after social defeat stress (i.e., increased anxiety-like behaviour in the elevated plus maze; Krishnan et al., 2007). If the authors want to claim the superiority in classification performance by neural signatures, it would be fair to compare with this defined behavioural feature in its ability to distinguish mice with stress experience.

We thank the reviewer for this important suggestion and agree that anxiety-like behavior should indeed be able to distinguish mice's stress experience and should be included in our analysis. We thus assessed anxiety-like behavior using the elevated plus maze as suggested by the reviewer. We found that in line with previous findings (Krishnan et al., Cell, 2007), both susceptible and resilient mice showed increased anxiety (reduced % open arm exploration time) in comparison to controls (Extended Data Fig. 1C). We then included this measure of anxiety-related behavior as one of the behavioral features for decoding an animal's group identity. We found that while we could decode control vs. CSDS mice with ~65% decoding accuracy (Fig. 5G), decoding using BLA neural features still outperformed behavior decoder, with an accuracy of ~80% (Fig. 5G). In addition, decoding of control vs. susceptible mice using BLA neural features (Fig. 5E) also outperformed behavioral feature decoding (Extended Data Fig. 1D-E). This suggests that BLA neural activity patterns are more powerful biomarkers to distinguish whether mice had chronic stress or not than using behavioral features that include anxiety-related behavior, social avoidance, and anhedonia-related behaviors. We have now updated Fig. 5G and Extended Data Fig. 1D-E to reflect the inclusion of the elevated plus maze in behavioral decoding.

5. In Fig. 4E, the authors tested whether the decoder based on neural activity at rest can distinguish between stress-susceptible and resilient animals (not necessarily good performance,

though). But it would be important to perform a similar analysis by using the authors' other identified neural signatures during the task phase (representations of reward choice and intention), which would be a critical test of the usefulness of these neural signatures in distinguishing mice with stress susceptibility and resilience without relying on behavioural features.

We thank the reviewer for this suggestion. As suggested, we performed the decoding analysis using neural signatures of the task phase, specifically reward choice decoding during the Pre- and Post-reward period, and the fraction of intention-selective states. We found that we could decode susceptible vs. resilient mice, as well as susceptible vs. control mice very well, much greater than chance, while decoding of resilient vs. control mice remains at chance level (Extended Data Fig. 3O). The results suggest that similar to the spontaneous Pre-task period, BLA neural signatures during the task period are powerful biomarkers in distinguishing control and susceptible mice. Furthermore, as the reviewer pointed out, using on-task neural signatures, we could decode susceptible vs. resilient mice better than during the spontaneous Pre-task period. This suggests that neural activity patterns that emerge as a result of task demands may help further differentiate between susceptible and resilient animals. These data have been added to Results (lines 223-225) of the revised manuscript.

6. For Fig. 5F, in the same line as my previous comment 1, it is important to show that the increased decoding performance is not a simple consequence of the difference in licking rates between water and sucrose because licking behaviour itself might affect firing rates directly.

We thank the reviewer for suggesting this important control analysis. We performed a similar analysis as the one in response to comment 1, where we performed the decoding of reward choice using only the subset of trials where the lick rates are not statistically different between saline- and CNO-treated susceptible mice (Extended Data Fig. 4L). We found a similar pattern of reward choice decoding performance as the original data without lick rate equalization, where CNO-treated susceptible mice showed enhanced reward choice decoding accuracy during the Post-reward period (Extended Data Fig. 4M). The results suggest that the reward choice decoding differences we observed before and after CNO were unlikely due to lick rate changes. Details of this new analysis have been added to Methods (lines 1192-1196) of the revised manuscript.

7. In Fig. 5HI, although the authors consider the impaired decoding of stay/switch intention after the CNO injection as evidence of rescued abnormality, when considering the DREADDs enhancer is often used as activity perturbation, the disturbance in certain neural representations and resulting impairment in decoding ability would naturally be expected by this manipulation. I would thus consider it necessary for the authors to show the specificity of the manipulation on a

particular neural feature, rather than its general disturbance of neural representations. For example, I wonder whether hidden-state representations in the BLA behave in the same way as control and resilient animals during the task (Fig. 3) and the rest (Fig. 4) after the CNO injection to susceptible mice?

We agree with the reviewer that it is possible that DREADD-induced changes in neural activity patterns could disturb the circuits indiscriminately, however, when examining all the neural features, we found that the effects that we observed with CNO were specific to select neural features. First, we found that, rather than a general impairment, CNO actually enhanced reward choice decoding in the BLA and vCA1 (Fig. 4F-G). Second, when examining intention decoding, we found the opposite effect, where CNO reduced intention decoding and intention-selective states (Fig. 4H-K). Third, CNO increased firing rates in vCA1 neurons, but had no effect on overall firing rates in the BLA (Extended Data Fig. 4B, K). Fourth, we did not observe any effects of CNO on the dimensionality and hidden states during the Pre-task period, including cumulative variance of the first 3 PCs and the % or the number of HMM hidden states across varying clustering thresholds (Extended Data Fig. 4C-I). These results suggest that the DREADD enhancer is not simply disturbing all neural representations and impairing decoding ability irrespective of circuit or function.

With regards to hidden states, as the reviewer speculated, we indeed observed that CNO reduced the decoding accuracy and fraction of intention-selective hidden states, a neural signature that was uniquely present in BLA of susceptible mice (but not in controls and resilient mice, Fig. 4H-K). In the revised manuscript we discuss these points in Results (lines 276-289) and Discussion (lines 408-413).

8. In Fig. 5L, the authors only tested the sucrose preference as a behavioural feature rescued by the manipulation, which I do not think is sufficient, and they should also examine the social interaction score to align with the definition of stress susceptibility (Fig. 1). It would also be ideal if the authors construct a classifier of animal susceptibility/resilience using either behavioural features or neural signatures, and examine whether the DREADD manipulation can change the decoder's classification of the treated mice from stress susceptible to resilient.

We thank the reviewer for these interesting suggestions. In our original experiment, the sucrose preference test occurred after the social interaction test, and since CNO was given online during the second half of the sucrose preference test, we could not assess the effect of CNO on the social interaction test. To address how activation of vCA1->BLA pathway may influence social interaction behavior, we generated new cohorts of mice where we expressed the DREADD enhancer in the BLA-projecting vCA1 neurons as in the original experiment (AAV-DIO-hM3Dq in vCA1, and AAV-retroCre in BLA) and then put mice through the CSDS (or control, non-defeat) protocol for 10 days. We then

assessed their behavior during the social interaction test with and without CNO. Specifically, we performed 2 days of social interaction tests, where mice received saline injection (i.p.) 20 min prior to the test on day 1, and CNO injection 20 min prior to the test on day 2. We found that CNO increased the social interaction time of susceptible mice in the presence of CD1 mice, in comparison to the saline-treated session. These data are now presented in Extended Data Fig. 4W-X and Results (lines 295-296).

We also performed the interesting analysis suggested by the reviewer; we built a classifier to decode whether an animal is susceptible or resilient, and then examined how well the decoder generalizes to susceptible mice before and after the treatment of CNO. We trained a linear classifier in the feature space defined by the 3 behavioral features (sucrose preference, lick rate discrimination index in the Pre- and Post-reward) and the 4 neural features (reward decoding accuracy in the Pre- and Post-reward, the intention decoding accuracy Pre-reward using raw firing rates, and the intention decoding accuracy Pre-reward using HMM states). Notably, the cross-validated decoding performance of susceptible/resilient is 100% and the generalization accuracy to saline/CNO is ~ 90%. The results suggest that activation of the vCA1->BLA pathway reversed the susceptibility phenotype to be more similar to resilient mice. These data are now presented in Extended Data Fig. 4P-R and Results (lines 289-292).

9. CNO has been shown to have a nonspecific impact through its metabolite clozapine (e.g. Gomez et al., 2017). While the authors confirmed that CNO alone (without hM3D) did not affect the animal's sucrose preference (Ext Data Fig. 5R), similar control experiments would be necessary for other behavioural features (Fig. 1) and neural signatures (Fig. 5F-K), as the impact of clozapine is uncertain here.

We thank the reviewer for the suggestion. As requested, we generated new cohorts of mice where we expressed the control construct (AAV-DIO-mCherry in vCA1, and AAV-retroCre in BLA), and then put mice through CSDS and the exact same behavioral and neural recording protocols as the hM3Dq cohort in the original experiment (Fig. 4). Behaviorally, we found that in contrast to the hM3Dq mice, mCherry-infused mice showed no significant differences in sucrose preference, lick rates for sucrose and water trials, and lick rate discrimination index before and after CNO treatment. We then assessed neural signatures with Neuropixels recordings. We found that in contrast to the hM3Dq cohort, in the mCherry-infused mice, CNO did not have any effect on the decoding accuracy of reward choice, or the decoding accuracy of stay vs switch intention. Altogether, these important control experiments and analyses suggest that CNO does not have nonspecific effects on behavioral or neural signatures in the absence of the DREADD receptor (hM3Dq). These data are now presented in Extended Data Fig.5 and Results (lines 296-297).

Referee #3 (Remarks to the Author):

This innovative and important study effectively couples behavior, neural recordings and powerful computational analytical approaches to provide genuinely novel insight into the differential effects of (social defeat) stress on neuronal population dynamics in the basolateral amygdala and ventral hippocampus. It's also noteworthy that although the computational treatment of the data is unusually comprehensive for a mouse behavioral study, the authors are to be commended for doing a good job of keeping the main text accessible to a broad readership. Nonetheless, there are several areas where further clarification is needed.

The observation that BLA population activity exhibits higher dimensionality in stress-susceptible mice (and cross-group decodability) during periods in which there are no explicit task demands is very interesting. As this phenotype is evident on the period in which the mouse is head-fixed and awaiting the start of the task ('stimulus-free pre-task condition'), can it be solely attributable to neuronal activity 'at rest,' or is there potentially an element of expectation about the upcoming task demands? Given the susceptible mice exhibit evidence of 'rumination-like' activity on-task, this may be a non-trivial consideration – and will partly depend on how much task-training the mice have had by this point. Whatever is driving the phenotype makes it no less interesting, but it would be good to get some clarity or discussion of this.

We thank the reviewer for their enthusiastic response to our study. The increased dimensionality in the BLA of susceptible mice during the Pre-task period is indeed a compelling neural signature of susceptibility. At this point in the task, the mice were well-habituated to head fixation and well-trained in the sucrose preference task. As the reviewer suggested, the higher dimensionality during the Pre-task period might reflect an expectation or rumination about the upcoming task demands. Indeed, we agree with the reviewer that this increased dimensionality and higher number of states during the Pre-task period could be linked to the emergence of rumination-like (or intention-selective) states observed in the same susceptible animals during the task.

We speculate that under normal conditions, such as in control mice, the BLA plays a crucial role in evaluating reward values, which subsequently influences the decision to switch or stay. The decision likely occurs downstream of the BLA, because we cannot decode the intention to switch or stay in the BLA of control mice. However, in susceptible mice, the BLA's ability to evaluate reward values may be disrupted by the emergence of these intrusive, intention-selective states, which we could decode clearly in these mice. These intrusive states may interfere with downstream activity, biasing the decision to switch or stay towards the lower value reward.

Interestingly, these intrusive states are not merely noise, as we can decode the signal as the intention to switch or stay. The ultimate effects on decision-making are likely probabilistic, with the downstream region reading out all states (both normal and

intrusive) from the BLA. In susceptible mice, these intrusive states may sometimes increase the probability of staying, while in other instances, they may increase the probability of switching. Consequently, susceptible mice exhibit an aberrant reward decision-making process, resulting in anhedonia. A similar process might govern the Pre-task period in the absence of reward stimuli, where the higher dimensionality reflects additional intrusive states in susceptible mice.

We have now added these interesting discussion points in the revised manuscript on lines 414-431.

The enhanced representation of upcoming responses based on past choice in the stress-susceptible mice raises the perennial issue with behavior-neuronal correlations as to the extent to which the neuronal activity reflects differences in movement rather than/in addition to intention (e.g., do stress-susceptible mice have slower response latencies during these trials than could impact the presentations). Please address this question.

We thank the reviewer for bringing up this important point. As suggested, we analyzed the lick rates during switch vs. stay trials across groups during the 4s Pre-reward period (same time window used to identify intention-selective states), and we did not observe any differences between groups. This suggests that intention decoding in susceptible mice was unlikely due to lick rate differences. These data have been added to Extended Data Fig. 3M and Results (lines 218-220) of the revised manuscript.

We also have performed other control experiments and analyses to rule out that the intention signals we have recorded were not driven by differences in lick rates or lick action plans. This was also addressed above in response to Reviewer 2, Point 2. For ease of reading, we have copied over the response from above below:

To address the possibility that intention (stay/switch) decoding that we observed in the BLA may be driven by action sequence (left vs. right) as opposed to staying or switching depending on the previous reward type, we performed the experiments suggested by the reviewer: mice were exposed to CSDS (or control, non-defeat) then recordings were taken in the BLA as mice licked from the two spouts that delivered the same value reward (both sucrose, thus the “Same reward” condition, as opposed to “Different reward” condition in sucrose preference test). Since the reward value was the same between the two spouts, if we could decode intention, it would suggest that the action sequence (left vs. right) was encoded in BLA neurons. However, we found that decoding accuracy for intention was at chance level for all three groups. This was true if we used either raw firing rates or inferred neural activity of the hidden states (Hidden Markov Model) as inputs for the decoder. In line with this, the fraction of intention-specific states was virtually zero. This important control experiment suggests that when there is no

difference in the reward value between choices, we cannot decode intention or detect intention-specific states in stress or control mice. In other words, BLA intention states (switch/stay) do not rely solely on action sequences but are dependent on the different reward types, a signature that is specific to susceptible mice. These data have been added to Extended Data Fig. 3J-L and Results (lines 218-220) of the revised manuscript.

We also assessed reward choice decoding and found that the enhanced reward choice representations in resilient mice were not driven by action planning or lick rate differences between groups. These data are now presented in Extended Data Fig. 2, Results (lines 152-155).

These important control experiments suggest that the differences in intention and reward choice decoding that we observed were not likely to be driven by the direction of licking action or lick rate differences between groups.

The hippocampus-BLA chemogenetic stimulation representation-rescue experiment nicely sews up the narrative. Were these effects associated with a change in BLA neuronal activity (as they were in hippocampus)?

The activation of the BLA-projecting vCA1 neurons resulted in an increase in firing rates in vCA1 neurons, but not BLA neurons (Extended Data Fig. 4B, K). At the population level, we observed that activation of the vCA1->BLA pathway resulted in an increase in the decoding accuracy of reward choice (Fig. 4F-G), and abolishment of the ability to decode intention in both the BLA and vCA1 (Fig. 4H-I). We have now further clarified the findings in the Results (lines 276-292) and Discussion (lines 408-413).

Please provide an explanatory note for the reader for cases in which a subsample of neurons were analyzed (e.g., due to matching population size across groups). Also for clarification, my reading is that the definition of reward-selective neurons is that they are preferentially responses to either sucrose or water (rather than sucrose-preferring) – if this is the case please make it explicit for the reader because ‘reward-selective’ implies selectivity for the higher value reward option (i.e., sucrose).

We apologize for the lack of clarity on subsampling, and we have now further clarified the procedure in the Methods section (lines 857-859, lines 1182-1188) on how we subsampled neurons to account for the difference in population size across groups. Specifically, we first found the minimum number of neurons across groups in a region of interest. Then, we subsampled all the groups to that minimum number, and repeated the subsampling of neurons 10 times. The final reported decoding accuracy was the mean and distribution across the 10 random subsampling of neuron populations.

As the reviewer also pointed out, reward-selective neurons are defined as those that showed statistically significant response to either sucrose or water rewards across trials, and are not reward-type-specific or selective for just the higher value reward. To clarify, we have now made sure to consistently use the term “reward-choice-selective” neurons throughout the revised manuscript.

Line 804: ‘activity’ not ‘acidity’

We have corrected this typo.

Referee #4 (Remarks to the Author):

Xia et al. uncovered neuronal mechanisms underlying anhedonia. By applying high density spike recordings to stress mouse models, they identified reward-related and intention-related neuronal activity in BLA and vCA1 specific to susceptible mice, suggesting a rumination-like signature. BLA spike patterns in such susceptible mice showed larger numbers of spike activity states, correlated with their susceptibility. Chemogenetic activation of vCA1 inputs to the BLA in susceptible mice restored these neural dynamics and anhedonic behavior.

Overall, the experiments are well designed and provide novel findings. I am enthusiastic about this manuscript. Data and methodology is valid, but some experimental conditions need more explanation. Some figures were a bit too small, making the presentation hard to follow. The statistical values were appropriately used and reported. The conclusion is clear, but there were some challenging results to interpret, such as in Figure 4. The references were appropriate. I have summarized additional comments below.

(Major)

(1) What is meant by 'control'? I could not find a description regarding this. In the context of the CSDS model, 'control' should typically refer to conditions such as segregating mice within the same cage without physical contact. Is this the intended meaning, or does it refer to 'naive' mice with no interventions?

We thank the reviewer for the enthusiastic response to our study. We apologize for the confusion about control mice. As the reviewer pointed out, control mice are indeed BL/6 mice that were co-housed across a divider from another BL/6 mouse for 10 days, without any defeat by CD1s or physical contact with other mice. We have now further clarified this in the Methods on lines 689-691.

(2) Were there any differences in behavioral outcomes between male and female mice (Fig. 1G)? If resilient and susceptible phenotypes can be simply distinguished based on gender, the interpretation of the results in this manuscript could significantly change, necessitating clarification.

We agree with the reviewer that it is important to clarify the potential differences in behavior outcomes between male and female mice. We performed our experiments using both male and female mice and assessed their behavior including the sucrose preference and social interaction ratio. We found that sucrose preference and the social interaction ratio were significantly correlated in the CSDS group in both males and females. Given that male and female mice showed similar responses to CSDS as assessed using our two main behavioral metrics, we pooled together all the mice for subsequent analysis. We have now included this new analysis in the updated manuscript in Extended Data Fig. 1A-B, and further clarified in Results (lines 100-103) and Methods (lines 641-644).

(3) Figure 1F indicates that many susceptible mice show a preference for water intentionally, with 50% or less preference for sucrose. Indeed, as observed in Fig. 3B, susceptible mice exhibit a clear increase in WW (water-water preference) and a decrease in SS (sucrose-sucrose preference). Is this intentional avoidance of sucrose known as a sign of anhedonia?

As the reviewer pointed out, susceptible mice showed a reduced preference for sucrose and an increased preference for water. This may be an indication of avoidance of the higher value reward, which is indeed a known signature of anhedonia. For example, depressed individuals did not develop a reward response bias for higher reward value options in a reinforcement paradigm. In addition, depression patients showed reduced positive emotions in anticipation of positive (monetary) rewards, but not to less rewarding or non-rewarding stimuli (Treadway & Zald, 2011, Pizzagalli et al., 2005). The results suggest that depression may be related to both decreased motivation to seek out high-value rewards and experience of pleasure, leading to this avoidance of higher value reward. We have now added this in the revised manuscript on lines 104-107.

(4) In addition, if this is a prominent phenotype in susceptible mice, it suggests that WW and SS may have significantly different meanings for the mice. In that case, beyond combining them into an overall measure like 'intension,' it is necessary to conduct a more detailed analysis and provide an explanation, especially regarding the differences between WW (intentional reward-avoiding) and SS, to determine if there are variations in the representations of BLA neural activity.

We agree with the reviewer that the intention in WW and SS trials may be different for susceptible mice, that they may be intentionally avoiding sucrose rewards while

preferring water. We assessed the decoding of SS vs. WW trials during the 4s Pre-reward period. As the reviewer correctly predicted, we found higher decoding accuracy in susceptible mice, in comparison to control and resilient mice (Referee 4 Fig. 1). However, it is unclear whether the enhanced decoding accuracy can be purely attributed to different intention states in SS vs. WW in susceptible mice. The reason is that SS and WW trials differ in 2 aspects: 1) intention (intention to continuously stay on or switch away from sucrose or water), and 2) reward value (reward value of the previous and/or upcoming trial). In other words, we cannot disentangle the intention vs. the reward value signal when decoding SS vs. WW. That is why we combined trial types when decoding intention (switch = SW + WS, stay = WW + SS) to identify the intention to stay or switch from the previous to the current reward, irrespective of the specific reward value. In other words, in the way our experiment has been designed, we cannot distinguish between the intention to seek/avoid a specific reward type (sucrose or water) from the difference in the reward value of sucrose vs. water. Thus, the intention signal that we defined and observed in susceptible mice is an intention to switch away or stay on the same reward as the previous trial (irrespective of the specific reward value). The finding is in line with the idea that depression patients tend to ruminate more about potential choices prior to making a decision. We have now further clarified the trial-type design rationale in Results (lines 192-194) and Methods (lines (1174-1177)).

Referee 4 Fig. 1. Decoding accuracy of SS vs. WW trials in the BLA during the Pre-reward period. Susceptible mice showed better decoding accuracy than control and resilient mice (Kruskal-Wallis, $** P < 0.01$). Data are mean \pm SEM.

(5) Line 194-200: This paragraph might be challenging for someone not analytically inclined. As researchers interested in anhedonia and behavioral neuroscience are likely to be in the minority when it comes to those well-versed in such analyses, a more detailed explanation and interpretation from the analysis would be beneficial.

We thank the reviewer for pointing this out. We have further clarified this section in the revised manuscript (lines 205-209).

(6) From Line 238 and Figure 4: I could not understand how the story on spontaneous activities connects with the other task sections. Certainly, spontaneous activity is an intriguing aspect of brain function and could potentially serve as one of the physiological markers, as suggested by the authors. However, discussions and figures from 1 to 3 primarily focus on tasks, and Figure 5 specifically addresses the impact of chemogenetics on task performance. Furthermore, CNO had little impact on the spontaneous activity of the BLA, as shown in Extended Fig. 5I and 5J. Take together, the data on spontaneous activity is likely to be clearer if presented separately from the main task-related data, perhaps following all the data on the tasks.

We thank the reviewer for this suggestion on how to make the presentation of data clearer. As suggested, we have reorganized the manuscript, such that now we present all of the task data (including the DREADD experiment) first, and then show the data for the Pre-task period.

(7) Related to (6), the authors mentioned “functional connectivity” at Line 241. However, no tests on inter-regional correlations or coherent analyses between BLA and vHC are provided. To provide a more comprehensive description of such resting-state activity, it is crucial to investigate the extent of spike (or LFP) (cross-)correlations among neurons related to reward and intention in the vHC and BLA for each group. These analyses might contribute to strengthening the authors' hypothesis regarding the difference in the stress group between vCA1 and BLA (described at Line 388-393).

We thank the reviewer for the suggestion. Based on the reviewer's suggestions, we have further investigated functional connectivity between simultaneously recorded BLA and vCA1 neurons during the resting state (Pre-task period), intention-selective states, and reward choice signal.

To analyze the interaction during the resting state, we computed the firing rates of each recorded neuron in 1s bins within the same region (BLA and vCA1) during 6 minute window (min 2-8) of Pre-task recording. We set a minimum of 5 neurons simultaneously recorded in both BLA and vCA1. Given the matrix $N \times T$, where N neurons and T time bins, we computed the Principal Components (PCs) of this matrix for each area,

separately. We subsequently aligned the neural dynamics of the first PC between the two simultaneously recorded signals by employing Canonical Correlation Analysis (CCA). CCA is a linear transformation used to find common patterns between two signals defined in two different spaces, with the goal of maximizing their correlation. Given two matrices $X \in \mathbb{R}^{N \times T}$ and $Y \in \mathbb{R}^{M \times T}$, where N and M are the numbers of variables, and T is the number of time bins, CCA finds linear combinations U and V of the features in X and Y such that,

$$\begin{aligned} U &= a^T X, \\ V &= b^T Y, \end{aligned}$$

where the coefficients $a \in \mathbb{R}^{N \times 1}$ and $b \in \mathbb{R}^{M \times 1}$ are chosen to maximize the correlation between U and V . We refer to U and V as canonical components (CCs) for BLA and vCA1, respectively. We subsequently analyzed the cross-correlogram between the first CC component of BLA and vCA1 with time lags from (-50, +50) s and its corresponding power spectral density (PSD). We computed the PSD from the squared magnitude of the Fast Fourier Transform coefficients divided by the length of the input signal. We used the frequency at which the PSD peaked as an estimate of the dominant frequency of the oscillations between BLA and vCA1. The highest frequency we could access is determined by the Nyquist frequency $f = f_s/2 = 0.5\text{Hz}$, where f_s is the sampling frequency that in our case is 1 Hz. We tested smaller time bin sizes and chose 1s bins (hence 1Hz sampling frequency) due to low firing rates during the Pre-task period, which would otherwise result in many bins with 0 spikes/second.

We found that CSDS mice showed a reduced dominant frequency for BLA-vCA1 interaction than control mice (Extended Data Fig. 6S-T). This suggests that chronic stress altered the mode of communication between the two regions that could affect information transfer and processing. We now present this data in Results (lines 325-328).

We have also investigated the functional connectivity during task. For correlation analysis during task, we could not use the same CCA method to determine frequencies of interaction given the short time window ($\leq 4\text{s}$). Thus, to analyze the BLA-vCA1 interaction with respect to reward, we computed the correlation between regional average firing rates in simultaneously recorded BLA and vCA1 neurons (Fig. 4A). We found that resilience is positively correlated with how strongly BLA and vCA1 are interacting during sucrose reward trials in comparison to water trials (Fig. 4B). This led us to speculate that boosting the interaction between the two regions may reverse behavioral and neural signatures of anhedonia in susceptible mice. Indeed, that is what we found by activating specifically the BLA-projecting vCA1 neurons (Fig. 4).

In addition, we have also explored the functional connectivity between the two regions with respect to the intention to switch or stay. Here, we computed the correlation of BLA activity to the vCA1 during periods of intention-selective states in the BLA vs. periods of non-intention-selective states in susceptible mice. We focused our analysis on susceptible mice because control and resilient mice have very few intention-selective states. We found that BLA-vCA1 correlation was enhanced during intention-selective states, in comparison to non-intention-selective states (Extended Data Fig. 3N). As vCA1 and BLA are reciprocally connected, this would be expected as we could detect an intention signal in both vCA1 and BLA (albeit with much stronger in the BLA), suggesting some correlation in activity between the two regions during these time windows.

We hypothesize that in the BLA, estimation of reward values vs. intention to switch or stay may represent two distinct modes during reward decision-making; the former is dominant in control and resilient mice, and the latter is dominant in susceptible mice. We hypothesize that these intention-selective states are intrusive and disrupt normal decision-making in susceptible mice to promote anhedonic responding, as they are not present in control and resilient mice (see more discussion in response to Q8 below). Chemogenetic stimulation of BLA-projecting vCA1 neurons in susceptible mice disrupted the encoding of the intention to switch vs. stay in BLA and vCA1 (Fig. 4H-I), allowing for better reward value coding and reward-related information transfer between BLA and vCA1 (enhanced correlation of BLA-vCA1 firing prior to high-value reward choices, Fig. 4B). In addition, it may also be possible that when reward values are more distinctly represented, mice may rely less on intention and more on reward value for decision-making. We have now added these discussion points in the revised manuscript on lines 441-452.

(8) Related to (6), Fig. 4D: It is hard to interpret why a greater number of distinct neural population states of spontaneous activity in susceptible mice are related to anhedonic behavior and intention specific states in the task. Is it possible for the authors to provide some discussion on this?

We think that the increased dimensionality and higher number of states during the spontaneous activity period could be linked to the emergence of rumination-like (some of which could be intention-selective) states observed in the same susceptible animals during the task.

We speculate that under normal conditions, such as in control mice, the BLA plays a crucial role in evaluating reward values, which subsequently influences the decision to switch or stay. The decision likely occurs downstream of the BLA, because we cannot decode the intention to switch or stay in the BLA of control mice. However, in

susceptible mice, the BLA's ability to evaluate reward values may be disrupted by the emergence of intention-selective states, which we could decode clearly in these mice. These intrusive states may interfere with downstream activity, biasing the decision to switch or stay towards the lower value reward.

Interestingly, these intrusive states are not merely noise, as we can decode the signal as the intention to switch or stay. The ultimate effects on decision-making are likely probabilistic, with the downstream region reading out all states (both normal and intrusive) from the BLA. In susceptible mice, these intrusive states may sometimes increase the probability to stay, while in other instances, they may increase the probability to switch. Consequently, susceptible mice exhibit an aberrant reward decision-making process, resulting in anhedonia. A similar process might govern the Pre-task period in the absence of reward stimuli, where the higher dimensionality reflect additional intrusive states in susceptible mice.

We have now added these interesting discussion points in the revised manuscript on lines 414-431.

(9) Fig. 5A-B: This study simply subtracted correlations between sucrose and water. As the correlation coefficient is not an additive measure, this subtraction is mathematically not ideal. For example, please consider to use other more reliable measures such as Fisher's Z transformation, in which Pearson correlation coefficients (cofiring) were transformed into Z value that can be used to calculate a confidence interval for coefficients.

We thank the reviewer for this suggestion. We performed the analysis as suggested, we performed the Fisher's Z transformation on the BLA-vCA1 correlation for sucrose and water trials, prior to performing the subtraction. We again found that the mice with greater BLA-vCA1 correlation in sucrose vs. water trials were more resilient (higher SPT*SI score). We have updated the figure (updated manuscript Fig. 4B) and the Methods (lines 1328-1331) in the revised manuscript.

(Minor)

(1) As this recording is acute, not chronic (implantation), recording, please mention this more clearly in the first Results section.

We clarified in the first Results section that the Neuropixels recordings were acute (lines 97-98).

(2) Fig. 1D: I could not correctly read this rasterplot. Why plots from multiple cells seem to be combined in time? Some spaces between cells may be needed.

We have added more spaces between the cells to make the plots more clear.

(3) Fig. 1I (and some figures): they say “Controls”. How it was different from “Control”?

We apologize for the confusion. “Controls” are the same as “Control” mice. To make the language consistent, we have now changed all figures to be “Control” mice.

(4) Fig. 2B: Please explain what is “auROC (acronym)” in the manuscript.

We have now explained the acronym in the figure legend.

(5) Fig. 2B: The claim of these traces is that water (blue) and sucrose (magenta) trials are well separated in resilient mice. But it is not clear from these traces. Please consider to provide more representative traces that better illustrate this claim.

We have updated the example traces for the resilient mice so it is more visually clear that the water and sucrose trials are well-separated.

(6) Fig. 3A-E (and overall figures): The use of colors is confusing. For instance, in Fig. 3B, 'control' is represented in purple and 'resilient' in orange, while in Fig. 3C, 'stay trial' is in purple and 'switch trial' in orange. Furthermore, in Fig. 3H, orange and cyan are used for another purpose. Furthermore, some bar graphs (e.g. Fig. 2C, 3B, 3L...) use gray for control while it is represented in purple in the legends. Throughout the manuscript, please ensure a more coherent and consistent use of colors for better clarity.

We have updated the color schemes to make the different groups clearer.

(7) Fig. 3H, 3J, 3K and Extended Fig. 3G: it was hard to understand what these graphs of HMM models meant (while their conclusions are understandable). Please add more explanations.

We have now added more explanation to the respective figure legends, and also updated the schematic in Fig. 3G to better explain the decoding process using HMM hidden states.

(8) Fig. 4C and 4D: Why the Y-axis (%) is so different between these two graphs?

We apologize for the confusion. In Fig. 4C (updated manuscript Fig. 5C), we are showing on Y-axis, the distribution of % clusters retained across all 1-p thresholds (X-axis, from 0 to 1). In Fig. 4D (updated manuscript Fig. 5D), we are showing the fraction of clusters

retained at threshold of $1-p = 0.5$, and correlating the values at that threshold to the behavior of mice. We have updated the Y-axis labels and further clarified the threshold in the figure legend (now Fig. 5C-D).

(9) Fig. 5D: the word “DAPI (?)” is hard to see.

We changed the color so “DAPI” is now more visible (now Fig. 4D).

(10) Fig. 5I legends: “n = 100” (spaces needed)

We added spaces as suggested (now Fig. 4I).

(11) Fig. 5L: gray lines are too thin.

We increased the thickness of the gray lines as suggested (now Fig. 4L).

(12) Line 321: should be “Fig. 5J” (period needed)

We corrected the typo.

(13) Line 660: “4 C” (misspelling)

We corrected the typo.

(14) Line 803: “neural acidity”?

We corrected the typo. It should be “neural activity”.